# Non-Asymptotic Analysis for Single-Loop (Natural) Actor-Critic with Compatible Function Approximation

## Abstract

Actor-critic (AC) is a powerful method for learning an optimal policy in reinforcement learning, where the critic uses algorithms, e.g., temporal difference (TD) learning with function approximation, to evaluate the current policy and the actor updates the policy along an approximate gradient direction using information from the critic. This paper provides the *tightest* non-asymptotic convergence bounds for both the AC and natural AC (NAC) algorithms. Specifically, existing studies show that AC converges to an $\epsilon + \varepsilon_{\text{critic}}$ neighborhood of stationary points with the best known sample complexity of $\mathcal{O}(\epsilon^{-2})$ (up to a log factor), and NAC converges to an $\epsilon + \varepsilon_{\text{critic}} + \sqrt{\varepsilon_{\text{actor}}}$ neighborhood of the global optimum with the best known sample complexity of $\mathcal{O}(\epsilon^{-3})$, where $\varepsilon_{\text{critic}}$ is the approximation error of the critic and $\varepsilon_{\text{actor}}$ is the approximation error induced by the insufficient expressive power of the parameterized policy class. This paper analyzes the convergence of both AC and NAC algorithms with compatible function approximation. Our analysis eliminates the term $\varepsilon_{\text{critic}}$ from the error bounds while still achieving the best known sample complexities. Moreover, we focus on the challenging single-loop setting with a single Markovian sample trajectory. Our major technical novelty lies in analyzing the stochastic bias due to policy-dependent and time-varying compatible function approximation in the critic, and handling the non-ergodicity of the MDP due to the single Markovian sample trajectory.

## 1 Introduction

Actor-Critic (AC) (Barto et al., 1983; Konda & Tsitsiklis, 2003) is a reinforcement learning algorithm that combines the advantages of actor-only methods and critic-only methods by alternatively performing policy gradient update (actor) and action-value function estimation (critic) in an online fashion. Specifically, the critic uses a parameterized function to estimate the value function of the current policy, e.g., temporal difference (TD) (Sutton, 1988) and Q-learning (Watkins & Dayan, 1992). Then the actor updates the policy along an approximate gradient direction based on the estimate from the critic using approaches such as policy gradient (Sutton et al., 1999) and natural policy gradient (Kakade, 2001). In contrast to critic-only methods, AC methods, which are gradient based, usually have desirable convergence properties when combined with the approach of function approximation. However, critic-only methods may not converge or even diverge when applied together with function approximation (Baird, 1995; Gordon, 1996). Moreover, AC methods also enjoy a reduced variance due to the critic, and thus their convergence is typically more stable and faster than actor only methods.

While the asymptotic convergence for AC and NAC has been well understood in the literature, e.g., (Bhatnagar et al., 2009; Kakade, 2001; Konda & Tsitsiklis, 2003), its non-asymptotic convergence analysis has been largely open until very recently. The non-asymptotic analysis is of great practical importance as it answers the questions that how many samples are needed and how to appropriately choose the different learning rates for the actor and the critic. Existing studies show that AC converges to an $\epsilon + \varepsilon_{\text{critic}}$ neighborhood of stationary points with the best known sample complexity of $\mathcal{O}\left(\epsilon^{-2}\right)$ (Chen et al., 2021; Olshevsky & Gharesifard, 2022; Xu et al., 2020a), and NAC converges to an $\epsilon + \varepsilon_{\text{critic}} + \sqrt{\varepsilon_{\text{actor}}}$ neighborhood of the global optimum with the best known sample complexity of $\mathcal{O}(\epsilon^{-3})$ (Chen et al., 2022; Xu et al., 2020a), where $\varepsilon_{\text{critic}}$ is the approximation error of the

critic and $\varepsilon_{\text{actor}}$ is the approximation error induced by the insufficient expressive power of the parameterized policy class. In this paper, when presenting sample complexity, we omit the log factors. In these studies, the critic employs a fixed class of parameterized functions (typically linear function approximation with fixed feature), which may not satisfy the compatible condition (Sutton et al., 1999) (see Section 2 for details). This will result in a non-diminishing bias in the policy gradient estimate, and therefore, an additional error term $\varepsilon_{\text{critic}}$ is incurred in the overall error bound. Several works (Cayci et al., 2022; Wang et al., 2019) propose to use overparameterized neural networks in the critic to mitigate this issue, where $\varepsilon_{\text{critic}}$ diminishes as the network size increases. However, the convergence of the critic requires stringent conditions that are hard to verify (Cayci et al., 2022; Wang et al., 2019), and large neural network introduces expensive computational and memory costs. Actually, if the critic employs the approach of compatible function approximation, which is *linear*, then $\varepsilon_{\text{critic}}$ vanishes without introducing additional computational and memory costs (Sutton et al., 1999) (see details in Section 2). Moreover, for NAC applied with fixed function approximation in the critic, one needs to explicitly estimate the Fisher information matrix and compute its inverse, which will be computationally and memory expensive. Another advantage of compatible function approximation when applied with NAC is that the inverse of the Fisher information in the natural gradient will cancel out with the policy gradient (see Proposition 2), and thus there is no need to estimate the Fisher information matrix anymore.

## 1.1 CHALLENGES AND CONTRIBUTIONS

Though AC and NAC with compatible function approximation enjoy no approximation error from the critic and no need of estimating the Fisher information matrix (for NAC), their non-asymptotic convergence analyses are much more challenging than the ones with fixed function approximation. To the best of the authors' knowledge, this paper develops the tightest non-asymptotic error bounds for AC and NAC algorithms, and our analyses are for the challenging case of single Markovian sample trajectory. We prove that AC with compatible function approximation converges to an $\epsilon$ stationary point with sample complexity $\mathcal{O}(\epsilon^{-2})$, and NAC with compatible function approximation converges to an $\epsilon + \sqrt{\varepsilon_{\text{actor}}}$ neighborhood of the globally optimal policy with sample complexity $\mathcal{O}(\epsilon^{-3})$. Our non-asymptotic error bounds outperform the best known AC and NAC bounds in the literature by a constant $\varepsilon_{\text{critic}}$ and achieve the same sample complexity: $\mathcal{O}(\epsilon^{-2})$ for AC and $\mathcal{O}(\epsilon^{-3})$ for NAC (see Tables 1 and 2). We note that this constant $\varepsilon_{\text{critic}}$ is due to the approximation error of the function class used by the critic, and does not diminish with time.

One of the biggest challenges in the analysis is due to the time-varying critic feature function. Specifically, the critic with compatible function approximation employs an $\omega$-dependent linear function class, where $\omega$ is the policy parameter. As the actor updates the policy, the feature function of the critic also changes with $\omega$. Therefore, the critic is using a linear function with time-varying $\omega$-dependent feature to track the value function of the current policy $\pi_\omega$, which is also time varying. This makes the analysis of the tracking error, i.e., the error between the ideal limit of the critic given the current policy and its current estimate, challenging. In this paper, we design a novel approach to explicitly bound this error. The central idea is to construct an auxiliary eligibility trace with fixed feature to approximate the eligibility trace with time-varying feature (in the critic, we use $k$-step TD with compatible function approximation).

In this paper, we focus on the challenging single-loop setting with a single Markovian sample trajectory. Some studies tried to decouple the updates of the actor and the critic using approaches, e.g., nested loop (Qiu et al., 2021; Agarwal et al., 2021; Chen et al., 2022; Xu et al., 2020a), and to further develop the non-asymptotic analysis. Specifically, after the actor updates the policy, then the policy is fixed and the critic starts an inner loop to iterate sufficient number of steps until it gets a perfect evaluation of the current policy. This decoupling approach makes it easier to analyze as there is no need to analyze the interaction between the actor and the critic. However, this decoupling approach does not enjoy benefits from the two time-scale structure in the original AC and NAC algorithms (Konda & Tsitsiklis, 2003; Bhatnagar et al., 2009), e.g., algorithmic simplicity and statistical efficiency, and techniques therein cannot be generalized to analyze the single-loop single-trajectory two time-scale AC and NAC algorithms. Moreover, analyses therein require some kind of i.i.d. assumptions or require trajectories starting from any arbitrary state, which might be difficult to guarantee in practice. To develop the tightest bound, we develop a novel approach that bounds the tracking error as a function of the policy gradient norm (for AC) and the optimality gap (for NAC). We also note that our analysis for NAC does not need the smoothness assumption on the

Table 1: Comparison of sample complexity of AC

| Reference | Single-loop | Sample size | Error | Comments |
|---|---|---|---|---|
| (Wang et al., 2019) | $\times$ | $\mathcal{O}\left(\epsilon^{-6}\right)$ | $\epsilon + \varepsilon_{\text{critic}}$ | Critic: neural |
| (Zhou & Lu, 2022) | $\surd$ | $\mathcal{O}\left(\epsilon^{-1}\right)$ | $\epsilon$ | LQR |
| (Zhang et al., 2020b) | $\surd$ | Asymptotic | | |
| (Qiu et al., 2021) | $\times$ | $\mathcal{O}\left(\epsilon^{-4}\right)$ | | Actor: |
| (Kumar et al., 2023) | $\times$ | $\mathcal{O}(\epsilon^{-3})$ | | non-linear, smooth |
| (Kumar et al., 2023) (Xu et al., 2020b) | $\times$ | $\mathcal{O}\left(\epsilon^{-2.5}\right)$ | $\epsilon + \varepsilon_{\text{critic}}$ | Critic: linear |
| (Xu et al., 2020a) | $\times$ | $\mathcal{O}\left(\epsilon^{-2}\right)$ | | function approx. |
| (Barakat et al., 2022) (Wu et al., 2020) | $\surd$ | $\mathcal{O}\left(\epsilon^{-2.5}\right)$ | | |
| (Olshevsky & Gharesifard, 2022) (Chen et al., 2021) | $\surd$ | $\mathcal{O}\left(\epsilon^{-2}\right)$ | | |
| Our Work | $\surd$ | $\mathcal{O}(\epsilon^{-2})$ | $\epsilon$ | |

Table 2: Comparison of sample complexity of NAC

| Reference | Single-loop | Sample size | Error | Comments |
|---|---|---|---|---|
| (Khodadadian et al., 2022) | $\surd$ | $\mathcal{O}\left(\epsilon^{-6}\right)$ | | |
| (Khodadadian et al., 2021) | $\times$ | $\mathcal{O}\left(\epsilon^{-3}\right)$ | $\epsilon$ | Tabular case |
| (Wang et al., 2019) | $\times$ | $\mathcal{O}\left(\epsilon^{-6}\right)$ | $\epsilon + \varepsilon_{\text{critic}}$ | Critic: neural |
| (Cayci et al., 2022) | $\times$ | $\mathcal{O}\left(\epsilon^{-3}\right)$ | $+\sqrt{\varepsilon_{\text{actor}}}$ | |
| (Agarwal et al., 2021) | $\times$ | $\mathcal{O}\left(\epsilon^{-6}\right)$ | | Actor: |
| (Xu et al., 2020a) | $\times$ | $\mathcal{O}\left(\epsilon^{-3}\right)$ | $\epsilon + \varepsilon_{\text{critic}}$ | non-linear, smooth |
| (Xu et al., 2020b) | $\times$ | $\mathcal{O}\left(\epsilon^{-4}\right)$ | $+\sqrt{\varepsilon_{\text{actor}}}$ | Critic: linear |
| (Chen et al., 2022) | $\times$ | $\mathcal{O}\left(\epsilon^{-3}\right)$ | | function approx. |
| Our Work | $\surd$ | $\mathcal{O}(\epsilon^{-3})$ | $\epsilon + \sqrt{\varepsilon_{\text{actor}}}$ | |

parameterized policy, which is typically required in existing NAC and AC analyses (Chen et al., 2021; Olshevsky & Gharesifard, 2022).

## 1.2 RELATED WORK

In this section, we review recent relevant works on non-asymptotic analyses on reinforcement learning algorithms with function approximation. We provide a detailed comparison between our results and existing studies on AC and NAC in Tables 1 and 2. The "Sample complexity" in the table is the one needed to guarantee the gradient norm/optimality gap less than or equal to the "Error".

**Actor-critic analyses.** We list recent works on non-asymptotic analyses for AC in Table 1. Based on whether the updates of actor and critic are decoupled, the results can be grouped into "single-loop" and "nested-loop/decoupling" approaches. For a general MDP, the best known sample complexity for both single-loop and nested-loop approaches is $\mathcal{O}(\epsilon^{-2})$ (Chen et al., 2021; Olshevsky & Gharesifard, 2022; Xu et al., 2020a). The only exception is (Zhou & Lu, 2022), which is due to the special structure of the LQR problem. These studies all use a fixed function class in the critic, and therefore, the convergence error consists of a non-diminishing constant term of $\varepsilon_{\text{critic}}$. In this paper, we analyze the AC with compatible function approximation, and we obtain a strictly tighter error bound without $\varepsilon_{\text{critic}}$. Our analysis is also much more challenging than the ones in the literature, which is mainly due to that the function class in the critic varies with the policy in the actor.

**Natural actor-critic analyses.** We list recent works on non-asymptotic analyses for NAC in Table 2. The best sample complexity for single-loop NAC is $\mathcal{O}\left(\epsilon^{-6}\right)$ and it is for the tabular case (Khodadadian et al., 2022), whereas the best sample complexity for nested-loop/decoupling NAC is $\mathcal{O}\left(\epsilon^{-3}\right)$ with an error of $\epsilon + \varepsilon_{\text{critic}} + \sqrt{\varepsilon_{\text{actor}}}$ (Chen et al., 2022; Xu et al., 2020a). There exists a gap of $\mathcal{O}\left(\epsilon^{-3}\right)$ between these two approaches, which is mainly due to the challenge in bounding the tracking error for NAC in the single-loop setting. In this paper, we close this gap and show that NAC

in the single-loop setting can also achieve the sample complexity of $\mathcal{O}(\epsilon^{-3})$, and more importantly with a reduced error of $\epsilon + \sqrt{\varepsilon_{\text{actor}}}$.

**Actor/critic only analyses.** Non-asymptotic analyses for critic only methods have been extensively studied recently, e.g., TD (Srikant & Ying, 2019; Lakshminarayanan & Szepesvari, 2018; Bhandari et al., 2018; Cai et al., 2019; Sun et al., 2019; Xu & Gu, 2020), SARSA (Zou et al., 2019), gradient TD (GTD) method (Dalal et al., 2018; Xu et al., 2019; Wang et al., 2021; 2017; Liu et al., 2015; Gupta et al., 2019; Kaledin et al., 2020; Ma et al., 2020; 2021; Wang & Zou, 2020). There are also non-asymptotic analyses for actor only method, e.g., (Bhandari & Russo, 2021; 2019; Agarwal et al., 2021; Mei et al., 2020; Li et al., 2021a; Laroche & des Combes, 2021; Zhang et al., 2021; Cen et al., 2021; Zhang et al., 2020a; Lin, 2022). In this paper, we focus on AC and NAC algorithms, where how the errors in the actor and the critic affects the other needs to be analyzed.

## 2 PRELIMINARIES

**Markov Decision Processes** Consider a general reinforcement learning setting, where an agent interacts with a stochastic environment modeled as a Markov decision process (MDP). An MDP can be represented by a tuple $\langle \mathcal{S}, \mathcal{A}, P, R \rangle$, where $\mathcal{S}$ denotes the state space, $\mathcal{A}$ denotes the discrete finite action space, $R(\cdot, \cdot) : \mathcal{S} \times \mathcal{A} \to [0, R_{\max}]$ is the reward function. The transition kernel $P(\cdot | s, a)$ denotes the distribution of the next state if taking action $a$ at state $s$, $\forall s \in \mathcal{S}, a \in \mathcal{A}$.

A stationary policy $\pi$ maps a state $s \in \mathcal{S}$ to a probability distribution $\pi(\cdot | s)$ over the action space $\mathcal{A}$. Then the expected long term average reward for a policy $\pi$ is defined as follows: $J(\pi) = \lim_{N \to \infty} \frac{1}{N} \mathbb{E}\left[\sum_{t=0}^{N-1} R(s_t, a_t) | \pi\right] = \mathbb{E}_{s \sim d_\pi, a \sim \pi(\cdot|s)}[R(s, a)]$, where we denote by $d_\pi$ the stationary distribution $d_\pi(s) = \lim_{N \to \infty} \frac{1}{N} \sum_{t=0}^{N-1} \mathbb{P}(s_t = s | \pi)$, denote by $D_\pi = d_\pi \times \pi$ the state-action stationary distribution. We rewrite $R_t := R(s_t, a_t)$. For a given policy $\pi$ and an initial state $s$, the relative value function is defined as $V^\pi(s) = \mathbb{E}\left[\sum_{t=0}^\infty R_t - J(\pi) | s_0 = s, \pi\right], \forall s \in \mathcal{S}$. Given initial state $s$ and action $a$, the relative action value function ($Q$ function) for a given policy $\pi$ is defined as $Q^\pi(s, a) = \mathbb{E}\left[\sum_{t=0}^\infty R_t - J(\pi) | s_0 = s, a_0 = a, \pi\right], \forall (s, a) \in \mathcal{S} \times \mathcal{A}$. The goal is to find the optimal policy $\pi^*$ that maximizes the long term average reward: $\max_\pi J(\pi)$.

**(Natural) Actor-Critic with Compatible Function Approximation** Consider a parameterized policy class $\Pi_\omega = \{\pi_\omega : \omega \in \mathcal{W}\}$, where $\mathcal{W} \subseteq \mathbb{R}^d$. Then the problem in Section 2 can be solved by optimizing over the parameter space $\mathcal{W}$. Specifically, the actor updates the policy via the approach of (natural) policy gradient, where the policy gradient is given by (Sutton et al., 1999)

$$\nabla J(\pi) = \mathbb{E}_{D_{\pi_\omega}}\left[Q^{\pi_\omega}(s, a) \phi_\omega(s, a)\right], \tag{1}$$

where $\phi_\omega(s, a) = \nabla_\omega \log \pi_\omega(a|s)$. We further let $\Phi_\omega$ denote the feature matrix, which is the stack of all feature vectors. Specifically, $\Phi_\omega \in \mathbb{R}^{|\mathcal{S}||\mathcal{A}| \times d}$ and the $(s, a)$-row of $\Phi_\omega$ is $\phi_\omega^\top(s, a)$. On the other hand, the critic estimates the $Q$ function in Equation (1) via the approach of TD learning, and the $Q$ function is usually parameterized using linear function approximation in the existing literature, i.e., $\mathcal{Q} = \{Q_\theta(s, a) = \phi(s, a)^\top \theta, \theta \in \Theta\}$ where $\phi$ denotes the feature vector and $\Theta \subseteq \mathbb{R}^{\bar{d}}$. However, as summarized in Tables 1 and 2, using a fixed $\phi$ introduces an additional non-vanishing error term $\varepsilon_{\text{critic}}$ to the gradient estimate.

To avoid the critic's function approximation error, (Konda & Tsitsiklis, 2003) proposed a smart idea of compatible function approximation, which uses the compatible feature vector $\phi_\omega$ that depends on the policy parameter $\omega$. To explain, in order to approximate the value function $Q^{\pi_\omega}$ associated with policy $\pi_\omega$, we can set the feature vector as $\phi_\omega(s, a) := \nabla_\omega \log \pi_\omega(a|s)$ and solve for the best linear approximation parameter $\bar{\theta}_\omega^*$ via the following optimization problem.

$$\bar{\theta}_\omega^* \in \arg\min_\theta \mathbb{E}_{D_{\pi_\omega}}\left[\left(Q^{\pi_\omega}(s, a) - \phi_\omega^\top(s, a)\theta\right)^2\right]. \tag{2}$$

**Proposition 1** (Konda & Tsitsiklis (2003)). *With compatible function approximation, the policy gradient $\nabla J(\pi_\omega)$ can be rewritten as:*

$$\nabla J(\pi_\omega) = \mathbb{E}_{D_{\pi_\omega}}\left[\nabla \log \pi_\omega(a|s) Q^{\pi_\omega}(s, a)\right] = \mathbb{E}_{D_{\pi_\omega}}\left[\phi_\omega(s, a)(\phi_\omega^\top(s, a)\bar{\theta}_\omega^*)\right]. \tag{3}$$

This implies that as long as we can solve the finite dimensional problem Equation (2), linear function approximation with the compatible feature $\phi_\omega$ and parameter $\bar{\theta}_\omega^*$ does not induce any

---

**Algorithm 1** (Natural) Actor-Critic with Compatible Function Approximation

---

1: **Initialization:** $k, \eta_0, \omega_0, \pi_0 = \pi_{\omega_0}, \theta_0, \phi_0 = \nabla \log \pi_0, s_0, a_0 \sim \pi_0(\cdot|s_0), z_0 = 0$
2: **for** $t = 0, ..., T - 1$ **do**
3:     Observe $R_t$
4:     $s_{t+1} \sim P(\cdot|s_t, a_t); a_{t+1} \sim \pi_t(\cdot|s_{t+1})$
5:     $\phi_t(s, a) = \nabla_\omega \log \pi_t(a|s)$       /*Compatible function approximation*/
6:     **Critic**: $\delta_t(\theta_t) = R_t - \eta_t + \phi_t^\top(s_{t+1}, a_{t+1})\theta_t - \phi_t^\top(s_t, a_t)\theta_t$     /*TD error*/
7:         $z_t = \sum_{j=t-k}^t \phi_j(s_j, a_j)$         /*eligibility trace*/
8:         $\eta_{t+1} = \eta_t + \gamma_t(R_t - \eta_t)$         /*average reward update*/
9:         $\theta_{t+1} = \Pi_{2,B}(\theta_t + \alpha_t \delta_t(\theta_t) z_t)$         /*TD update*/
10:    **Option I**: $\omega_{t+1} = \omega_t + \beta_t \phi_t^\top(s_t, a_t)\theta_t \phi_t(s_t, a_t)$    /*Actor update in AC*/
11:    **Option II**: $\omega_{t+1} = \omega_t + \beta_t \theta_t$         /*Actor update in NAC*/
12: **end for**

---

function approximation error. This approach is referred to as compatible function approximation (Konda & Tsitsiklis, 2003), i.e., estimating $Q^{\pi_\omega}$ using an $\omega$-dependent linear function class: $\mathcal{Q}_\omega = \{\phi_\omega^\top(s, a)\theta, \theta \in \Theta\}$. To solve Equation (2) for the compatible function approximation parameter, we use the $k$-step TD algorithm with compatible feature $\phi_\omega$ (Konda & Tsitsiklis, 2003).

The actor can also use the following natural policy gradient to update the policy (Kakade, 2001). $\widetilde{\nabla} J(\pi_\omega) = F_\omega^{-1} \nabla J(\pi_\omega)$, where the matrix $F_\omega$ denotes the Fisher information matrix: $F_\omega = \mathbb{E}_{D_{\pi_\omega}} \left[ \nabla \log \pi_\omega(a|s) \left( \nabla \log \pi_\omega(a|s) \right)^\top \right]$.

**Proposition 2** (Peters & Schaal (2008)). *With compatible function approximation, natural policy gradient is reduced to:* $\widetilde{\nabla} J(\pi_\omega) = \bar{\theta}_\omega^*$.

That is, there is no need to estimate the Fisher information matrix and compute its inverse, which is typically computationally expensive.

## 3 MAIN RESULTS

The detailed AC and NAC algorithms with compatible function approximation is summarized in Algorithm 1. In the critic update, $\alpha_t$ is the stepsize, and denote by $\Pi_{2,B}(v) = \arg\min_{\|\omega\|_2 \leq B} \|v - \omega\|_2$ for any $v \in \mathbb{R}^d$ the project operator, and $B$ is the radius. Next, we present the non-asymptotic bounds for the AC and NAC with compatible function approximation in Algorithm 1.

**Assumption 1.** *(Uniform Ergodicity) Consider the MDP with policy $\pi_\omega$ and transition kernel $P$, there exists constants $m > 0$, and $\rho \in (0, 1)$ such that $\sup_{s \in \mathcal{S}} \|\mathbb{P}(s_t \in \cdot|s_0 = s) - D_{\pi_\omega}(\cdot)\|_{\mathcal{TV}} \leq m\rho^t$.*

Here $\|\cdot\|_{\mathcal{TV}}$ denotes the total variation distance between two distributions. Assumption 1 is widely used in the literature to handle the Markovian noise, e.g., (Srikant & Ying, 2019; Zou et al., 2019; Bhandari et al., 2018). We further assume that the $d$ feature functions, $\phi_{\omega,i}, i = 1, ..., d$, are linearly independent, i.e., the feature matrix $\Phi_\omega$ is full rank when $|\mathcal{S}||\mathcal{A}| \geq d$. This is also commonly used in the literature of analyzing RL algorithms with linear function approximation (Srikant & Ying, 2019; Zou et al., 2019; Bhandari et al., 2018).

### 3.1 CRITIC:$k$-STEP TD

Consider the critic update, where the TD method is used to learn the relative value function under the average-reward setting. It is known that the feature function needs to satisfy certain condition (Assumption 2 in (Tsitsiklis & Van Roy, 1999)) so that the limit of the TD method is unique. In the following proposition, we show that compatible function approximation automatically satisfy the assumption needed in (Tsitsiklis & Van Roy, 1999), and therefore guarantees the convergence of the critic without the need of any additional assumptions.

**Proposition 3.** *For any $\omega \in \mathcal{W}$ and $\theta \in \Theta$, $\Phi_\omega \theta \neq \mathbf{e}$, where $\mathbf{e} \in \mathbb{R}^d$ is an all-one vector.*

We note that the results in (Wu et al., 2020) use a different assumption from the one in (Tsitsiklis & Van Roy, 1999) to guarantee the convergence of the critic in the average-reward setting (Assumption

---

**Algorithm 2** Compatible $k$-step TD Algorithm

---

1: **Initialization:** $k, \eta, \theta_0, \phi = \nabla \log \pi_\omega, s_0, a_0 \sim \pi_\omega(\cdot|s_0), z_0 = 0$
2: **for** $t = 0, ..., T - 1$ **do**
3:      Observe $R_t$
4:      $s_{t+1} \sim P(\cdot|s_t, a_t); a_{t+1} \sim \pi_\omega(\cdot|s_{t+1})$
5:      $\delta(\theta_t) = R_t - \eta + \phi^\top(s_{t+1}, a_{t+1})\theta_t - \phi^\top(s_t, a_t)\theta_t$           /*TD error*/
6:      $z_t = \sum_{j=t-k}^t \phi(s_j, a_j)$           /*eligibility trace*/
7:      $\eta = \eta + \gamma_t(R_t - \eta)$           /*average reward update*/
8:      $\theta_{t+1} = \Pi_{2,B}(\theta_t + \alpha_t \delta(\theta_t)z_t)$           /*TD update*/
9: **end for**

---

4.1 in (Wu et al., 2020)): the matrix $\mathbb{E}[\phi(s)(\phi(s') - \phi(s))^\top]$ is negative definite, where $\phi$ is the fixed feature function, $s$ is the current state and $s'$ is the subsequent state.

As discussed in Section 2, we would like the critic to find the solution of Equation (2). However, the objective in Equation (2) requires the knowledge of $Q^{\pi_\omega}$, which is unavailable. Therefore, in the critic, we propose to use the method of $k$-step TD, so that as $k$ enlarges, the solution from the $k$-step TD converges to the solution of Equation (2). We present the $k$-step TD algorithm in Algorithm 2. Here, the AC and NAC algorithms in Algorithm 1 are single-loop, single sample trajectory and two time-scale. We introduce the $k$-step TD algorithm in Algorithm 2 only to illustrate the basic idea.

Based on Proposition 3, Assumption 1, and the assumption that $\Phi_\omega$ is full rank, from (Tsitsiklis & Van Roy, 1999, Theorem 1), we can show that the $k$-step TD algorithm in Algorithm 2 has a unique solution, denoted by $\theta_\omega^*$:

$$\mathbb{E}_{D_{\pi_\omega}} \left[ \phi_\omega^\top(s, a) \left( \mathcal{T}_{\pi_\omega}^{(k)} \phi_\omega^\top(s, a)\theta_\omega^* - \phi_\omega^\top(s, a)\theta_\omega^* \right) \right] = \mathbf{0}, \tag{4}$$

where $\mathcal{T}_{\pi_\omega}^{(k)}(Q(s, a)) = \mathbb{E}[\sum_{j=0}^{k-1}(R_j - J(\pi_\omega)) + Q(s_k, a_k)|s_0 = s, a_0 = a, \pi_\omega]$.

Assume that $\mathbb{E}_{D_w}\left[\phi_w(s, a)\phi_w^\top(s, a)\right]$ is positive definite with the minimum eigenvalue $\lambda_{\min} > 0$. This is to guarantee that the solution to Equation (2) is unique. We can remove this assumption by adding a regularizer $\lambda\|\theta\|_2^2$ to Equation (2) to guarantee the solution to the regularized Equation (2) is unique, and bounding the difference.

Then we bound the difference between the solution to Equation (2) and the solution to the $k$-step TD algorithm in the following proposition.

**Proposition 4.** *For any $\omega \in \mathcal{W}$, the difference between $\theta_\omega^*$ and $\bar{\theta}_\omega^*$ can be bounded as follows:* $\left\|\theta_\omega^* - \bar{\theta}_\omega^*\right\|_2 \leq \frac{C_{gap} m \rho^k}{\lambda_{\min}}$*, where $C_{gap} = C_\phi^2 B + C_\phi R_{\max}\frac{1}{1-\rho}$.*

It can be seen that the bound diminishes exponentially with $k$. Therefore by picking a large $k$, the $k$-step TD is expected to solve Equation (2) to a desired accuracy.

### 3.2 Non-asymptotic Bound for AC

**Assumption 2.** *(Smoothness and Boundedness) For any $\omega, \omega' \in \mathbb{R}^d$ and any state-action pair $(s, a) \in \mathcal{S} \times \mathcal{A}$, there exist positive constants $L_\phi, C_\phi, C_\pi$ and $L_\delta$ such that*

*1)* $\|\phi_\omega(s, a) - \phi_{\omega'}(s, a)\|_2 \leq L_\phi \|\omega - \omega'\|_2$;      *2)* $\|\phi_\omega(s, a)\|_2 \leq C_\phi$;
*3)* $\|\pi_\omega(\cdot|s) - \pi_{\omega'}(\cdot|s)\|_{\mathcal{TV}} \leq C_\pi \|\omega - \omega'\|_2$;      *4)* $\left\|\nabla^2\pi_\omega(\cdot|s) - \nabla^2\pi_{\omega'}(\cdot|s)\right\|_2 \leq L_\delta \|\omega - \omega'\|_2$.

The first three assumptions in Assumption 2 assume the policy and feature function $\phi_\omega$ is smooth and bounded. The fourth one in Assumption 2 is only needed for the AC analysis. For the NAC analysis, it is not necessary. We note that these assumptions can be easily satisfied by choosing a proper policy parameterization. For example, if the policy is parameterized using neural network, then these assumptions can be satisfied if the activation function is Lipschitz and smooth (Du et al., 2019; Miyato et al., 2018; Neyshabur, 2017).

We first present the bound on the tracking error, which measures how the critic tracks its ideal limit: $\frac{1}{T} \sum_{t=0}^{T-1} \mathbb{E}\left[\|\theta_t^* - \theta_t\|_2^2\right]$. Here, $\theta_t$ is the critic parameter at time $t$ of Algorithm 1, and we rewrite $\theta_t^* = \theta_{\omega_t}^*$ and $J(\omega_t) = J(\pi_{\omega_t})$ for convenience. In the AC algorithm, we set $\alpha_t = \alpha$, $\beta_t = \beta$, $\gamma_t = \gamma$, and $k = \mathcal{O}(\log T)$ such that $\gamma \geq \alpha \geq \beta \geq m\rho^k$. Note that we use a projection in Line 8 in Algorithm 1. In order for convergence and optimality, we require that all $\|\theta_\omega^*\| \leq B$. A sufficient condition to guarantee this is to set $B = \frac{mR_{\max}C_\phi}{(1-\rho)\left(\lambda_{\min} - C_\phi^2 dm\rho^k\right)}$ (see Appendix A for the proof).

**Proposition 5.** *The tracking error of the AC algorithm in Algorithm 1 can be bounded as follows:*

$$\frac{1}{T} \sum_{t=0}^{T-1} \mathbb{E}\left[\|\theta_t^* - \theta_t\|_2^2\right] \leq \left(\frac{c_\alpha \beta}{\alpha} + \frac{c_\eta \beta}{\gamma}\right) \frac{1}{T} \sum_{t=0}^{T-1} \mathbb{E}\left[\|\nabla J(\omega_t)\|_2^2\right] + \mathcal{O}\left(\frac{1}{T\alpha}\right) + \mathcal{O}\left(\frac{\log^2 T}{T\gamma}\right)$$

$$+ \mathcal{O}\left(\alpha \log^2 T\right) + \mathcal{O}\left(\beta \log^3 T\right) + \mathcal{O}\left(\gamma \log^3 T\right) + \mathcal{O}\left(\beta^2 \log^2 T\alpha^{-1}\right) + \mathcal{O}\left(\beta^2 \gamma^{-1}\right),$$

*where $c_\alpha$ and $c_\eta$ is a positive constant defined in Appendix B. Set $\gamma = \mathcal{O}(\frac{1}{\sqrt{T}})$, $\alpha, \beta = \mathcal{O}(\frac{1}{\sqrt{T}\log^2 T})$, we have $\frac{1}{T} \sum_{t=0}^{T-1} \mathbb{E}\left[\|\theta_t^* - \theta_t\|_2^2\right] \leq \frac{1}{4C_\phi^4} \frac{1}{T} \sum_{t=0}^{T-1} \mathbb{E}\left[\|\nabla J(\omega_t)\|_2^2\right] + \mathcal{O}\left(\frac{\log^3 T}{\sqrt{T}}\right)$.*

For simplicity, we only present the order of the bound here, and the detailed non-asymptotic bound can be found in the Appendix B. The key novelty in the analysis is that we bound the tracking error as a function of the policy gradient, and we also bound the policy gradient as a function of the tracking error. By applying the bound recursively, we get a tight bound on the tracking error in Proposition 5. Many existing studies in the two time-scale analysis upper bound the policy gradient in the tracking error using its maximum norm, which is constant-level. However, as we see in the following theorem, the policy gradient shall also decrease to zero. Therefore, the above approach does not obtain the tightest bound, and leads to a higher-order sample complexity.

**Theorem 1.** *Consider the AC algorithm in Algorithm 1. It can be shown that*

$$\frac{1}{T} \sum_{t=0}^{T-1} \mathbb{E}\left[\|\nabla J(\omega_t)\|_2^2\right] \leq C_\phi^4 \frac{1}{T} \sum_{t=0}^{T-1} \mathbb{E}\left[\|\theta_t^* - \theta_t\|_2^2\right] + \mathcal{O}\left(\frac{1}{T\beta}\right) + \mathcal{O}\left(\beta \log^2 T\right). \quad (5)$$

*Set $\gamma = \mathcal{O}(\frac{1}{\sqrt{T}})$, $\alpha, \beta = \mathcal{O}(\frac{1}{\sqrt{T}\log^2 T})$, then $\frac{1}{T} \sum_{t=0}^{T-1} \mathbb{E}\left[\|\nabla J(\omega_t)\|_2^2\right] \leq \mathcal{O}\left(\frac{\log^3 T}{\sqrt{T}}\right)$.*

Theorem 1 implies that the AC algorithm with compatible function approximation converges to an $\epsilon$-stationary point with sample complexity $\epsilon^{-2}$. This improves the best known error bound by a constant $\varepsilon_{\text{critic}}$ (Wang et al., 2019; Zhang et al., 2020b; Qiu et al., 2021; Kumar et al., 2023; Xu et al., 2020b; Barakat et al., 2022; Wu et al., 2020; Chen et al., 2021; Olshevsky & Gharesifard, 2022; Xu et al., 2020a), and matches the best known sample complexity (Chen et al., 2021; Olshevsky & Gharesifard, 2022; Xu et al., 2020a).

### 3.3 NON-ASYMPTOTIC BOUND FOR NAC

In this section, we present the non-asymptotic bound for the NAC algorithm in Algorithm 1. It was shown in (Agarwal et al., 2021) that due to the parameter invariant property of the natural policy gradient update, natural policy gradient is able to converge to the globally optimal policy with a gap that depends on the capacity of the policy class. Define the compatible linear function approximation error $\varepsilon_{\text{actor}} = \max_{\omega \in \mathcal{W}} \left\{\min_\theta \mathbb{E}_{D_{\pi_\omega}}\left[\|Q^{\pi_\omega}(s,a) - \phi_\omega^\top(s,a)\theta\|_2^2\right]\right\}$. This error represents the approximation error due to the insufficient expressive power of the policy parameterization, and shall decrease if a large neural network is used.

Using the same idea as the one in AC, we can also develop a tight bound on the tracking error: $\mathcal{O}\left(T^{-\frac{1}{3}}\right)$, where now we bound the tracking error as a function of the optimality gap instead of the gradient norm. We then also develop bound of the optimality gap as a function of the tracking error. Applying them recursively, we obtain the tightest bound on the tracking error and the tightest bound on the optimality gap in the following theorem. We set $\alpha_t = \alpha$, $\beta_t = \beta$, $\gamma_t = \gamma$, and $k = \mathcal{O}(\log T)$ such that $\gamma \geq \alpha \geq \beta \geq m\rho^k$.

**Assumption 3.** *There exist a constant $C_\infty < \infty$ such that $\sup_{\omega \in \mathcal{W}} \left\| \frac{D_{\pi^*}(s,a)}{D_{\pi_\omega}(s,a)} \right\|_\infty \leq C_\infty$.*

Assumption 3 guarantees that the policy is sufficiently exploratory, and is commonly used in NAC analyses, e.g., (Cayci et al., 2022; Xu et al., 2020a; Agarwal et al., 2021). Approaches to guarantee this assumption were also studied in (Khodadadian et al., 2021; 2022).

**Theorem 2.** *Consider the NAC algorithm in Algorithm 1. Then, we have that*

$$\min_{t<T} \mathbb{E}\left[J(\pi^*) - J(\omega_t)\right] \leq \mathcal{O}\left(\frac{\log^2 T}{T\alpha}\right) + \mathcal{O}\left(\frac{\log T}{T\beta}\right) + \mathcal{O}\left(\frac{\log^2 T}{T\sqrt{\alpha\beta}}\right) + \mathcal{O}\left(\frac{\sqrt{\gamma\beta}\log^2 T}{\sqrt{\alpha}}\right)$$

$$+ \mathcal{O}\left(\frac{\gamma\log^2 T}{\sqrt{\alpha}}\right) + \mathcal{O}\left(\frac{\beta\log^2 T}{\sqrt{\alpha}}\right) + \mathcal{O}\left(\sqrt{\alpha}\log^2 T\right) + \mathcal{O}\left(\sqrt{\beta}\log^2 T\right) + \mathcal{O}\left(\sqrt{\varepsilon_{actor}}\right).$$

*If we set $\gamma = \mathcal{O}(T^{-\frac{2}{3}}), \alpha = \mathcal{O}(T^{-\frac{2}{3}}\log^{-2}T), \beta = \mathcal{O}(T^{-\frac{2}{3}}\log^{-2}T)$, we have*

$$\min_{t<T} \mathbb{E}[J(\pi^*) - J(\omega_t)] \leq \mathcal{O}\left(T^{-\frac{1}{3}}\log^4 T\right) + \mathcal{O}\left(\sqrt{\varepsilon_{actor}}\right). \tag{6}$$

**Remark 1.** *Unlike the results for AC in Theorem 1, Theorem 2 for NAC only needs the first three assumptions in Assumption 2. This is one advantage of using compatible function approximation in NAC. As we can see from Line 11 in Algorithm 1 and Proposition 2, the inverse of the Fisher information matrix is cancelled out. Therefore, there is no stochastic noise from using $\phi_\omega(s_t, a_t)\phi_\omega^\top(s_t, a_t)$ in the analysis of NAC. However, in AC, we need to handle this noise, and therefore, the fourth assumption in Assumption 2 is needed for the AC algorithm.*

Theorem 2 implies that NAC with compatible function approximation converges to an $\epsilon + \sqrt{\varepsilon_{actor}}$-neighborhood of the globally optimal policy $\pi^*$ with sample complexity $\mathcal{O}(\epsilon^{-3})$. Compared to existing studies, our work eliminate the approximation error of the critic, $\varepsilon_{critic}$, from the overall error bound (Wang et al., 2019; Cayci et al., 2022; Agarwal et al., 2021; Xu et al., 2020a;b; Chen et al., 2022). Moreover, as summarized in Table 2, the best known sample complexity of NAC is $\epsilon^{-3}$, which however is for the nested-loop NAC variant (Xu et al., 2020a; Chen et al., 2022). Our results achieves this sample complexity, and is for the challenging single-loop NAC algorithm with a single Markovian sample trajectory.

Here we provide a proof sketch for the NAC algorithm to highlight major challenges and our technical novelties. The analysis of NAC contains of most major technical novelty in the AC analysis.

*Proof sketch.* For simplicity of presentation, we set $\hat{t} = \lceil \frac{3\log T}{\lambda_{\min}\alpha} \rceil$ and $\widetilde{T} = \hat{t}\lceil \frac{T}{\hat{t}\log T} \rceil$. We denote by $M_t = \mathbb{E}[||\theta_t - \theta_t^*||_2^2] + \mathbb{E}[(\eta_t - J(\omega_t))^2]$ the sum of the tracking error and the estimation error of the average reward. Denote by $D(\omega_t) = KL(\pi^*|\pi_t)$ the KL divergence between policy $\pi^*$ and $\pi_t$.

**Step 1 (Error decomposition):** According to the smoothness property of $D(\omega)$ with respect to $\omega$, we bound the performance gap between the current policy and the optimal policy (optimality gap) as follows: $\frac{1}{\widetilde{T}} \sum_{j=t}^{t+\widetilde{T}-1} \mathbb{E}[J(\pi^*) - J(\omega_j)] \leq \frac{D(\omega_{t+\widetilde{T}}) - D(\omega_t)}{\widetilde{T}\beta} + \mathcal{O}\left(\sqrt{\frac{1}{\widetilde{T}}\sum_{j=t}^{t+\widetilde{T}-1} M_j}\right) + \mathcal{O}(C_\infty\sqrt{\varepsilon_{actor}} + \beta + m\rho^k)$.

**Step 2 (Estimation error in the average reward):** In this step, we analyze estimation error in the average reward: $\eta_t - J(\omega_t)$. We provide a tight characterization of this error:

$$\mathbb{E}[(\eta_{t+1} - J(\omega_{t+1}))^2] \leq (1-\gamma)\mathbb{E}[(\eta_t - J(\omega_t))^2] + \mathcal{O}(\beta\mathbb{E}[||\nabla J(\omega_t)||_2^2])$$
$$+ \mathcal{O}(m\rho^k\gamma + k^2\gamma^2 + k^2\beta^2). \tag{7}$$

One of our key novelties lies in that we bound this estimation error using the gradient norm $\mathbb{E}[||\nabla J(\omega_t)||_2^2]$. The above bound itself is tighter than the existing one in (Wu et al., 2020).

**Step 3 (Tracking error):** In this step, we bound the tracking error in the critic: $||\theta_t - \theta_t^*||_2^2$. By the TD error step in Algorithm 1, we decompose the term $||\theta_{t+1} - \theta_{t+1}^*||_2^2$ as follows: $||\theta_{t+1} - \theta_{t+1}^*||_2^2 \leq ||\theta_t - \theta_t^*||_2^2 + \alpha^2||\delta_t z_t||_2^2 + ||\theta_t^* - \theta_{t+1}^*||_2^2 + 2\alpha\langle\theta_t - \theta_t^*, \delta_t z_t\rangle + 2\alpha\langle\delta_t z_t, \theta_t^* - \theta_{t+1}^*\rangle + 2\langle\theta_t - \theta_t^*, \theta_t^* - \theta_{t+1}^*\rangle$.

Another key challenge lies in how to bound the term $\mathbb{E}[\langle\theta_t - \theta_t^*, \delta_t z_t\rangle]$. We develop a novel technique of auxiliary Markov chain to decompose this error into two parts: 1) error due to time-varying

feature function and 2) error due to time-varying policy. Specifically, consider the first Markov chain generated from the algorithm:

$$s_0, a_0 \stackrel{\pi_0 \times P}{\to} s_1, a_1 \to ... \to s_t, a_t \stackrel{\pi_t \times P}{\to} s_{t+1}, a_{t+1},$$

where at each time $j$, the action is chosen according to $\pi_j$ and the transition kernel is $P$. Here $z_t = \sum_{j=t-k}^{t} \phi_j(s_j, a_j)$ is the eligibility trace used in the algorithm. It can be seen that in $z_t$, the feature $\phi_j$ changes with $j$, and the distribution of $s_j, a_j$ depends on the time-varying policy $\pi_j$. We then design an auxiliary eligibility trace $\hat{z}_t = \sum_{j=t-k}^{t} \phi_t(s_j, a_j)$, where the feature is fixed to be $\phi_t$, and only the the distribution of $s_j, a_j$ depends on the time-varying policy $\pi_j$. To further handle the time-varying distribution of $s_j, a_j$, we design an auxiliary Markov chain (denoted by $A1$) as follows:

$$A1 : (s_0, \widetilde{a}_0) \sim \pi_t \stackrel{\pi_t \times P}{\to} \widetilde{s}_1, \widetilde{a}_1 \stackrel{\pi_t \times P}{\to} ... \stackrel{\pi_t \times P}{\to} \widetilde{s}_t, \widetilde{a}_t \stackrel{\pi_t \times P}{\to} \widetilde{s}_{t+1}, \widetilde{a}_{t+1},$$

where the action at each time $j$ is always chosen according to a fixed policy $\pi_t$. Based on this auxiliary Markov chain, we introduce another auxiliary eligibility trace $\widetilde{z}_t = \sum_{j=t-k}^{t} \phi_t(\widetilde{s}_j, \widetilde{a}_j)$, where it uses a fixed feature $\phi_t$, and samples from this auxiliary Markov chain. Lastly, we design a second auxiliary Markov chain (denoted by $A2$):

$$A2 : (\bar{s}_0, \bar{a}_0) \sim D_t \stackrel{\pi_t \times P}{\to} \bar{s}_1, \bar{a}_1 \stackrel{\pi_t \times P}{\to} ... \stackrel{\pi_t \times P}{\to} \bar{s}_t, \bar{a}_t \stackrel{\pi_t \times P}{\to} \bar{s}_{t+1}, \bar{a}_{t+1}$$

where the only difference between A2 and A1 lies in the initial state distribution. Then we define the last auxiliary eligibility trace as $\bar{z}_t = \sum_{j=t-k}^{t} \phi_t(\bar{s}_j, \bar{a}_j)$.

The difference between $z_t$ and $\hat{z}_t$ measures the error due to the time-varying compatible feature function. We bound this error using the Lipschitz continuity of the feature function. The difference between $\hat{z}_t$ and $\widetilde{z}_t$ measures the error due to the time-varying sampling policy. The difference between $\widetilde{z}_t$ and $\bar{z}_t$ measures the error due to the difference between the stationary distribution and the actual distribution of the samples, which can be bounded based on Assumption 1. By such a error decomposition, we can show that

$$\mathbb{E}[||\theta_{t+1} - \theta_{t+1}^*||_2^2] \leq (1 - \bar{\lambda}_{\min}\alpha/2)\mathbb{E}[||\theta_t - \theta_t^*||_2^2] + \mathcal{O}(k^2\alpha\mathbb{E}[(\eta_t - J(\omega_t))^2])$$
$$+ \mathcal{O}(\beta\mathbb{E}[||\nabla J(\omega_t)||_2^2]) + \mathcal{O}(k^2\alpha^2 + k^3\beta^2 + m\rho^k\alpha). \tag{8}$$

**Step 4 (Bound on gradient):** As we can see from Steps 2 and 3, we bound the estimation error of the average reward and the tracking error using the gradient norm $||\nabla J(\omega_t)||_2^2$. Therefore, in order to derive the tightest bound, we further develop a novel bound on the gradient norm $||\nabla J(\omega_t)||_2^2$. Note that the idea is novel as it serves as a pivotal link connecting the analysis of the tracking error/estimation error in the average reward and the optimality gap. Specifically, we bound the gradient norm using the estimation error in the average reward and tracking error. By the smoothness of $J(\omega)$, we have that

$$\sum_{j=t}^{t+\widetilde{T}-1} \frac{\mathbb{E}[||\nabla J(\omega_j)||_2^2]}{\widetilde{T}} \leq 2C_\phi \frac{J(\omega^*) - E[J(\omega_t)]}{\beta\widetilde{T}} + \mathcal{O}\left(\frac{1}{\widetilde{T}} \sum_{j=t}^{t+\widetilde{T}-1} \mathbb{E}[||\theta_j - \theta_j^*||_2^2]\right) + \mathcal{O}(m\rho^k + \beta).$$

We also note that we bound the gradient norm using the optimality gap, and this is of great importance to establish the tight bound in this paper. In previous works, this term $\mathbb{E}[J(\omega_{t+\widetilde{T}})] - E[J(\omega_t)]$ is bounded by a constant, and thus the overall complexity is not as tight.

**Step 5:** Combining steps 1-4, we conclude the proof. $\square$

## 4 CONCLUSION

In this paper, we develop the tightest non-asymptotic convergence bounds for both the AC and NAC algorithms with compatible function approximation. For the AC algorithm, our results achieve the best sample complexity of $\epsilon^{-2}$ with a reduced error from $\epsilon + \varepsilon_{\text{critic}}$ to $\epsilon$, where $\varepsilon_{\text{critic}}$ is a non-diminishing constant. For the NAC algorithm, our results is the first one in the literature that analyze the single-loop NAC with a single Markovian trajectory, and we achieve the best known sample complexity of $\epsilon^{-3}$ also with a reduced error of $\epsilon + \sqrt{\varepsilon_{\text{actor}}}$. Our results demonstrate the advantage of compatible function approximation when applied in AC and NAC algorithms, including relaxed technical condition to guarantee convergence, no need of estimating Fisher information matrix, and no approximation error from the critic. Our technical novelty lies in analyzing the error due to use of a time-varying and policy dependent feature in the critic.

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
