# Appendix

## Table of Contents

## A  SUPPORTING LEMMAS AND PROOFS FOR PROPOSITIONS 3 AND 4

In this section, we provide a number of supporting lemmas, and proofs for Proposition 3 and Proposition 4. In the following proofs, $\|a\|_2$ denotes the $\ell_2$ norm if $a$ is a vector; and $\|A\|_2$ denotes the operator norm if $A$ is a matrix.

### A.1  SUPPORTING LEMMAS

For convenience, we denote $J(\omega) = J(\pi_\omega)$. We first prove a lemma showing that both $J(\omega)$ and $\nabla J(\omega)$ are Lipschitz in $\omega$.

**Lemma 1.** *Under Assumptions 1 and 2, for any $\omega, \omega' \in \mathcal{W}$, we have that*

$$\|\nabla J(\omega)\|_2 \leq C_J, \tag{9}$$

*where $C_J = C_\phi^2 \left( B + \frac{C_{gap} m \rho^k}{\lambda_{\min}} \right)$, and*

$$\|\nabla J(\omega) - \nabla J(\omega')\|_2 \leq L_J \|\omega - \omega'\|_2, \tag{10}$$

*where $L_J = \frac{m R_{\max}}{1-\rho} (4 L_\pi C_\phi + L_\phi)$ and $L_\pi = \frac{1}{2} C_\pi \left( 1 + \lceil \log m^{-1} \rceil + \frac{1}{1-\rho} \right)$.*

Recall Equation (2). The solution $\bar{\theta}_\omega^*$ given the feature function satisfies that

$$\bar{\theta}_\omega^* = \arg\min_\theta \mathbb{E}_{D_\omega} \left[ \left\| Q^{\pi_\omega}(s,a) - \phi_\omega^\top(s,a)\theta \right\|_2^2 \right]. \tag{11}$$

We show that the solution $\bar{\theta}_\omega^*$ is (nearly) Lipschitz in $\omega$ in the following lemma.

**Lemma 2.** *For any* $\omega, \omega' \in \mathcal{W}$*, it holds that*

$$\left\| \bar{\theta}_\omega^* - \bar{\theta}_{\omega'}^* \right\|_2 \leq C_\Theta \left\| \omega - \omega' \right\|_2, \tag{12}$$

*where* $C_\Theta = \frac{C_J}{\lambda_{\min}^2} \left( 2 C_\phi L_\phi + C_\phi^2 L_\pi \right) + \frac{L_J}{\lambda_{\min}}$.

For any $\omega \in \mathbb{R}^d$, let

$$A_\omega(s, a) = \mathbb{E} \left[ \phi_\omega(s_0, a_0) \left( \phi_\omega(s_k, a_k) - \phi_\omega(s_0, a_0) \right)^\top | s_0 = s, a_0 = a, \pi_\omega \right],$$
$$A_\omega = \mathbb{E}_{D_{\pi_\omega}} \left[ A_\omega(s, a) \right]. \tag{13}$$

**Lemma 3.** *For* $k > \left\lceil \frac{\log(m d C_\phi^2) - \log \lambda_{\min}}{1 - \rho} \right\rceil$*, it holds that*

$$\lambda_{\max} \left( \frac{A_\omega + A_\omega^\top}{2} \right) \leq C_\phi^2 d m \rho^k - \lambda_{\min} = -\bar{\lambda}_{\min} < 0,$$

*where* $\lambda_{\max}(X)$ *is the largest eigenvalue of symmetric matrix* $X$.

When $k > \left\lceil \frac{\log(m d C_\phi^2) - \log \lambda_{\min}}{1 - \rho} \right\rceil$,

$$\begin{aligned}
\bar{\lambda}_{\min} &= \lambda_{\min} - C_\phi^2 d m \rho^k \\
&> \lambda_{\min} - C_\phi^2 d m e^{-k(1-\rho)} \\
&\geq \lambda_{\min} - C_\phi^2 d m e^{-\log\left( \frac{m d C_\phi^2}{\lambda_{\min}} \right)} = 0, \tag{14}
\end{aligned}$$

$\bar{\lambda}_{\min}$ is positive.

The following lemma bounds the distance between the stationary distribution induced by $\pi_t$ and the distribution of $s_t, a_t$ in Algorithm 1. Define $\mathcal{F}_j$ to be $\sigma$-field generated by all the randomness until the $j$-th time-step. For simplicity, we write $D_{\pi_t}$ as $D_t$.

**Lemma 4.** *For any* $0 \leq k \leq t$*, it can be shown that*

$$\left\| \mathbb{P}\left( s_t, a_t | \mathcal{F}_{t-k} \right) - D_t \right\|_{\mathcal{TV}} \leq C_\pi \sum_{j=t-k}^{t-1} \left\| \omega_t - \omega_j \right\|_2 + m \rho^k. \tag{15}$$

We rewrite $\theta_t^* = \theta_{\omega_t}^*$, where $\theta_\omega^*$ is the solution to Equation (4).

**Lemma 5.** *Consider the term* $\mathbb{E}\left[ \langle \theta_t - \theta_t^*, \delta_t z_t \rangle \right]$*. It can be shown that*

$$\begin{aligned}
\mathbb{E}\left[ \langle \theta_t - \theta_t^*, \delta_t z_t \rangle \right] \leq &-\frac{\bar{\lambda}_{\min}}{2} \mathbb{E}\left[ \| \theta_t - \theta_t^* \|_2^2 \right] \\
&+ \frac{(k+1)^2 C_\phi^2}{2 \bar{\lambda}_{\min}} \mathbb{E}\left[ \| J(\omega_t) - \eta_t \|_2^2 \right] + G_t^\delta, \tag{16}
\end{aligned}$$

*where* $U_\delta = R_{\max} + C_\phi B$*. For AC,*

$$\begin{aligned}
G_t^\delta = &2 B^2 C_\phi^2 U_\delta C_\pi \sum_{j=t-k}^{t} \sum_{i=j}^{t-1} \beta_i + 4 B C_\phi U_\delta \sum_{j=t-k}^{t} \left( B C_\phi^2 C_\pi \sum_{i=j-k}^{j-1} \sum_{\iota=i}^{j-1} \beta_\iota + m \rho^k + B C_\phi^2 L_\pi \sum_{i=j}^{t-1} \beta_i \right) \\
&+ 2(k+1) C_\phi U_\delta \left( (k+1) C_\phi U_\delta \sum_{j=t-2k}^{t-1} \alpha_j + C_\Theta C_\phi^2 B \sum_{j=t-2k}^{t-1} \beta_j + \frac{2 C_{gap} m \rho^k}{\lambda_{\min}} \right), \tag{17}
\end{aligned}$$

*and for NAC,*

$$G_t^\delta = 2 B^2 U_\delta C_\pi \sum_{j=t-k}^{t} \sum_{i=j}^{t-1} \beta_i + 4 B C_\phi U_\delta \sum_{j=t-k}^{t} \left( B C_\pi \sum_{i=j-k}^{j-1} \sum_{\iota=i}^{j-1} \beta_\iota + m \rho^k + B L_\pi \sum_{i=j}^{t-1} \beta_i \right)$$

$$+ 2(k+1)C_\phi U_\delta \left( (k+1)C_\phi U_\delta \sum_{j=t-2k}^{t-1} \alpha_j + C_\Theta B \sum_{j=t-2k}^{t-1} \beta_j + \frac{2C_{gap}m\rho^k}{\lambda_{\min}} \right). \tag{18}$$

In the following, we prove that $\theta_\omega^*$ defined in Equation (4) is bounded.

**Lemma 6.** *The solution $\theta_\omega^*$ to Equation (4) is bounded:*

$$\|\theta_\omega^*\|_2 \leq \frac{1}{\lambda_{\min} - dC_\phi^2 m\rho^k} \frac{mC_\phi R_{\max}}{1-\rho} = \frac{mC_\phi R_{\max}}{\bar{\lambda}_{\min}(1-\rho)}. \tag{19}$$

The proof of above Lemmas could be found in Appendix D.

## A.2 PROOFS FOR PROPOSITIONS 1 TO 4

We include the proof of Proposition 1 and Proposition 2 for completeness.

*Proof.* By the Equation (2), $\bar{\theta}_\omega^*$ satisfies that

$$\mathbb{E}_{D_{\pi_\omega}} \left[ (Q^{\pi_\omega}(s,a) - \phi_\omega^\top(s,a)\bar{\theta}_\omega^*)\phi_\omega(s,a) \right] = 0. \tag{20}$$

Since $\phi_\omega^\top \bar{\theta}_\omega^*$ is a scalar, we can get that:

$$\mathbb{E}_{D_{\pi_\omega}} \left[ Q^{\pi_\omega}(s,a)\phi_\omega(s,a) \right] = \mathbb{E}_{D_{\pi_\omega}} \left[ \phi_\omega(s,a)\phi_\omega^\top(s,a)\bar{\theta}_\omega^* \right] \tag{21}$$

For the policy gradient $\nabla J(\pi_\omega)$, we get that:

$$\nabla J(\pi_\omega) = \mathbb{E}_{D_{\pi_\omega}} \left[ \nabla \log \pi_\omega(a|s)Q^{\pi_\omega}(s,a) \right] = \mathbb{E}_{D_{\pi_\omega}} \left[ \phi_\omega(s,a)(\phi_\omega^\top(s,a)\bar{\theta}_\omega^*) \right]. \tag{22}$$

Furthermore, we have that:

$$\begin{aligned} \widetilde{\nabla} J(\pi_\omega) &= F_\omega^{-1} \mathbb{E}_{D_{\pi_\omega}} \left[ \phi_\omega(s,a)\phi_\omega(s,a)^\top \bar{\theta}_\omega^* \right] \\ &= F_\omega^{-1} \mathbb{E}_{D_{\pi_\omega}} \left[ \phi_\omega(s,a)\phi_\omega(s,a)^\top \right] \bar{\theta}_\omega^* = \bar{\theta}_\omega^*. \end{aligned} \tag{23}$$

$\square$

We present the proof of Proposition 3.

**Proposition 6.** *(Restatement of Proposition 3) For any $\omega \in \mathcal{W}$ and $\theta \in \Theta$, $\Phi_\omega \theta \neq \mathbf{e}$, where $\mathbf{e} \in \mathbb{R}^{|\mathcal{S}||\mathcal{A}|}$ is an all-one vector.*

*Proof.* Assume that there exists $\theta_c \in \Theta$ such that $\Phi_\omega \theta_c = \mathbf{e}$, then $\mathbb{E}_{D_{\pi_\omega}} \left[ \phi_\omega^\top(S,A)\theta_c \right] = 1$.

However, note that

$$\begin{aligned} \mathbb{E}_{D_{\pi_\omega}} \left[ \phi_\omega^\top(S,A)\theta_c \right] &= \sum_s d_{\pi_\omega}(s) \sum_a \pi_\omega(a|s)\phi_\omega^\top(s,a)\theta_c \\ &= \sum_s d_{\pi_\omega}(s) \sum_a \pi_\omega(a|s)\nabla \log \pi_\omega(a|s)^\top \theta_c \\ &= \sum_s d_{\pi_\omega}(s) \sum_a \pi_\omega(a|s)\frac{\nabla_\omega \pi_\omega(a|s)^\top}{\pi_\omega(a|s)}\theta_c \\ &= \sum_s d_{\pi_\omega}(s) \sum_a \nabla_\omega \pi_\omega(a|s)^\top \theta_c \\ &= \sum_s d_{\pi_\omega}(s)\nabla_\omega \left( \sum_a \pi_\omega(a|s) \right)^\top \theta_c \\ &= 0, \end{aligned} \tag{24}$$

where the last equation is from the fact that $\sum_a \pi_\omega(a|s) = 1$, and hence the gradient of it is $0$.

This hence results in a contradiction, which completes the proof. $\square$

We then present the proof of Proposition 4.

**Proposition 7.** *(Restatement of Proposition 4) For any $\omega \in \mathcal{W}$, denote the fixed point of $k$-step TD operator by $\theta_\omega^*$, and the solution to Equation (2) by $\bar{\theta}_\omega^*$, then*

$$\left\|\theta_\omega^* - \bar{\theta}_\omega^*\right\|_2 \le \frac{C_{gap} m \rho^k}{\lambda_{\min}}, \tag{25}$$

*where $C_{gap} = C_\phi^2 B + \frac{C_\phi R_{\max}}{1-\rho}$.*

*Proof.* From the definition, it holds that

$$\theta_\omega^* = \left(\mathbb{E}_{D_{\pi_\omega}}\left[\phi_\omega(s,a)\phi_\omega^\top(s,a)\right]\right)^{-1}\left(\mathbb{E}_{D_{\pi_\omega}}\left[\phi_\omega(s,a)\left(\mathcal{T}_{\pi_\omega}^{(k)}\phi_\omega^\top(s,a)\theta_\omega^*\right)\right]\right), \tag{26}$$

and

$$\bar{\theta}_\omega^* = \left(\mathbb{E}_{D_{\pi_\omega}}\left[\phi_\omega(s,a)\phi_\omega^\top(s,a)\right]\right)^{-1}\left(\mathbb{E}_{D_{\pi_\omega}}\left[\phi_\omega(s,a)Q^{\pi_\omega}(s,a)\right]\right). \tag{27}$$

Thus, we have that

$$\left\|\theta_\omega^* - \bar{\theta}_\omega^*\right\|_2$$

$$= \left\|\left(\mathbb{E}_{D_{\pi_\omega}}\left[\phi_\omega(s,a)\phi_\omega^\top(s,a)\right]\right)^{-1}\left(\mathbb{E}_{D_{\pi_\omega}}\left[\phi_\omega(s,a)\left(\mathcal{T}_{\pi_\omega}^{(k)}\phi_\omega^\top(s,a)\theta_\omega^* - Q^{\pi_\omega}(s,a)\right)\right]\right)\right\|_2$$

$$\le \frac{1}{\lambda_{\min}}\left\|\mathbb{E}_{D_{\pi_\omega}}\left[\phi_\omega(s,a)\left(\mathcal{T}_{\pi_\omega}^{(k)}\phi_\omega^\top(s,a)\theta_\omega^* - Q^{\pi_\omega}(s,a)\right)\right]\right\|_2$$

$$= \frac{1}{\lambda_{\min}}\left\|\mathbb{E}_{D_{\pi_\omega}}\left[\phi_\omega(s,a)\left(\mathbb{E}\left[\sum_{j=0}^{k-1}R_j - J(\omega) + \phi_\omega(s_k,a_k)^\top\theta_\omega^*|s_0=s,a_0=a,\pi_\omega\right] - Q^{\pi_\omega}(s,a)\right)\right]\right\|$$

$$= \frac{1}{\lambda_{\min}}\left\|\mathbb{E}_{D_{\pi_\omega}}\left[\phi_\omega(s,a)\left(\mathbb{E}\left[\sum_{j=0}^{k-1}R_j - J(\omega) + \phi_\omega(s_k,a_k)^\top\theta_\omega^*|s_0=s,a_0=a,\pi_\omega\right]\right.\right.\right.$$

$$\left.\left.\left. - \mathbb{E}\left[\sum_{j=0}^{\infty}R_j - J(\omega)|s_0=s,a_0=a,\pi_\omega\right]\right)\right]\right\|_2$$

$$= \frac{1}{\lambda_{\min}}\left\|\mathbb{E}_{D_{\pi_\omega}}\left[\phi_\omega(s,a)\left(\mathbb{E}\left[\sum_{j=k}^{\infty}R_j - J(\omega) + \phi_\omega(s_k,a_k)^\top\theta_\omega^*|s_0=s,a_0=a,\pi_\omega\right]\right)\right]\right\|_2$$

$$\overset{(a)}{\le} \frac{1}{\lambda_{\min}}\left\|\mathbb{E}_{D_{\pi_\omega}}\left[\phi_\omega(s,a)\left(\mathbb{E}\left[\sum_{j=k}^{\infty}R_j - J(\omega)\Big|(s_k,a_k)\sim D_{\pi_\omega},\pi_\omega\right]\right)\right]\right\|_2$$

$$+ \frac{1}{\lambda_{\min}}\left\|\mathbb{E}_{D_{\pi_\omega}}\left[\phi_\omega(s,a)\left(\mathbb{E}\left[\phi_\omega(s_k,a_k)^\top\theta_\omega^*|(s_k,a_k)\sim D_{\pi_\omega}\right]\right)\right]\right\|_2$$

$$+ \frac{1}{\lambda_{\min}}C_\phi^2 B \mathbb{E}_{D_{\pi_\omega}}\left[\|\mathbb{P}(s_k,a_k|s_0=s,a_0=a,\pi_\omega) - D_{\pi_\omega}\|_{\mathcal{TV}}\right]$$

$$+ \frac{1}{\lambda_{\min}}C_\phi\sum_{j=k}^{\infty}R_{\max}\mathbb{E}_{D_{\pi_\omega}}\left[\|\mathbb{P}(s_j,a_j|s_0=s,a_0=s,\pi_\omega) - D_{\pi_\omega}\|_{\mathcal{TV}}\right]$$

$$\overset{(b)}{\le} \frac{1}{\lambda_{\min}}\left\|\mathbb{E}_{D_{\pi_\omega}}\left[\phi_\omega(s,a)\left(\mathbb{E}\left[\sum_{j=k}^{\infty}R_j - J(\omega)\Big|(s_k,a_k)\sim D_{\pi_\omega},\pi_\omega\right]\right)\right]\right\|_2$$

$$+ \frac{1}{\lambda_{\min}}\left\|\mathbb{E}_{D_{\pi_\omega}}\left[\phi_\omega(s,a)\left(\mathbb{E}\left[\phi_\omega(s_k,a_k)^\top\theta_\omega^*|(s_k,a_k)\sim D_{\pi_\omega}\right]\right)\right]\right\|_2$$

$$+ \frac{C_\phi}{\lambda_{\min}}\left(C_\phi B m \rho^k + \sum_{j=k}^{\infty}R_{\max}m\rho^j\right)$$

$$\overset{(c)}{\leq} \frac{1}{\lambda_{\min}} \left\| \mathbb{E}_{D_{\pi_\omega}} \left[ \phi_\omega(s,a) \left( \mathbb{E}_{D_{\pi_\omega}} \left[ \phi_\omega(s,a)^\top \theta_\omega^* \right] \right) \right] \right\|_2 + \frac{1}{\lambda_{\min}} C_\phi \left( C_\phi B m \rho^k + \sum_{j=k}^\infty R_{\max} m \rho^j \right)$$

$$\overset{(d)}{\leq} \frac{1}{\lambda_{\min}} \left( C_\phi^2 B m \rho^k + C_\phi R_{\max} \frac{m\rho^k}{1-\rho} \right)$$

$$= \frac{C_{\mathrm{gap}} m \rho^k}{\lambda_{\min}}, \tag{28}$$

where $C_{\mathrm{gap}} = C_\phi^2 B + C_\phi R_{\max} \frac{1}{1-\rho}$, $(a)$ follows from the triangular inequality and the fact that for any probability distribution $P_1$ and $P_2$, and any random variable $X$, s.t. $|X| \leq X_{\max}$, $|\mathbb{E}_{P_1}[X] - \mathbb{E}_{P_2}[X]| \leq X_{\max} \|P_1 - P_2\|_{\mathcal{TV}}$, $(b)$ follows from Assumption 1, $(c)$ follows from $J(\omega) = \mathbb{E}_{D_{\pi_\omega}}[R(s,a)]$, and $(d)$ follows from $\mathbb{E}_{D_{\pi_\omega}} \left[ \phi_\omega(s,a) \left( \mathbb{E}_{D_{\pi_\omega}} \left[ \phi_\omega(s,a)^\top \theta_\omega^* \right] \right) \right] = 0$. $\qquad\square$

## B    AC Sample Complexity Analysis

In this section, we provide the sample complexity analysis for our single-loop AC algorithm.

### B.1    Bound on Gradient Norm in AC

In this section, we first present a preliminary bound on the gradient norm $\|\nabla J(\omega)\|$.

**Lemma 7.** *It holds that*

$$\frac{\beta_t}{2} \mathbb{E}\left[ \|\nabla J(\omega_t)\|_2^2 \right] \leq \mathbb{E}\left[ J(\omega_{t+1}) \right] - \mathbb{E}\left[ J(\omega_t) \right] + C_\phi^4 \beta_t \mathbb{E}\left[ \|\theta_t - \theta_t^*\|_2^2 \right] + G_t^\omega, \tag{29}$$

*where*

$$G_t^\omega = \frac{L_J C_\phi^4 B^2 \beta_t^2}{2} + \left( \frac{C_\phi^4 C_{gap}^2 m \rho^k}{\lambda_{\min}^2} + C_J C_\phi^2 B \right) \beta_t m \rho^k + 2 C_\phi^4 B^2 L_J \beta_t \sum_{j=t-k}^{t-1} \beta_j$$

$$+ C_J C_\phi^4 B^2 C_\pi \beta_t \sum_{j=t-k}^{t-1} \sum_{i=j}^{t-1} \beta_i. \tag{30}$$

*Proof.* Recall that in the update of AC algorithm, $\omega_{t+1} - \omega_t = \beta_t \phi_t^\top(s_t, a_t) \theta_t \phi_t(s_t, a_t)$. Following Lemma 1, it can be shown that

$$J(\omega_{t+1}) \geq J(\omega_t) + \langle \nabla J(\omega_t), \omega_{t+1} - \omega_t \rangle - \frac{L_J}{2} \|\omega_{t+1} - \omega_t\|_2^2$$

$$= J(\omega_t) + \beta_t \langle \nabla J(\omega_t), \phi_t^\top(s_t, a_t) \theta_t \phi_t(s_t, a_t) \rangle - \frac{L_J \beta_t^2}{2} \left\| \phi_t^\top(s_t, a_t) \theta_t \phi_t(s_t, a_t) \right\|_2^2$$

$$= J(\omega_t) + \beta_t \langle \nabla J(\omega_t), \phi_t^\top(s_t, a_t) \theta_t \phi_t(s_t, a_t) - \mathbb{E}_{D_t} \left[ \phi_t^\top(s,a) \theta_t \phi_t(s,a) \right] \rangle$$

$$\quad + \beta_t \langle \nabla J(\omega_t), \mathbb{E}_{D_t} \left[ \phi_t^\top(s,a) \left( \theta_t - \theta_t^* \right) \phi_t(s,a) \right] \rangle + \beta_t \langle \nabla J(\omega_t), \nabla J(\omega_t) \rangle$$

$$\quad + \beta_t \langle \nabla J(\omega_t), \mathbb{E}_{D_t} \left[ \phi_t^\top(s,a) \theta_t^* \phi_t(s,a) \right] - \nabla J(\omega_t) \rangle - \frac{L_J \beta_t^2}{2} \left\| \phi_t^\top(s_t, a_t) \theta_t \phi_t(s_t, a_t) \right\|_2^2$$

$$\geq J(\omega_t) + \beta_t \langle \nabla J(\omega_t), \phi_t^\top(s_t, a_t) \theta_t \phi_t(s_t, a_t) - \mathbb{E}_{D_t} \left[ \phi_t^\top(s,a) \theta_t \phi_t(s,a) \right] \rangle$$

$$\quad + \beta_t \|\nabla J(\omega_t)\|_2^2 - \frac{\beta_t}{4} \|\nabla J(\omega_t)\|_2^2 - \beta_t \left\| \mathbb{E}_{D_t} \left[ \phi_t^\top(s,a) \left( \theta_t - \theta_t^* \right) \phi_t(s,a) \right] \right\|_2^2$$

$$\quad - \frac{\beta_t}{4} \|\nabla J(\omega_t)\|_2^2 - \beta_t \left\| \mathbb{E}_{D_t} \left[ \phi_t^\top(s,a) \theta_t^* \phi_t(s,a) \right] - \nabla J(\omega_t) \right\|_2^2 - \frac{L_J C_\phi^4 B^2}{2} \beta_t^2$$

$$\overset{(a)}{\geq} J(\omega_t) + \beta_t \underbrace{\langle \nabla J(\omega_t), \phi_t^\top(s_t, a_t) \theta_t \phi_t(s_t, a_t) - \mathbb{E}_{D_t} \left[ \phi_t^\top(s,a) \theta_t \phi_t(s,a) \right] \rangle}_{\text{part I}}$$

$$+ \frac{\beta_t}{2} \|\nabla J(\omega_t)\|_2^2 - \beta_t \left\|\mathbb{E}_{D_t} \left[\phi_t^\top (s,a)(\theta_t - \theta_t^*)\phi_t(s,a)\right]\right\|_2^2 - \frac{L_J C_\phi^4 B^2}{2} \beta_t^2$$

$$- \beta_t \left\|\mathbb{E}_{D_t} \left[\phi_t^\top (s,a)(\theta_t^* - \bar{\theta}_t^*)\phi_t(s,a)\right]\right\|_2^2, \tag{31}$$

where we write $\bar{\theta}_{\omega_t}^*$ as $\bar{\theta}_t^*$ for convenience, $(a)$ follows from Equation (3) that $\nabla J(\omega_t) = \mathbb{E}_{D_t} \left[\phi_t^\top (s,a)\bar{\theta}_t^*\phi_t(s,a)\right]$.

We then bound part I in Equation (31). Note that

$$\left|\mathbb{E}\left[\left\langle \nabla J(\omega_t), \phi_t^\top (s_t,a_t)\theta_t\phi_t(s_t,a_t) - \mathbb{E}_{D_t}\left[\phi_t^\top (s,a)\theta_t\phi_t(s,a)\right]\right\rangle\right]\right|$$

$$\leq \left|\mathbb{E}\left[\left\langle \nabla J(\omega_t) - \nabla J(\omega_{t-k}), \phi_t^\top (s_t,a_t)\theta_t\phi_t(s_t,a_t) - \mathbb{E}_{D_t}\left[\phi_t^\top (s,a)\theta_t\phi_t(s,a)\right]\right\rangle\right]\right|$$

$$+ \left|\mathbb{E}\left[\left\langle \nabla J(\omega_{t-k}), \phi_t^\top (s_t,a_t)\theta_t\phi_t(s_t,a_t) - \mathbb{E}_{D_t}\left[\phi_t^\top (s,a)\theta_t\phi_t(s,a)\right]\right\rangle\right]\right|$$

$$\leq \mathbb{E}\left[\|\nabla J(\omega_t) - \nabla J(\omega_{t-k})\|_2 \left\|\phi_t^\top (s_t,a_t)\theta_t\phi_t(s_t,a_t) - \mathbb{E}_{D_t}\left[\phi_t^\top (s,a)\theta_t\phi_t(s,a)\right]\right\|_2\right]$$

$$+ \left|\mathbb{E}\left[\nabla^\top J(\omega_{t-k})\mathbb{E}\left[\phi_t^\top (s_t,a_t)\theta_t\phi_t(s_t,a_t) - \mathbb{E}_{D_t}\left[\phi_t^\top (s,a)\theta_t\phi_t(s,a)\right]|\mathcal{F}_{t-k}\right]\right]\right|$$

$$\overset{(a)}{\leq} 2C_\phi^2 B L_J \mathbb{E}\left[\|\omega_t - \omega_{t-k}\|_2\right] + C_J \mathbb{E}\left[\|\mathbb{P}(s_t,a_t|\mathcal{F}_{t-k}) - D_t\|_{\mathcal{TV}}\right]C_\phi^2 B$$

$$\overset{(b)}{\leq} 2C_\phi^4 B^2 L_J \sum_{j=t-k}^{t-1} \beta_j + C_J C_\phi^2 B \left(C_\pi \sum_{j=t-k}^{t-1} \mathbb{E}\left[\|\omega_t - \omega_j\|_2\right] + m\rho^k\right)$$

$$\leq 2C_\phi^4 B^2 L_J \sum_{j=t-k}^{t-1} \beta_j + C_J C_\phi^2 B \left(C_\pi C_\phi^2 B \sum_{j=t-k}^{t-1}\sum_{i=j}^{t-1} \beta_i + m\rho^k\right), \tag{32}$$

where $(a)$ is from the $L_J$-smoothness of $J$, and $(b)$ is from Lemma 4. On the other hand, from Proposition 4, we can show that

$$\left\|\mathbb{E}_{D_t}\left[\phi_t^\top (s,a)\left(\bar{\theta}_t^* - \theta_t^*\right)\phi_t(s,a)\right]\right\|_2^2 \leq C_\phi^4 \left\|\theta_t^* - \bar{\theta}_t^*\right\|_2^2 \leq C_\phi^4 \left(\frac{C_{\text{gap}}m\rho^k}{\lambda_{\min}}\right)^2. \tag{33}$$

Thus, combining Equation (31), Equation (32) and Equation (33), we have that

$$\mathbb{E}\left[J(\omega_{t+1})\right] \geq \mathbb{E}\left[J(\omega_t)\right] + \frac{\beta_t}{2}\mathbb{E}\left[\|\nabla J(\omega_t)\|_2^2\right] - \frac{L_J C_\phi^4 B^2}{2}\beta_t^2 - C_\phi^4 \beta_t \mathbb{E}\left[\|\theta_t - \theta_t^*\|_2^2\right]$$

$$- C_\phi^4 \beta_t \left(\frac{C_{\text{gap}}m\rho^k}{\lambda_{\min}}\right)^2 - 2C_\phi^4 B^2 L_J \beta_t \sum_{j=t-k}^{t-1} \beta_j$$

$$- C_J C_\phi^2 B \beta_t \left(C_\pi C_\phi^2 B \sum_{j=t-k}^{t-1}\sum_{i=j}^{t-1} \beta_i + m\rho^k\right), \tag{34}$$

which further implies that

$$\frac{\beta_t}{2}\mathbb{E}\left[\|\nabla J(\omega_t)\|_2^2\right] \leq \mathbb{E}\left[J(\omega_{t+1})\right] - \mathbb{E}\left[J(\omega_t)\right] + \frac{L_J C_\phi^4 B^2}{2}\beta_t^2 + C_\phi^4 \beta_t \mathbb{E}\left[\|\theta_t - \theta_t^*\|_2^2\right]$$

$$+ \left(\frac{C_\phi^4 C_{\text{gap}}^2 m\rho^k}{\lambda_{\min}^2} + C_J C_\phi^2 B\right)m\rho^k \beta_t$$

$$+ 2C_\phi^4 B^2 L_J \beta_t \sum_{j=t-k}^{t-1} \beta_j + C_J C_\phi^4 B^2 C_\pi \beta_t \sum_{j=t-k}^{t-1}\sum_{i=j}^{t-1} \beta_i. \tag{35}$$

This hence completes the proof. $\qquad\square$

## B.2 Bound on $\|\eta_t - J(\omega_t)\|$ in AC

In this section, we bound the error between $\eta_t$ and $J(\omega_t)$, where $J(\omega_t) = \lim_{N\to\infty} \mathbb{E}\left[\frac{1}{N}\sum_{t=0}^{N-1} R_t | \pi_{\omega_t}\right]$ is the average-reward for policy $\pi_{\omega_t}$.

**Lemma 8.** *If $\gamma_t - \gamma_t^2 \geq \beta_t$, then it holds that*

$$\mathbb{E}\left[\|\eta_{t+1} - J(\omega_{t+1})\|_2^2\right]$$
$$\leq (1 - \gamma_t)\,\mathbb{E}\left[\|\eta_t - J(\omega_t)\|_2^2\right] + \frac{C_\phi^4 B^2}{2}\beta_t \mathbb{E}\left[\|\nabla J(\omega_t)\|_2^2\right] + G_t^\eta, \tag{36}$$

*where*

$$G_t^\eta = 2\gamma_t \left( R_{\max}^2 C_\pi \sum_{j=t-k}^{t-1}\sum_{i=j}^{t-1}\beta_j + R_{\max}^2 m\rho^k + R_{\max}^2 \sum_{j=t-k}^{t-1}\gamma_j + R_{\max}C_J C_\phi^2 B \sum_{j=t-k}^{t-1}\beta_j \right)$$
$$+ R_{\max}^2\gamma_t^2 + C_J^2 C_\phi^4 B^2 \beta_t^2 + 2R_{\max}C_J C_\phi^2 B\beta_t\gamma_t + R_{\max}L_J C_\phi^4 B^2 \beta_t^2. \tag{37}$$

*Proof.* Recall the update rule in Algorithm 1. Then we have that

$$\eta_{t+1} - J(\omega_{t+1}) = \eta_t + \gamma_t\,(R_t - \eta_t) - J(\omega_{t+1}) + J(\omega_t) - J(\omega_{t+1}). \tag{38}$$

It then follows that

$$\|\eta_{t+1} - J(\omega_{t+1})\|_2^2 = \|(1 - \gamma_t)\,(\eta_t - J(\omega_t)) + \gamma_t(R_t - J(\omega_t)) + J(\omega_t) - J(\omega_{t+1})\|_2^2$$
$$\leq (1-\gamma_t)^2\|\eta_t - J(\omega_t)\|_2^2 + \gamma_t^2\|R_t - J(\omega_t)\|_2^2 + \|J(\omega_t) - J(\omega_{t+1})\|_2^2$$
$$+ 2\gamma_t \underbrace{\langle R_t - J(\omega_t), J(\omega_t) - J(\omega_{t+1})\rangle}_{\text{I}} + 2\gamma_t(1-\gamma_t)\underbrace{\langle \eta_t - J(\omega_t), R_t - J(\omega_t)\rangle}_{\text{II}}$$
$$+ 2(1-\gamma_t)\underbrace{\langle \eta_t - J(\omega_t), J(\omega_t) - J(\omega_{t+1})\rangle}_{\text{III}}. \tag{39}$$

The term $\|J(\omega_t) - J(\omega_{t+1})\|_2$ can be bounded using its Lipschitz smoothness:

$$\|J(\omega_t) - J(\omega_{t+1})\|_2 \leq C_J\|\omega_t - \omega_{t+1}\|_2 \leq C_J C_\phi^2 B\beta_t. \tag{40}$$

Term I in Equation (39) can be bounded as follows:

$$|\mathbb{E}\left[\langle R_t - J(\omega_t), J(\omega_t) - J(\omega_{t+1})\rangle\right]| \leq \mathbb{E}\left[\|R_t - J(\omega_t)\|_2\|J(\omega_t) - J(\omega_{t+1})\|_2\right]$$
$$\leq R_{\max}C_J\mathbb{E}\left[\|\omega_{t+1} - \omega_t\|_2\right]$$
$$\leq R_{\max}C_J C_\phi^2 B\beta_t. \tag{41}$$

Term II in Equation (39) can be bounded as follows

$$|\mathbb{E}\left[\langle \eta_t - J(\omega_t), R_t - J(\omega_t)\rangle\right]|$$
$$\leq |\mathbb{E}\left[\langle \eta_{t-k} - J(\omega_{t-k}), R_t - J(\omega_t)\rangle\right]| + |\mathbb{E}\left[\langle \eta_t - \eta_{t-k} - J(\omega_t) + J(\omega_{t-k}), R_t - J(\omega_t)\rangle\right]|$$
$$\leq |\mathbb{E}\left[\mathbb{E}\left[\langle \eta_{t-k} - J(\omega_{t-k}), R_t - J(\omega_t)\rangle | \mathcal{F}_{t-k}\right]\right]|$$
$$\quad + \mathbb{E}\left[\|\eta_t - \eta_{t-k} - J(\omega_t) + J(\omega_{t-k})\|_2\|R_t - J(\omega_t)\|_2\right]$$
$$\overset{(a)}{\leq} R_{\max}^2 \mathbb{E}\left[\|\mathbb{P}\left(s_t, a_t | \mathcal{F}_{t-k}\right), D_t\|_{\mathcal{TV}}\right] + \mathbb{E}\left[\|\eta_t - \eta_{t-k}\|_2 + \|J(\omega_t) - J(\omega_{t-k})\|_2\right]R_{\max}$$
$$\overset{(b)}{\leq} \left| R_{\max}^2 \left( C_\pi \sum_{j=t-k}^{t-1}\mathbb{E}\left[\|\omega_t - \omega_j\|_2\right] + m\rho^k \right) \right|$$
$$\quad + \left( R_{\max}\sum_{j=t-k}^{t-1}\gamma_j + C_J\mathbb{E}\left[\|\omega_t - \omega_{t-k}\|_2\right] \right)R_{\max}$$

$$\leq R_{\max}^2 C_\pi \sum_{j=t-k}^{t-1} \sum_{i=j}^{t-1} \beta_j + R_{\max}^2 m\rho^k + R_{\max}^2 \sum_{j=t-k}^{t-1} \gamma_j + R_{\max} C_J C_\phi^2 B \sum_{j=t-k}^{t-1} \beta_j, \tag{42}$$

where $(a)$ follows from that $0 \leq \eta_t \leq R_{\max}$ when $\eta_0 = 0$ and $\mathbb{E}_{D_t}[R(s,a) - J(\omega_t)] = 0$. From $0 \leq J(\omega_t) \leq R$ and $0 \leq R_t \leq R_{\max}$, it holds that $|\langle \eta_{t-k} - J(\omega_{t-k}), R_t - J(\omega_t) \rangle| \leq R_{\max}^2$. $(b)$ follows from Lemma 4.

Term III in Equation (39) can be bounded as follows

$$|\mathbb{E}\left[\langle \eta_t - J(\omega_t), J(\omega_t) - J(\omega_{t+1})\rangle\right]|$$

$$\overset{(a)}{\leq} \left|\mathbb{E}\left[\langle \eta_t - J(\omega_t), -\nabla^\top J(\omega_t)(\omega_{t+1} - \omega_t)\rangle\right]\right|$$

$$\quad + \left|\mathbb{E}\left[\left\langle \eta_t - J(\omega_t), (\omega_{t+1} - \omega_t)^\top \frac{\nabla^2 J(\hat{\omega}_t)}{2}(\omega_{t+1} - \omega_t)\right\rangle\right]\right|$$

$$= \beta_t \left|\mathbb{E}\left[\langle \eta_t - J(\omega_t), -\nabla^\top J(\omega_t)(\phi_t^\top(s_t, a_t)\theta_t\phi_t(s_t, a_t))\rangle\right]\right|$$

$$\quad + \beta_t^2 \left|\mathbb{E}\left[\left\langle \eta_t - J(\omega_t), (\phi_t^\top(s_t, a_t)\theta_t\phi_t(s_t, a_t))^\top \frac{\nabla^2 J(\hat{\omega}_t)}{2}(\phi_t^\top(s_t, a_t)\theta_t\phi_t(s_t, a_t))\right\rangle\right]\right|$$

$$\overset{(b)}{\leq} \frac{\beta_t}{2}\mathbb{E}\left[\|\eta_t - J(\omega_t)\|_2^2\right] + \frac{C_\phi^4 B^2}{2}\beta_t\mathbb{E}\left[\|\nabla J(\omega_t)\|_2^2\right] + R_{\max}L_J C_\phi^4 B^2 \beta_t^2, \tag{43}$$

where $(a)$ follows from the Lagrange's Mean Value Theorem and Lemma 1 for some $\hat{\omega}_t = \lambda\omega_t + (1-\lambda)\omega_{t+1}$ with $\lambda \in [0,1]$; and $(b)$ follows from $\langle a, b\rangle \leq \frac{\|a\|^2 + \|b\|^2}{2}$ and the fact that $\left\|\nabla^2 J(\omega)\right\|_2 \leq L_J$.

Combining Equation (39), Equation (41), Equation (42) and Equation (43) implies

$$\mathbb{E}\left[\|\eta_{t+1} - J(\omega_{t+1})\|_2^2\right] \leq \left((1-\gamma_t)^2 + \beta_t\right)\mathbb{E}\left[\|\eta_t - J(\omega_t)\|_2^2\right] + \beta_t C_\phi^4 B^2 \mathbb{E}\left[\|\nabla J(\omega_t)\|_2^2\right]$$

$$+ 2\gamma_t\left(R_{\max}^2 C_\pi \sum_{j=t-k}^{t-1} \sum_{i=j}^{t-1} \beta_i + R_{\max}^2 m\rho^k + R_{\max}^2 \sum_{j=t-k}^{t-1} \gamma_j + R_{\max} C_J C_\phi^2 B\beta_t\right)$$

$$+ R_{\max}^2 \gamma_t^2 + C_J^2 C_\phi^4 B^2 \beta_t^2 + 2R_{\max} C_J C_\phi^2 B\gamma_t \sum_{j=t-k}^{t-1} \beta_j + 2R_{\max}L_J C_\phi^4 B^2 \beta_t^2, \tag{44}$$

which completes the proof. $\qquad\square$

### B.3 Tracking Error Analysis of AC

In this section, we bound the tracking error $\|\theta_t - \theta_t^*\|_2$. Recall that we write $\theta_t^* = \theta_{\omega_t}^*$ and $\theta_\omega^*$ is the solution to Equation (4), i.e.,

$$\mathbb{E}_{D_{\pi_\omega}}\left[\phi_\omega^\top(s,a)\left(\mathcal{T}_{\pi_\omega}^{(k)}\phi_\omega^\top(s,a)\theta_\omega^* - \phi_\omega^\top(s,a)\theta_\omega^*\right)\right] = \mathbf{0}. \tag{45}$$

We then present a recursive bound on the tracking error in the following lemma.

**Lemma 9.** *Set the step sizes such that*

$$\frac{\bar{\lambda}_{\min}\alpha_t}{2} \geq \frac{L_J + 2C_\phi}{\lambda_{\min}}\beta_t, \tag{46}$$

*then it holds that*

$$\mathbb{E}\left[\|\theta_{t+1} - \theta_{t+1}^*\|_2^2\right] \leq \left(1 - \frac{\bar{\lambda}_{\min}\alpha_t}{2}\right)\mathbb{E}\left[\|\theta_t - \theta_t^*\|_2^2\right]$$

$$+ \frac{L_J}{\lambda_{\min}}\beta_t\mathbb{E}\left[\|\nabla J(\omega_t)\|_2^2\right] + \frac{(k+1)^2 C_\phi^2}{\bar{\lambda}_{\min}}\alpha_t\mathbb{E}\left[\|\eta_t - J(\omega_t)\|_2^2\right] + G_t^\theta, \tag{47}$$

*where*

$$G_t^\theta = \frac{(k+1)C_\phi U_\delta L_J}{\lambda_{\min}}\beta_t \sum_{j=t-k}^{t-1}\alpha_t + 2(k+1)C_\phi^3 BU_\delta C_\Theta \alpha_t \beta_t + \frac{2(k+1)C_\phi U_\delta L_J}{\lambda_{\min}}\beta_t \sum_{j=t-k}^{t-1}\alpha_t$$

$$+ 2\left(BC_\phi^2 C_{gap}L_J + 2L_J C_{gap} + 2BC_\phi^2 L_J \lambda_{\min}\right)\frac{m\rho^k}{\lambda_{\min}^2}\beta_t + \frac{4(k+1)C_\phi U_\delta C_{gap}m\rho^k}{\lambda_{\min}}\alpha_t + 2\alpha_t G_t^\delta$$

$$+ \left(2BC_\phi L_J(C_\phi L_\pi + 2L_\phi) + 2BL_\Theta \lambda_{\min} + 2BC_\phi^2 L_J C_\pi \lambda_{\min} + C_\Theta \lambda_{\min}\right)\frac{BC_\phi^2}{\lambda_{\min}^2}\beta_t \sum_{j=t-k}^{t}\beta_j. \tag{48}$$

*Proof.* From Algorithm 1, it holds that

$$
\begin{aligned}
\left\|\theta_{t+1} - \theta_{t+1}^*\right\|_2^2 &= \left\|\Pi_B\left(\theta_t + \alpha_t \delta_t z_t\right) - \theta_{t+1}^*\right\|_2^2 \\
&\overset{(a)}{\le} \left\|\theta_t + \alpha_t \delta_t z_t - \theta_{t+1}^*\right\|_2^2 \\
&\le \left\|\theta_t + \alpha_t \delta_t z_t - \theta_t^* + \theta_t^* - \theta_{t+1}^*\right\|_2^2 \\
&\le \left\|\theta_t - \theta_t^*\right\|_2^2 + \alpha_t^2 \left\|\delta_t z_t\right\|_2^2 + \left\|\theta_t^* - \theta_{t+1}^*\right\|_2^2 + 2\alpha_t \left\langle \theta_t - \theta_t^*, \delta_t z_t\right\rangle \\
&\quad + 2\alpha_t \left\langle \delta_t z_t, \theta_t^* - \theta_{t+1}^*\right\rangle + 2\left\langle \theta_t - \theta_t^*, \theta_t^* - \theta_{t+1}^*\right\rangle, \tag{49}
\end{aligned}
$$

where $(a)$ follows from the fact $\|\Pi_B(x) - y\|_2 \le \|x - y\|_2$ when $\|y\|_2 \le B$ and $\left\|\theta_{t+1}^*\right\|_2 \le B$. Taking expectations on both sides further implies that

$$
\begin{aligned}
\mathbb{E}\left[\left\|\theta_{t+1} - \theta_{t+1}^*\right\|_2^2\right] &\le \mathbb{E}\left[\left\|\theta_t - \theta_t^*\right\|_2^2\right] + \alpha_t^2 \mathbb{E}\left[\left\|\delta_t z_t\right\|_2^2\right] + \mathbb{E}\left[\left\|\theta_t^* - \theta_{t+1}^*\right\|_2^2\right] \\
&\quad + 2\alpha_t \underbrace{\mathbb{E}\left[\left\langle \theta_t - \theta_t^*, \delta_t z_t\right\rangle\right]}_{\text{I}} + 2\alpha_t \underbrace{\mathbb{E}\left[\left\langle \delta_t z_t, \theta_t^* - \theta_{t+1}^*\right\rangle\right]}_{\text{II}} + 2\underbrace{\mathbb{E}\left[\left\langle \theta_t - \theta_t^*, \theta_t^* - \theta_{t+1}^*\right\rangle\right]}_{\text{III}}. \tag{50}
\end{aligned}
$$

The term $\left\|\theta_t^* - \theta_{t+1}^*\right\|_2$ can be bounded as follows

$$
\begin{aligned}
\left\|\theta_t^* - \theta_{t+1}^*\right\|_2 &= \left\|\bar\theta_t^* - \bar\theta_{t+1}^* + \theta_t^* - \bar\theta_t^* - \theta_{t+1}^* + \bar\theta_{t+1}^*\right\|_2 \\
&\le \left\|\bar\theta_t^* - \bar\theta_{t+1}^*\right\|_2 + \left\|\theta_t^* - \bar\theta_t^*\right\|_2 + \left\|\theta_{t+1}^* - \bar\theta_{t+1}^*\right\|_2 \\
&\overset{(a)}{\le} \left\|\bar\theta_t^* - \bar\theta_{t+1}^*\right\|_2 + \frac{C_{gap}m\rho^k}{\lambda_{\min}} + \frac{C_{gap}m\rho^k}{\lambda_{\min}} \\
&\overset{(b)}{\le} C_\Theta \left\|\omega_t - \omega_{t+1}\right\|_2 + \frac{2C_{gap}m\rho^k}{\lambda_{\min}} \\
&\le \beta_t C_\Theta \left\|\phi_t^\top(s_t, a_t)\theta_t \phi_t(s_t, a_t)\right\|_2 + \frac{2C_{gap}m\rho^k}{\lambda_{\min}} \\
&\le C_\Theta C_\phi^2 B\beta_t + \frac{2C_{gap}m\rho^k}{\lambda_{\min}}, \tag{51}
\end{aligned}
$$

where $(a)$ follows from Proposition 4; and $(b)$ follows from Lemma 2. Equation (51) further implies that

$$\mathbb{E}\left[\left\|\theta_t^* - \theta_{t+1}^*\right\|_2^2\right] \le 2C_\Theta^2 C_\phi^4 B^2 \beta_t^2 + \frac{8C_{gap}^2 m^2 \rho^{2k}}{\lambda_{\min}^2}. \tag{52}$$

By Lemma 5, we can bound term I in Equation (50) as follows

$$\mathbb{E}\left[\left\langle \theta_t - \theta_t^*, \delta_t z_t\right\rangle\right] \le -\frac{\bar\lambda_{\min}}{2}\mathbb{E}\left[\left\|\theta_t - \theta_t^*\right\|_2^2\right] + \frac{(k+1)^2 C_\phi^2}{2\bar\lambda_{\min}}\mathbb{E}\left[\left\|J(\omega_t) - \eta_t\right\|_2^2\right] + G_t^\delta. \tag{53}$$

For term II in Equation (50), we have that

$$\mathbb{E}\left[\langle \delta_t z_t, \theta_t^* - \theta_{t+1}^* \rangle\right] \le \mathbb{E}\left[\|\delta_t z_t\|_2 \|\theta_t^* - \theta_{t+1}^*\|_2\right]$$

$$\le (k+1)C_\phi U_\delta \left(C_\Theta C_\phi^2 B\beta_t + \frac{2C_{\text{gap}}m\rho^k}{\lambda_{\min}}\right)$$

$$= (k+1)C_\phi^3 B U_\delta C_\Theta \beta_t + \frac{2(k+1)C_\phi U_\delta C_{\text{gap}}m\rho^k}{\lambda_{\min}}, \tag{54}$$

where the last inequality is from Lemma 2 and Proposition 4.

To bound term III in Equation (50), note that

$$\mathbb{E}\left[\langle \theta_t - \theta_t^*, \theta_t^* - \theta_{t+1}^* \rangle\right]$$

$$= \mathbb{E}\left[\langle \theta_t - \theta_t^*, \bar{\theta}_t^* - \bar{\theta}_{t+1}^* \rangle\right] + \mathbb{E}\left[\langle \theta_t - \theta_t^*, \theta_t^* - \bar{\theta}_t^* \rangle\right] + \mathbb{E}\left[\langle \theta_t - \theta_t^*, \theta_{t+1}^* - \bar{\theta}_{t+1}^* \rangle\right]$$

$$\le \mathbb{E}\left[\langle \theta_t - \theta_t^*, F_t^{-1}\nabla J(\omega_t) - F_{t+1}^{-1}\nabla J(\omega_{t+1}) \rangle\right] + \frac{\beta_t}{2}\mathbb{E}\left[\|\theta_t - \theta_t^*\|_2^2\right] + \frac{1}{2\beta_t}\mathbb{E}\left[\|\theta_t^* - \bar{\theta}_t^*\|_2^2\right]$$

$$\quad + \frac{\beta_t}{2}\mathbb{E}\left[\|\theta_t - \theta_t^*\|_2^2\right] + \frac{1}{2\beta_t}\mathbb{E}\left[\|\theta_{t+1}^* - \bar{\theta}_{t+1}^*\|_2^2\right]$$

$$\overset{(a)}{\le} \mathbb{E}\left[\langle \theta_t - \theta_t^*, F_{t+1}^{-1}\left(\nabla J(\omega_t) - \nabla J(\omega_{t+1})\right)\rangle\right] + \mathbb{E}\left[\langle \theta_t - \theta_t^*, \left(F_t^{-1} - F_{t+1}^{-1}\right)\nabla J(\omega_t)\rangle\right]$$

$$\quad + \beta_t\mathbb{E}\left[\|\theta_t - \theta_t^*\|_2^2\right] + \frac{1}{\beta_t}\left(\frac{C_{\text{gap}}m\rho^k}{\lambda_{\min}}\right)^2$$

$$\overset{(b)}{\le} \mathbb{E}\left[\langle \theta_t - \theta_t^*, F_{t+1}^{-1}\nabla^2 J(\hat{\omega}_t)\left(\omega_t - \omega_{t+1}\right)\rangle\right] + \frac{1}{\lambda_{\min}^2}\mathbb{E}\left[\|\theta_t - \theta_t^*\|_2 \|F_t - F_{t+1}\|_2 \|\nabla J(\omega_t)\|_2\right]$$

$$\quad + \beta_t\mathbb{E}\left[\|\theta_t - \theta_t^*\|_2^2\right] + \frac{1}{\beta_t}\left(\frac{C_{\text{gap}}m\rho^k}{\lambda_{\min}}\right)^2$$

$$\overset{(c)}{\le} \beta_t\mathbb{E}\left[\langle \theta_t - \theta_t^*, F_{t+1}^{-1}\nabla^2 J(\hat{\omega}_t)\left(\phi_t^\top(s_t, a_t)\theta_t\phi_t(s_t, a_t)\right)\rangle\right] + \frac{BC_\phi^3\left(C_\phi L_\pi + 2L_\phi\right)}{2\lambda_{\min}^2}\beta_t\mathbb{E}\left[\|\theta_t - \theta_t^*\|_2^2\right]$$

$$\quad + \frac{BC_\phi^3\left(C_\phi L_\pi + 2L_\phi\right)}{2\lambda_{\min}^2}\beta_t\mathbb{E}\left[\|\nabla J(\omega_t)\|_2^2\right] + \beta_t\mathbb{E}\left[\|\theta_t - \theta_t^*\|_2^2\right] + \frac{1}{\beta_t}\left(\frac{C_{\text{gap}}m\rho^k}{\lambda_{\min}}\right)^2$$

$$= \underbrace{\beta_t\mathbb{E}\left[\langle \theta_t - \theta_t^*, F_{t+1}^{-1}\nabla^2 J(\hat{\omega}_t)\mathbb{E}_{D_t}\left[\phi_t^\top(s, a)\bar{\theta}_t^*\phi_t(s, a)\right]\rangle\right]}_{(i)}$$

$$\quad + \underbrace{\beta_t\mathbb{E}\left[\langle \theta_t - \theta_t^*, F_{t+1}^{-1}\nabla^2 J(\hat{\omega}_t)\mathbb{E}_{D_t}\left[\phi_t^\top(s, a)(\theta_t^* - \bar{\theta}_t^*)\phi_t(s, a)\right]\rangle\right]}_{(ii)}$$

$$\quad + \underbrace{\beta_t\mathbb{E}\left[\langle \theta_t - \theta_t^*, F_{t+1}^{-1}\nabla^2 J(\hat{\omega}_t)\mathbb{E}_{D_t}\left[\phi_t^\top(s, a)(\theta_t - \theta_t^*)\phi_t(s, a)\right]\rangle\right]}_{(iii)}$$

$$\quad + \underbrace{\beta_t\mathbb{E}\left[\langle \theta_t - \theta_t^*, F_{t+1}^{-1}\nabla^2 J(\hat{\omega}_t)\left(\phi_t^\top(s_t, a_t)\theta_t\phi_t(s_t, a_t)\right) - \mathbb{E}_{D_t}\left[\phi_t^\top(s, a)\theta_t\phi_t(s, a)\right]\rangle\right]}_{(iv)}$$

$$\quad + \frac{BC_\phi^3\left(C_\phi L_\pi + 2L_\phi\right)}{2\lambda_{\min}^2}\beta_t\mathbb{E}\left[\|\nabla J(\omega_t)\|_2^2\right] + \frac{BC_\phi^3\left(C_\phi L_\pi + 2L_\phi\right) + 2\lambda_{\min}^2}{2\lambda_{\min}^2}\beta_t\mathbb{E}\left[\|\theta_t - \theta_t^*\|_2^2\right]$$

$$\quad + \frac{1}{\beta_t}\left(\frac{C_{\text{gap}}m\rho^k}{\lambda_{\min}}\right)^2. \tag{55}$$

where $(a)$ follows from Proposition 4, $(b)$ follows from that $\left\|F_t^{-1} - F_{t+1}^{-1}\right\|_2 \le \left\|F_t^{-1}F_{t+1}^{-1}\right\|_2 \|F_t - F_{t+1}\|_2$ and $(c)$ is from that

$$\|F_{t+1} - F_t\|_2 = \left\|\mathbb{E}_{D_{t+1}}\left[\phi_{t+1}(s, a)\phi_{t+1}^\top(s, a)\right] - \mathbb{E}_{D_t}\left[\phi_t(s, a)\phi_t^\top(s, a)\right]\right\|_2$$

$$\le \left\|\mathbb{E}_{D_{t+1}}\left[\phi_{t+1}(s, a)\phi_{t+1}^\top(s, a)\right] - \mathbb{E}_{D_t}\left[\phi_{t+1}(s, a)\phi_{t+1}^\top(s, a)\right]\right\|_2$$

$$+ \left\| \mathbb{E}_{D_t} \left[ \phi_{t+1}(s,a)\phi_{t+1}^\top(s,a) \right] - \mathbb{E}_{D_t} \left[ \phi_t(s,a)\phi_t^\top(s,a) \right] \right\|_2$$

$$\leq C_\phi^2 \| D_{t+1} - D_t \|_{\mathcal{TV}} + \mathbb{E} \left[ \left( \| \phi_t(s,a) \|_2 + \| \phi_{t+1}(s,a) \|_2 \right) \| \phi_t(s,a) - \phi_{t+1}(s,a) \|_2 \right]$$

$$\leq C_\phi^2 L_\pi \| \omega_t - \omega_{t+1} \|_2 + 2C_\phi L_\phi \| \omega_t - \omega_{t+1} \|_2$$

$$= \left( C_\phi^2 L_\pi + 2C_\phi L_\phi \right) \beta_t \left\| \phi_t^\top(s_t,a_t)\theta_t \phi_t(s_t,a_t) \right\|_2$$

$$\leq BC_\phi^3 \left( C_\phi L_\pi + 2L_\phi \right) \beta_t. \tag{56}$$

**Lemma 10.** *For any* $\omega, \omega' \in \mathcal{W}$, $\left\| \nabla^2 J(\omega) - \nabla^2 J(\omega') \right\|_2 \leq L_\Theta \| \omega - \omega' \|_2$, *where* $L_\Theta > 0$ *is the Lipschitz constant.*

*Proof.* According to (Heidergott & Hordijk, 2003), $\nabla D_{\pi_\omega}$ and $\nabla D_{\pi_\omega}$ are Lipschitz and bounded when $\nabla \pi_\omega(s,a)$ and $\nabla^2 \pi_\omega(s,a)$ are Lipschitz and bounded, which also are applied in (Olshevsky & Gharesifard, 2022).

From Assumption 2, $\nabla \phi_\omega(s,a) = \nabla^2 \log \pi_\omega(s,a)$ is bounded and Lipschitz continuous. From Lemma 1, $\nabla J(\omega)$ is bounded and Lipschitz continuous. Next, $\nabla^2 J(\omega) = \sum_{(s,a)} \nabla^2 D_{\pi_\omega}(s,a) R(s,a) = \mathbb{E}_{D_{\pi_\omega}} \left[ \frac{\nabla^2 D_{\pi_\omega}(s,a)}{D_{\pi_\omega}(s,a)} R(s,a) \right]$ is bounded and Lipschitz continuous. $\square$

We then consider the term $(i)$,

$$(i) = \beta_t \mathbb{E} \left[ \langle \theta_t - \theta_t^*, F_{t+1}^{-1} \nabla^2 J(\hat{\omega}_t) \nabla J(\omega_t) \rangle \right]$$

$$\leq \beta_t \mathbb{E} \left[ \| \theta_t - \theta_t^* \|_2 \left\| F_{t+1}^{-1} \right\|_2 \left\| \nabla^2 J(\hat{\omega}_t) \right\|_2 \| \nabla J(\omega_t) \|_2 \right]$$

$$\leq \frac{L_J \beta_t}{\lambda_{\min}} \left( \frac{1}{2} \mathbb{E} \left[ \| \theta_t - \theta_t^* \|_2^2 \right] + \frac{1}{2} \mathbb{E} \left[ \| \nabla J(\omega_t) \|_2^2 \right] \right), \tag{57}$$

where the first inequality follows from Lemma 1.

Next, we consider the term $(ii)$,

$$(ii) \leq \beta_t \mathbb{E} \left[ \| \theta_t - \theta_t^* \|_2 \left\| F_{t+1}^{-1} \right\|_2 \left\| \nabla^2 J(\hat{\omega}_t) \right\|_2 C_\phi^2 \left\| \theta_t^* - \bar{\theta}_t^* \right\|_2 \right]$$

$$\leq \frac{BC_\phi^2 L_J \beta_t}{\lambda_{\min}} \frac{C_{\text{gap}} m \rho^k}{\lambda_{\min}} = \frac{BC_\phi^2 C_{\text{gap}} L_J m \rho^k \beta_t}{\lambda_{\min}^2}, \tag{58}$$

where the last inequality follows from Lemma 1 and Proposition 4.

Then, consider the term $(iii)$,

$$(iii) \leq \beta_t \mathbb{E} \left[ \| \theta_t - \theta_t^* \|_2 \left\| F_{t+1}^{-1} \right\|_2 \left\| \nabla^2 J(\hat{\omega}_t) \right\|_2 C_\phi^2 \| \theta_t - \theta_t^* \|_2 \right]$$

$$\leq \frac{C_\phi^2 L_J \beta_t}{\lambda_{\min}} \mathbb{E} \left[ \| \theta_t - \theta_t^* \|_2^2 \right]. \tag{59}$$

Consider the term $(iv)$,

$$(iv) = \beta_t \mathbb{E} \left[ \langle (F_{t+1}^{-1} \nabla^2 J(\hat{\omega}_t))^\top (\theta_t - \theta_t^*), \left( \phi_t^\top(s_t,a_t)\theta_t \phi_t(s_t,a_t) \right) - \mathbb{E}_{D_t} \left[ \phi_t^\top(s,a)\theta_t \phi_t(s,a) \right] \rangle \right]$$

$$\leq \beta_t \mathbb{E}[\langle ((F_{t+1}^{-1} \nabla^2 J(\hat{\omega}_t))^\top (\theta_t - \theta_t^*) - (F_{t-k}^{-1} \nabla^2 J(\hat{\omega}_{t-k}))^\top (\theta_{t-k} - \theta_{t-k}^*)),$$

$$\left( \phi_t^\top(s_t,a_t)\theta_t \phi_t(s_t,a_t) \right) - \mathbb{E}_{D_t} \left[ \phi_t^\top(s,a)\theta_t \phi_t(s,a) \right] \rangle]$$

$$+ \beta_t \mathbb{E} \left[ \langle (F_{t-k}^{-1} \nabla^2 J(\hat{\omega}_{t-k}))^\top (\theta_{t-k} - \theta_{t-k}^*), \left( \phi_t^\top(s_t,a_t)\theta_t \phi_t(s_t,a_t) \right) - \mathbb{E}_{D_t} \left[ \phi_t^\top(s,a)\theta_t \phi_t(s,a) \right] \rangle \right]$$

$$\leq \beta_t \mathbb{E} \left[ \left\| (F_{t+1}^{-1} \nabla^2 J(\hat{\omega}_t))^\top (\theta_t - \theta_t^*) - (F_{t-k}^{-1} \nabla^2 J(\hat{\omega}_{t-k-1}))^\top (\theta_{t-k} - \theta_{t-k}^*) \right\|_2 2C_\phi^2 B \right]$$

$$+ \beta_t \mathbb{E}[\langle (F_{t-k}^{-1} \nabla^2 J(\hat{\omega}_{t-k}))^\top (\theta_{t-k} - \theta_{t-k}^*), \phi_t^\top(s_t,a_t)\theta_t \phi_t(s_t,a_t) - \mathbb{E}_{D_t} \left[ \phi_t^\top(s,a)\theta_t \phi_t(s,a) \right] \rangle]$$

$$\leq \beta_t \mathbb{E} \left[ \left\| (F_{t+1}^{-1} \nabla^2 J(\hat{\omega}_t))^\top (\theta_t - \theta_t^*) - (F_{t-k}^{-1} \nabla^2 J(\hat{\omega}_{t-k-1}))^\top (\theta_{t-k} - \theta_{t-k}^*) \right\|_2 2C_\phi^2 B \right]$$

$$+ \beta_t \mathbb{E} \left[ \frac{2BC_\phi^2 L_J}{\lambda_{\min}} \| \mathbb{P}(s_t,a_t | \mathcal{F}_{t-k}) - D_t \|_{\mathcal{TV}} \right]$$

$$\overset{(a)}{\leq} \beta_t \mathbb{E}\left[\left\|(F_{t+1}^{-1}\nabla^2 J(\hat{\omega}_t))^\top(\theta_t - \theta_t^*) - (F_{t-k}^{-1}\nabla^2 J(\hat{\omega}_{t-k-1}))^\top(\theta_{t-k} - \theta_{t-k}^*)\right\|_2 2C_\phi^2 B\right]$$

$$+ \frac{2BC_\phi^2 L_J \beta_t}{\lambda_{\min}}\left(C_\pi \sum_{j=t-k}^{t-1}\|\omega_t - \omega_j\|_2 + m\rho^k\right), \tag{60}$$

where $(a)$ follows from Lemma 4.

Consider the term $\left\|(F_{t+1}^{-1}\nabla^2 J(\hat{\omega}_t))^\top(\theta_t - \theta_t^*) - (F_{t-k}^{-1}\nabla^2 J(\hat{\omega}_{t-k-1}))^\top(\theta_{t-k} - \theta_{t-k}^*)\right\|_2$, we have that

$$\left\|(F_{t+1}^{-1}\nabla^2 J(\hat{\omega}_t))^\top(\theta_t - \theta_t^*) - (F_{t-k}^{-1}\nabla^2 J(\hat{\omega}_{t-k-1}))^\top(\theta_{t-k} - \theta_{t-k}^*)\right\|_2$$

$$\leq 2BL_J\left\|F_{t+1}^{-1} - F_{t-k}^{-1}\right\|_2 + \frac{2B}{\lambda_{\min}}\left\|\nabla^2 J(\hat{\omega}_t) - \nabla^2 J(\hat{\omega}_{t-k-1})\right\|_2$$

$$+ \frac{L_J}{\lambda_{\min}}\left(\|\theta_t - \theta_{t-k}\|_2 + \left\|\theta_t^* - \theta_{t-k}^*\right\|_2\right)$$

$$\overset{(a)}{\leq} \frac{2B^2 L_J C_\phi^3 (C_\phi L_\pi + 2L_\phi)}{\lambda_{\min}^2}\sum_{j=t-k}^{t}\beta_j + \frac{2B}{\lambda_{\min}}\left\|\nabla^2 J(\hat{\omega}_t) - \nabla^2 J(\hat{\omega}_{t-k-1})\right\|_2$$

$$+ \frac{L_J}{\lambda_{\min}}\left(\|\theta_t - \theta_{t-k}\|_2 + \left\|\bar{\theta}_t^* - \bar{\theta}_{t-k}^*\right\|_2 + \left\|\bar{\theta}_t^* - \theta_t^*\right\|_2 + \left\|\theta_{t-k}^* - \bar{\theta}_{t-k}^*\right\|_2\right)$$

$$\overset{(b)}{\leq} \frac{2B^2 L_J C_\phi^3 (C_\phi L_\pi + 2L_\phi)}{\lambda_{\min}^2}\sum_{j=t-k}^{t}\beta_j + \frac{2BL_\Theta}{\lambda_{\min}}\left\|\omega_{t+1} - \omega_{t-k-1}\right\|_2$$

$$+ \frac{L_J}{\lambda_{\min}}\left(\|\theta_t - \theta_{t-k}\|_2 + C_\Theta\|\omega_t - \omega_{t-k}\|_2 + \frac{2C_{\mathrm{gap}}m\rho^k}{\lambda_{\min}}\right)$$

$$\leq \frac{2B^2 L_J C_\phi^3 (C_\phi L_\pi + 2L_\phi)}{\lambda_{\min}^2}\sum_{j=t-k}^{t}\beta_j + \frac{2B^2 C_\phi^2 L_\Theta}{\lambda_{\min}}\sum_{j=t-k}^{t}\beta_t$$

$$+ \frac{L_J}{\lambda_{\min}}\left((k+1)C_\phi U_\delta \sum_{j=t-k}^{t-1}\alpha_t + BC_\phi^2 C_\Theta \sum_{j=t-k}^{t-1}\beta_j + \frac{2C_{\mathrm{gap}}m\rho^k}{\lambda_{\min}}\right)$$

$$= \frac{(k+1)C_\phi U_\delta L_J}{\lambda_{\min}}\sum_{j=t-k}^{t-1}\alpha_t + \frac{2C_{\mathrm{gap}}L_J m\rho^k}{\lambda_{\min}^2}$$

$$+ \frac{BC_\phi^2}{\lambda_{\min}^2}\left(2BC_\phi L_J(C_\phi L_\pi + 2L_\phi) + 2BL_\Theta\lambda_{\min} + C_\Theta\lambda_{\min}\right)\sum_{j=t-k}^{t}\beta_j, \tag{61}$$

where $(a)$ follows from Equation (60) and $(b)$ follows from Lemma 10.

Above all, the term $(iv)$ can be bounded as:

$$(iv) \leq \frac{(k+1)C_\phi U_\delta L_J}{\lambda_{\min}}\beta_t\sum_{j=t-k}^{t-1}\alpha_t + \frac{2C_{\mathrm{gap}}L_J m\rho^k}{\lambda_{\min}^2}\beta_t + \frac{2BC_\phi^2 L_J m\rho^k}{\lambda_{\min}}\beta_t$$

$$+ \frac{BC_\phi^2}{\lambda_{\min}^2}\left(2BC_\phi L_J(C_\phi L_\pi + 2L_\phi) + 2BL_\Theta\lambda_{\min} + 2BC_\phi^2 L_J C_\pi\lambda_{\min} + C_\Theta\lambda_{\min}\right)\beta_t\sum_{j=t-k}^{t}\beta_j. \tag{62}$$

Combine with term $(i)$, $(ii)$, $(iii)$ and $(iv)$, we have that

$$\mathbb{E}\left[\langle\theta_t - \theta_t^*, \theta_t^* - \theta_{t+1}^*\rangle\right] \leq \frac{L_J + 2C_\phi}{2\lambda_{\min}}\beta_t\mathbb{E}\left[\|\theta_t - \theta_t^*\|_2^2\right] + \frac{L_J}{2\lambda_{\min}}\beta_t\mathbb{E}\left[\|\nabla J(\omega_t)\|_2^2\right]$$

$$+ \left(BC_\phi^2 C_{\text{gap}} L_J + 2L_J C_{\text{gap}} + 2BC_\phi^2 L_J \lambda_{\min}\right) \frac{m\rho^k}{\lambda_{\min}^2} \beta_t + \frac{(k+1)C_\phi U_\delta L_J}{\lambda_{\min}} \beta_t \sum_{j=t-k}^{t-1} \alpha_t$$

$$+ \frac{BC_\phi^2}{\lambda_{\min}^2} \left(2BC_\phi L_J(C_\phi L_\pi + 2L_\phi) + 2BL_\Theta \lambda_{\min} + 2BC_\phi^2 L_J C_\pi \lambda_{\min} + C_\Theta \lambda_{\min}\right) \beta_t \sum_{j=t-k}^{t} \beta_j. \tag{63}$$

This bounds term III in Equation (50).

Plugging bounds on term I, II, III in Equation (50) further implies

$$E\left[\|\theta_{t+1} - \theta_{t+1}^*\|_2^2\right] \le \left(1 - \bar{\lambda}_{\min}\alpha_t + \frac{L_J + 2C_\phi}{\lambda_{\min}}\beta_t\right) \mathbb{E}\left[\|\theta_t - \theta_t^*\|_2^2\right] + \frac{L_J}{\lambda_{\min}}\beta_t \mathbb{E}\left[\|\nabla J(\omega_t)\|_2^2\right]$$

$$+ \frac{(k+1)^2 C_\phi^2}{\bar{\lambda}_{\min}}\alpha_t \mathbb{E}\left[\|\eta_t - J(\omega_t)\|_2^2\right] + \frac{(k+1)C_\phi U_\delta L_J}{\lambda_{\min}}\beta_t \sum_{j=t-k}^{t-1} \alpha_t + 2(k+1)C_\phi^3 BU_\delta C_\Theta \alpha_t \beta_t$$

$$+ \frac{2(k+1)C_\phi U_\delta L_J}{\lambda_{\min}}\beta_t \sum_{j=t-k}^{t-1} \alpha_t + 2\left(BC_\phi^2 C_{\text{gap}} L_J + 2L_J C_{\text{gap}} + 2BC_\phi^2 L_J \lambda_{\min}\right) \frac{m\rho^k}{\lambda_{\min}^2}\beta_t$$

$$+ \left(2BC_\phi L_J(C_\phi L_\pi + 2L_\phi) + 2BL_\Theta \lambda_{\min} + 2BC_\phi^2 L_J C_\pi \lambda_{\min} + C_\Theta \lambda_{\min}\right) \frac{BC_\phi^2}{\lambda_{\min}^2}\beta_t \sum_{j=t-k}^{t} \beta_j$$

$$+ \frac{4(k+1)C_\phi U_\delta C_{\text{gap}} m\rho^k}{\lambda_{\min}}\alpha_t + 2\alpha_t G_t^\delta, \tag{64}$$

which completes the proof. $\square$

## B.4 Sample Complexity of AC

We first present our proof of Proposition 5.

**Proposition 8.** *(Restatement of Proposition 5) With the constant step sizes, the tracking error of the AC algorithm in Algorithm 1 can be bounded as follows:*

$$\frac{1}{T}\sum_{t=0}^{T-1} \mathbb{E}\left[\|\theta_t^* - \theta_t\|_2^2\right] \le \left(\frac{c_\alpha \beta}{\alpha} + \frac{c_\eta \beta}{\gamma}\right) \frac{1}{T}\sum_{t=0}^{T-1} \mathbb{E}\left[\|\nabla J(\omega_t)\|_2^2\right] + \mathcal{O}\left(\frac{1}{T\alpha}\right) + \mathcal{O}\left(\frac{\log^2 T}{T\gamma}\right)$$

$$+ \mathcal{O}\left(\alpha \log^2 T\right) + \mathcal{O}\left(\beta \log^3 T\right) + \mathcal{O}\left(\gamma \log^3 T\right) + \mathcal{O}\left(\frac{\beta^2 \log^2 T}{\alpha}\right) + \mathcal{O}\left(\frac{\beta^2}{\gamma}\right), \tag{65}$$

*where $c_\alpha$ and $c_\eta$ are constants defined later.*

*Proof.* Recall that in Lemma 9, we showed

$$\mathbb{E}\left[\|\theta_{t+1} - \theta_{t+1}^*\|_2^2\right] \le \left(1 - \frac{\bar{\lambda}_{\min}\alpha}{2}\right) \mathbb{E}\left[\|\theta_t - \theta_t^*\|_2^2\right]$$

$$+ \frac{L_J}{\lambda_{\min}}\beta_t \mathbb{E}\left[\|\nabla J(\omega_t)\|_2^2\right] + \frac{(k+1)^2 C_\phi^2}{\bar{\lambda}_{\min}}\alpha_t \mathbb{E}\left[\|\eta_t - J(\omega_t)\|_2^2\right] + G_t^\theta, \tag{66}$$

where for any $0 \le t \le T$,

$$G_t^\theta \equiv \mathcal{O}\left(k(m\rho^k)\alpha + k(m\rho^k)\beta + \frac{(m\rho^k)^2}{\beta} + k^3\alpha^2 + k^3\alpha\beta + k^2\beta^2\right). \tag{67}$$

Apply this inequality recursively, we have that

$$\mathbb{E}\left[\|\theta_t - \theta_t^*\|_2^2\right] \le \left(1 - \frac{\bar{\lambda}_{\min}\alpha}{2}\right)^t \left[\|\theta_0 - \theta_0^*\|_2^2\right] + \frac{L_J}{\lambda_{\min}}\beta \sum_{j=0}^{t} \left(1 - \frac{\bar{\lambda}_{\min}\alpha}{2}\right)^{t-j} \mathbb{E}\left[\|\nabla J(\omega_j)\|_2^2\right]$$

$$+ \frac{(k+1)^2 C_\phi^2}{\bar{\lambda}_{\min}} \alpha \sum_{j=0}^{t} \left(1 - \frac{\bar{\lambda}_{\min}\alpha}{2}\right)^{t-j} \mathbb{E}\left[\|J(\omega_j) - \eta_j\|_2^2\right] + \sum_{j=0}^{t} \left(1 - \frac{\bar{\lambda}_{\min}\alpha}{2}\right)^{t-j} G_j^\theta$$

$$\leq (1-q)^t \left[\|\theta_0 - \theta_0^*\|_2^2\right] + \frac{L_J}{\lambda_{\min}} \beta \sum_{j=0}^{t} (1-q)^{t-j} \mathbb{E}\left[\|\nabla J(\omega_j)\|_2^2\right]$$

$$+ \frac{(k+1)^2 C_\phi^2}{\bar{\lambda}_{\min}} \alpha \sum_{j=0}^{t} (1-q)^{t-j} \mathbb{E}\left[\|J(\omega_j) - \eta_j\|_2^2\right] + \sum_{j=0}^{t} (1-q)^{t-j} G_j^\theta, \tag{68}$$

where $q = \frac{\bar{\lambda}_{\min}\alpha}{2}$.

Summing the inequality above w.r.t. $t$ from $0$ to $T-1$ further implies that

$$\frac{1}{T} \sum_{t=0}^{T-1} \mathbb{E}\left[\|\theta_t - \theta_t^*\|_2^2\right]$$

$$\leq \frac{1}{T} \sum_{t=0}^{T-1} (1-q)^t \left[\|\theta_0 - \theta_0^*\|_2^2\right] + \frac{L_J \beta}{\lambda_{\min} T} \sum_{t=0}^{T-1} \sum_{j=0}^{t} (1-q)^{t-j} \mathbb{E}\left[\|\nabla J(\omega_j)\|_2^2\right]$$

$$+ \frac{k^2 C_\phi^2}{\bar{\lambda}_{\min}} \alpha \frac{1}{T} \sum_{t=0}^{T-1} \sum_{j=0}^{t} (1-q)^{t-j} \mathbb{E}\left[\|J(\omega_j) - \eta_j\|_2^2\right] + \frac{1}{T} \sum_{t=0}^{T-1} \sum_{j=0}^{t} (1-q)^{t-j} G_j^\theta$$

$$\leq \frac{4B^2}{Tq} + \frac{\beta L_J}{\lambda_{\min} T} \sum_{j=0}^{T-1} \sum_{t=j}^{T-1} (1-q)^{t-j} \mathbb{E}\left[\|\nabla J(\omega_j)\|_2^2\right]$$

$$+ \frac{(k+1)^2 C_\phi^2}{\bar{\lambda}_{\min}} \alpha \frac{1}{T} \sum_{j=0}^{T-1} \sum_{t=j}^{T-1} (1-q)^{t-j} \mathbb{E}\left[\|J(\omega_j) - \eta_j\|_2^2\right] + \frac{1}{T} \sum_{t=0}^{T-1} \sum_{j=0}^{t} (1-q)^{t-j} G_j^\theta$$

$$\leq \frac{4B^2}{Tq} + \frac{L_J \beta}{\lambda_{\min} Tq} \sum_{j=0}^{T-1} \mathbb{E}\left[\|\nabla J(\omega_j)\|_2^2\right] + \frac{(k+1)^2 C_\phi^2}{\bar{\lambda}_{\min}} \frac{\alpha}{Tq} \sum_{j=0}^{T-1} \mathbb{E}\left[\|J(\omega_j) - \eta_j\|_2^2\right] + \frac{1}{Tq} \sum_{t=0}^{T-1} G_t^\theta, \tag{69}$$

where the last inequality is from the double-sum trick: $\sum_{t=0}^{T} \sum_{j=0}^{t} k^{t-j} X_j \leq (\sum_{t=0}^{T} X_t)(\sum_{t=0}^{T} k^t) \leq \frac{\sum_{t=0}^{T} X_t}{1-k}$.

Recall that we showed in Appendix B.2 that

$$\mathbb{E}\left[\|\eta_{t+1} - J(\omega_{t+1})\|_2^2\right] \leq (1-\gamma) \mathbb{E}\left[\|\eta_t - J(\omega_t)\|_2^2\right] + \frac{\beta}{2} C_\phi^4 B^2 \mathbb{E}\left[\|\nabla J(\omega_t)\|_2^2\right] + G_t^\eta, \tag{70}$$

where for $t$ from $0$ to $T$,

$$G_t^\eta \equiv \mathcal{O}\left((m\rho^k)\gamma + \beta^2 + k^2\beta\gamma + k\gamma^2\right). \tag{71}$$

Recursively applying this inequality implies that

$$\mathbb{E}\left[\|\eta_t - J(\omega_t)\|_2^2\right]$$

$$\leq (1-\gamma)^t \|\eta_0 - J(\omega_0)\|_2^2 + \frac{B^2 C_\phi^4}{2} \beta \sum_{j=0}^{t-1} (1-\gamma)^{t-j} \mathbb{E}\left[\|\nabla J(\omega_j)\|_2^2\right] + \sum_{j=0}^{t-1} (1-\gamma)^{t-j} G_t^\eta$$

$$\leq R_{\max}^2 (1-\gamma)^t + \frac{B^2 C_\phi^4}{2} \beta \sum_{j=0}^{t-1} (1-\gamma)^{t-j} \mathbb{E}\left[\|\nabla J(\omega_j)\|_2^2\right] + \sum_{j=0}^{t-1} (1-\gamma)^{t-j} G_t^\eta. \tag{72}$$

We then sum the above inequality w.r.t. $t$ from $0$ to $T-1$, and have that

$$\frac{1}{T} \sum_{t=0}^{T-1} \mathbb{E}\left[\|\eta_t - J(\omega_t)\|_2^2\right] \leq R_{\max}^2 \frac{1}{T} \sum_{t=0}^{T-1} (1-\gamma)^t \|\eta_0 - J(\omega_0)\|_2^2$$

$$+ \frac{B^2 C_\phi^4}{2} \frac{\beta}{T} \sum_{t=0}^{T-1} \sum_{j=0}^{t-1} (1-\gamma)^{t-j} \mathbb{E}\left[\|\nabla J(\omega_j)\|_2^2\right] + \frac{1}{T} \sum_{t=0}^{T-1} \sum_{j=0}^{t-1} (1-\gamma)^{t-j} G_t^\eta$$

$$\leq \frac{R_{\max}^2}{T\gamma} + \frac{B^2 C_\phi^4}{2} \frac{\beta}{T} \sum_{j=0}^{T-1} \sum_{t=j}^{T-1} (1-\gamma)^{t-j} \mathbb{E}\left[\|\nabla J(\omega_j)\|_2^2\right] + \frac{1}{T\gamma} \sum_{t=0}^{T-1} G_t^\eta$$

$$\leq \frac{R_{\max}^2}{T\gamma} + \frac{B^2 C_\phi^4}{2} \frac{\beta}{T} \sum_{j=0}^{T-1} \frac{1}{\gamma} \mathbb{E}\left[\|\nabla J(\omega_j)\|_2^2\right] + \frac{1}{T\gamma} \sum_{t=0}^{T-1} G_t^\eta$$

$$= \frac{R_{\max}^2}{T\gamma} + \frac{B^2 C_\phi^4}{2} \frac{\beta}{T\gamma} \sum_{j=0}^{T-1} \mathbb{E}\left[\|\nabla J(\omega_j)\|_2^2\right] + \frac{1}{T\gamma} \sum_{t=0}^{T-1} G_t^\eta, \tag{73}$$

where we use the double-sum trick again.

Plugging Equation (73) in Equation (69) further implies that

$$\frac{1}{T} \sum_{t=0}^{T-1} \mathbb{E}\left[\|\theta_t - \theta_t^*\|_2^2\right]$$

$$\leq \frac{4B^2}{Tq} + \frac{L_J}{\bar{\lambda}_{\min}^2} \frac{\beta}{T\alpha} \sum_{t=0}^{T-1} \mathbb{E}\left[\|\nabla J(\omega_t)\|_2^2\right]$$

$$+ \frac{2(k+1)^2 C_\phi^2}{(\bar{\lambda}_{\min})^2} \frac{1}{T} \sum_{t=0}^{T-1} \mathbb{E}\left[\|J(\omega_t) - \eta_t\|_2^2\right] + \frac{1}{Tq} \sum_{t=0}^{T-1} G_t^\theta$$

$$\leq \left(\frac{L_J \beta}{\bar{\lambda}_{\min}^2 \alpha} + \frac{B^2 (k+1)^2 C_\phi^6 \beta}{(\bar{\lambda}_{\min})^2 \gamma}\right) \frac{1}{T} \sum_{t=0}^{T-1} \mathbb{E}\left[\|\nabla J(\omega_t)\|_2^2\right] + \frac{4B^2}{Tq} + \frac{1}{Tq} \sum_{t=0}^{T-1} G_t^\theta$$

$$+ \frac{2(k+1)^2 C_\phi^2 R_{\max}^2}{(\bar{\lambda}_{\min})^2 T\gamma} + \frac{2(k+1)^2 C_\phi^2}{(\bar{\lambda}_{\min})^2} \frac{1}{T\gamma} \sum_{t=0}^{T-1} G_t^\eta$$

$$= \left(\frac{c_\alpha \beta}{\alpha} + \frac{c_\eta \beta}{\gamma}\right) \frac{1}{T} \sum_{t=0}^{T-1} \mathbb{E}\left[\|\nabla J(\omega_t)\|_2^2\right] + \frac{4B^2}{Tq} + \frac{1}{Tq} \sum_{t=0}^{T-1} G_t^\theta$$

$$+ \frac{2(k+1)^2 C_\phi^2 R_{\max}^2}{(\bar{\lambda}_{\min})^2 T\gamma} + \frac{2(k+1)^2 C_\phi^2}{(\bar{\lambda}_{\min})^2} \frac{1}{T\gamma} \sum_{t=0}^{T-1} G_t^\eta, \tag{74}$$

where $c_\alpha = \frac{L_J}{\bar{\lambda}_{\min}^2}$ and $c_\eta = \frac{B^2(k+1)^2 C_\phi^6}{(\bar{\lambda}_{\min})^2}$.

This completes the proof of Proposition 5. $\qquad\square$

We are now ready to prove Theorem 1.

**Theorem 3.** *(Restatement of Theorem 1) Consider the AC algorithm in Algorithm 1 with constant step sizes, it holds that*

$$\frac{1}{T} \sum_{t=0}^{T-1} \mathbb{E}\left[\|\nabla J(\omega_t)\|_2^2\right] \leq C_\phi^4 \frac{1}{T} \sum_{t=0}^{T-1} \mathbb{E}\left[\|\theta_t^* - \theta_t\|_2^2\right] + \mathcal{O}\left(\frac{1}{T\beta}\right) + \mathcal{O}\left(\beta \log^2 T\right). \tag{75}$$

*If further set* $\gamma = \mathcal{O}(\frac{1}{\sqrt{T}}), \alpha = \mathcal{O}(\frac{1}{\sqrt{T} \log^2 T}), \beta = \mathcal{O}(\frac{1}{\sqrt{T} \log^2 T})$, *we have that*

$$\frac{1}{T} \sum_{t=0}^{T-1} \mathbb{E}\left[\|\nabla J(\omega_t)\|_2^2\right] \leq \mathcal{O}\left(\frac{\log^3 T}{\sqrt{T}}\right). \tag{76}$$

*Proof.* In equation 74, if we set $\beta \leq \frac{\lambda_{\min}q}{8C_\phi^4 L_J} = \frac{\alpha\bar{\lambda}_{\min}^2}{16C_\phi^4 L_J}$, and plug Equation (29) in Equation (74), and we have that

$$\frac{1}{T}\sum_{t=0}^{T-1}\mathbb{E}\left[\|\theta_t - \theta_t^*\|_2^2\right]$$

$$\leq \left(\frac{c_\alpha\beta}{\alpha} + \frac{c_\eta\beta}{\gamma}\right)\frac{1}{T}\sum_{t=0}^{T-1}\mathbb{E}\left[\|\nabla J(\omega_t)\|_2^2\right] + \frac{4B^2}{Tq} + \frac{1}{Tq}\sum_{t=0}^{T-1}G_t^\theta$$

$$+ \frac{2(k+1)^2 C_\phi^2 R_{\max}^2}{(\bar{\lambda}_{\min})^2 T\gamma} + \frac{2(k+1)^2 C_\phi^2}{(\bar{\lambda}_{\min})^2}\frac{1}{T\gamma}\sum_{t=0}^{T-1}G_t^\eta$$

$$\leq \frac{4B^2}{Tq} + \frac{1}{4C_\phi^4 T\beta}\left(\mathbb{E}\left[J(\omega_{t+1})\right] - J(\omega_0)\right) + \frac{1}{4}\frac{1}{T}\sum_{t=0}^{T-1}\mathbb{E}\left[\|\theta_t - \theta_t^*\|_2^2\right] + \frac{1}{4C_\phi^4 T\beta}\sum_{t=0}^{T-1}G_t^\omega$$

$$+ \frac{2(k+1)^2 C_\phi^2}{(\bar{\lambda}_{\min})^2}\left(\frac{R_{\max}^2}{T\gamma} + \frac{B^2 C_\phi^4}{2}\frac{\beta}{T\gamma}\sum_{j=0}^{T-1}\mathbb{E}\left[\|\nabla J(\omega_j)\|_2^2\right] + \frac{1}{T\gamma}\sum_{t=0}^{T-1}G_t^\eta\right) + \frac{1}{Tq}\sum_{t=0}^{T-1}G_t^\theta$$

$$\leq \frac{1}{4C_\phi^4 T\beta}\left(\mathbb{E}\left[J(\omega_{t+1})\right] - J(\omega_0)\right) + \frac{1}{4}\frac{1}{T}\sum_{t=0}^{T-1}\mathbb{E}\left[\|\theta_t - \theta_t^*\|_2^2\right] + \frac{4B^2}{Tq} + \frac{1}{4C_\phi^4\beta}\frac{1}{T}\sum_{t=0}^{T-1}G_t^\omega$$

$$+ \frac{2(k+1)^2 C_\phi^2}{(\bar{\lambda}_{\min})^2}\left(\frac{R_{\max}^2}{T\gamma} + \frac{1}{T\gamma}\sum_{t=0}^{T-1}G_t^\eta\right) + \frac{(k+1)^2 C_\phi^6 B\beta}{(\bar{\lambda}_{\min})^2\gamma}\left(\frac{2}{T\beta}\left(\mathbb{E}\left[J(\omega_{t+1})\right] - J(\omega_0)\right)\right)$$

$$+ \frac{(k+1)^2 C_\phi^6 B\beta}{(\bar{\lambda}_{\min})^2\gamma}\left(2C_\phi^4\frac{1}{T}\sum_{t=0}^{T-1}\mathbb{E}\left[\|\theta_t - \theta_t^*\|_2^2\right] + \frac{2}{\beta}\frac{1}{T}\sum_{t=0}^{T-1}G_t^\omega\right) + \frac{1}{Tq}\sum_{t=0}^{T-1}G_t^\theta. \tag{77}$$

If we set $\frac{2(k+1)^2 C_\phi^{10} B\beta}{(\bar{\lambda}_{\min})^2\gamma} \leq \frac{1}{4}$, it then follows that

$$\frac{1}{T}\sum_{t=0}^{T-1}\mathbb{E}\left[\|\theta_t - \theta_t^*\|_2^2\right]$$

$$\leq \frac{1}{4C_\phi^4 T\beta}\left(\mathbb{E}\left[J(\omega_{t+1})\right] - J(\omega_0)\right) + \frac{1}{4}\frac{1}{T}\sum_{t=0}^{T-1}\mathbb{E}\left[\|\theta_t - \theta_t^*\|_2^2\right] + \frac{4B^2}{Tq} + \frac{1}{4C_\phi^4\beta}\frac{1}{T}\sum_{t=0}^{T-1}G_t^\omega$$

$$+ \frac{2(k+1)^2 C_\phi^2}{(\bar{\lambda}_{\min})^2}\left(\frac{R_{\max}^2}{T\gamma} + \frac{1}{T\gamma}\sum_{t=0}^{T-1}G_t^\eta\right) + \frac{1}{8C_\phi^4}\left(\frac{2}{T\beta}\left(\mathbb{E}\left[J(\omega_{t+1})\right] - J(\omega_0)\right)\right)$$

$$+ \frac{1}{8C_\phi^4}\left(2C_\phi^4\frac{1}{T}\sum_{t=0}^{T-1}\mathbb{E}\left[\|\theta_t - \theta_t^*\|_2^2\right] + \frac{2}{\beta}\frac{1}{T}\sum_{t=0}^{T-1}G_t^\omega\right) + \frac{1}{Tq}\sum_{t=0}^{T-1}G_t^\theta$$

$$= \frac{1}{2C_\phi^4 T\beta}\left(\mathbb{E}\left[J(\omega_{t+1})\right] - J(\omega_0)\right) + \frac{1}{2}\frac{1}{T}\sum_{t=0}^{T-1}\mathbb{E}\left[\|\theta_t - \theta_t^*\|_2^2\right] + \frac{4B^2}{Tq} + \frac{1}{2C_\phi^4\beta}\frac{1}{T}\sum_{t=0}^{T-1}G_t^\omega$$

$$+ \frac{2(k+1)^2 C_\phi^2}{(\bar{\lambda}_{\min})^2}\left(\frac{R_{\max}^2}{T\gamma} + \frac{1}{T\gamma}\sum_{t=0}^{T-1}G_t^\eta\right) + \frac{1}{Tq}\sum_{t=0}^{T-1}G_t^\theta. \tag{78}$$

This further implies that

$$\frac{1}{T}\sum_{t=0}^{T-1}\mathbb{E}\left[\|\theta_t - \theta_t^*\|_2^2\right] \leq \frac{1}{C_\phi^4 T\beta}\left(\mathbb{E}\left[J(\omega_{t+1})\right] - J(\omega_0)\right)$$

$$+ \frac{8B^2}{Tq} + \frac{1}{C_\phi^4 T\beta}\sum_{t=0}^{T-1}G_t^\omega + \frac{4(k+1)^2 C_\phi^2}{(\bar{\lambda}_{\min})^2}\left(\frac{R_{\max}^2}{T\gamma} + \frac{1}{T\gamma}\sum_{t=0}^{T-1}G_t^\eta\right) + \frac{2}{Tq}\sum_{t=0}^{T-1}G_t^\theta. \tag{79}$$

Next, we choose the stepsizes to minimize the tracking error and the gradient norm. We choose $\gamma = \frac{1}{\sqrt{T}}$ and $k \geq \left\lceil \frac{\log T}{1-\rho} \right\rceil$. Then, $\bar{\lambda}_{\min} \geq \frac{\lambda_{\min}}{2}$. We set $\alpha$ and $\beta$ such that $\alpha = \frac{C_\Theta \lambda_{\min}}{2k^2 C_\phi^6 B^2} \gamma$ and $\beta = \min \left\{ \frac{\lambda_{\min}}{2C_\Theta + 4C_\Theta C_\phi^2 + 4} \alpha, \frac{\lambda_{\min}}{16 C_\phi^4 C_\Theta} \alpha, \frac{\lambda_{\min}^2}{4k^2 C_\phi^6 B^2} \gamma \right\}$. It holds that

$$\gamma = \mathcal{O}(\frac{1}{\sqrt{T}}), \alpha = \mathcal{O}(\frac{1}{\sqrt{T} \log^2 T}), \beta = \mathcal{O}(\frac{1}{\sqrt{T} \log^2 T}), q = \mathcal{O}(\frac{1}{\sqrt{T} \log^2 T}). \tag{80}$$

With the setp size, the orders of the following terms can be determined:

$$\frac{1}{T} \sum_{t=0}^{T-1} G_t^\omega = \mathcal{O}\left((m\rho^k)\beta + k^2\beta^2\right) = \mathcal{O}\left(\frac{1}{T \log^2 T}\right);$$

$$\frac{1}{T} \sum_{t=0}^{T-1} G_t^\eta = \mathcal{O}\left((m\rho^k)\gamma + \beta^2 + k^2\beta\gamma + k\gamma^2\right) = \mathcal{O}\left(\frac{\log T}{T}\right);$$

$$\frac{1}{T} \sum_{t=0}^{T-1} G_t^\theta = \mathcal{O}\left(k(m\rho^k)\alpha + k(m\rho^k)\beta + \frac{(m\rho^k)^2}{\beta} + k^3\alpha^2 + k^3\alpha\beta + k^2\beta^2\right) = \mathcal{O}\left(\frac{1}{T \log T}\right). \tag{81}$$

Then Equation (79) can be bounded as

$$\frac{1}{T} \sum_{t=0}^{T-1} \mathbb{E}\left[\|\theta_t - \theta_t^*\|_2^2\right] = \mathcal{O}\left(\frac{\log^3 T}{\sqrt{T}}\right). \tag{82}$$

Now we involve Equation (29), and have that

$$\frac{1}{T} \sum_{t=0}^{T-1} \mathbb{E}\left[\|\nabla J(\omega_t)\|_2^2\right] \leq \frac{2\left(\mathbb{E}\left[J(\omega_{t+1})\right] - J(\omega_0)\right)}{T\beta} + \frac{2C_\phi^4}{T} \sum_{t=0}^{T-1} \mathbb{E}\left[\|\theta_t - \theta_t^*\|_2^2\right] + \frac{2}{T\beta} \sum_{t=0}^{T-1} G_t^\omega. \tag{83}$$

Plugging Equation (79) in Equation (83) implies

$$\frac{1}{T} \sum_{t=0}^{T-1} \mathbb{E}\left[\|\nabla J(\omega_t)\|_2^2\right]$$

$$\leq 2C_\phi^4 \left(\frac{\mathbb{E}\left[J(\omega_{t+1})\right] - J(\omega_0)}{C_\phi^4 T\beta} + \frac{8B^2}{Tq} + \frac{4(k+1)^2 C_\phi^2}{(\bar{\lambda}_{\min})^2} \frac{4R_{\max}^2}{T\gamma}\right) + \frac{2}{T\beta} \sum_{t=0}^{T-1} \mathbb{E}\left[\|\theta_t - \theta_t^*\|_2^2\right]$$

$$+ 2C_\phi^4 \left(\frac{1}{C_\phi^4 T\beta} \sum_{t=0}^{T-1} G_t^\omega + \frac{4(k+1)^2 C_\phi^2}{(\bar{\lambda}_{\min})^2} \frac{1}{T\gamma} \sum_{t=0}^{T-1} G_t^\eta + \frac{2}{Tq} \sum_{t=0}^{T-1} G_t^\theta\right) + \frac{2}{T\beta} \sum_{t=0}^{T-1} G_t^\omega$$

$$\leq \frac{16 C_\phi^4 B^2}{Tq} + \frac{8(k+1)^2 C_\phi^6}{(\bar{\lambda}_{\min})^2} \left(\frac{4R_{\max}^2}{T\gamma} + \frac{1}{T} \sum_{t=0}^{T-1} G_t^\eta\right) + \frac{4C_\phi^4}{Tq} \sum_{t=0}^{T-1} G_t^\theta$$

$$+ \frac{4}{T\beta}\left(\mathbb{E}\left[J(\omega_{t+1})\right] - J(\omega_0)\right) + \frac{4}{T\beta} \sum_{t=0}^{T-1} G_t^\omega. \tag{84}$$

After plugging in the step sizes set above, we have that

$$\frac{1}{T} \sum_{t=0}^{T-1} \mathbb{E}\left[\|\nabla J(\omega_t)\|_2^2\right] = \mathcal{O}\left(\frac{\log^3 T}{\sqrt{T}}\right), \tag{85}$$

which completes the proof. $\qquad\square$

## C  NAC Sample Complexity Analysis

In this section, we provide the sample complexity analysis of NAC.

### C.1  Bound on Gradient Norm in NAC

Recall that in Algorithm 1, NAC updates the policy parameter as follows: $\omega_{t+1} - \omega_t = \beta_t \theta_t$, which directly implies that

$$\|\omega_{t+1} - \omega_t\|_2 \le \beta_t \|\theta_t\|_2 \le B\beta_t. \tag{86}$$

We denote by $\lambda_{\mathrm{m}}$ the maximum eigenvalue of the matrix $\mathbb{E}_{D_t}\left[\phi_t^\top(s,a)\phi_t(s,a)\right]$. Then, by Lemma 1, we can show that

$$
\begin{aligned}
J(\omega_{t+1}) &\ge J(\omega_t) + \langle \nabla J(\omega_t), \omega_{t+1} - \omega_t \rangle - \frac{L_J}{2}\|\omega_{t+1} - \omega_t\|_2^2 \\
&\ge J(\omega_t) + \beta_t \langle \nabla J(\omega_t), \theta_t \rangle - \frac{\beta_t^2 L_J}{2}\|\theta_t\|_2^2 \\
&\ge J(\omega_t) + \beta_t \langle \nabla J(\omega_t), \theta_t - \theta_t^* \rangle + \beta_t \langle \nabla J(\omega_t), \theta_t^* - \bar{\theta}_t^* \rangle + \beta_t \langle \nabla J(\omega_t), \bar{\theta}_t^* \rangle - \frac{L_J B^2 \beta_t^2}{2} \\
&\ge J(\omega_t) - \frac{\beta_t}{4\lambda_{\mathrm{m}}}\|\nabla J(\omega_t)\|_2^2 - \lambda_{\mathrm{m}}\beta_t \|\theta_t - \theta_t^*\|_2^2 - \frac{\beta_t}{4\lambda_{\mathrm{m}}}\|\nabla J(\omega_t)\|_2^2 - \lambda_{\mathrm{m}}\beta_t \|\theta_t^* - \bar{\theta}_t^*\|_2^2 \\
&\quad + \beta_t \left\langle \nabla J(\omega_t), \left(\mathbb{E}_{D_t}\left[\phi_t^\top(s,a)\phi_t(s,a)\right]\right)^{-1} \nabla J(\omega_t) \right\rangle - \frac{L_J B^2 \beta_t^2}{2} \\
&\ge J(\omega_t) + \frac{\beta_t}{\lambda_{\mathrm{m}}}\|\nabla J(\omega_t)\|_2^2 - \frac{\beta_t}{2\lambda_{\mathrm{m}}}\|\nabla J(\omega_t)\|_2^2 - \lambda_{\mathrm{m}}\beta_t \|\theta_t - \theta_t^*\|_2^2 - \frac{C_{\mathrm{gap}} m\rho^k \beta_t}{\lambda_{\min}} \\
&\quad - \frac{L_J B^2 \beta_t^2}{2} \\
&= J(\omega_t) + \frac{\beta_t}{2\lambda_{\mathrm{m}}}\|\nabla J(\omega_t)\|_2^2 - \lambda_{\mathrm{m}}\beta_t \|\theta_t - \theta_t^*\|_2^2 - \frac{C_{\mathrm{gap}} m\rho^k \beta_t}{\lambda_{\min}} - \frac{L_J B^2 \beta_t^2}{2}. 
\end{aligned} \tag{87}
$$

Taking the expectation on both sides, we have that

$$\mathbb{E}\left[\|\nabla J(\omega_t)\|_2^2\right] \le \frac{\mathbb{E}\left[J(\omega_{t+1})\right] - \mathbb{E}\left[J(\omega_t)\right]}{\beta_t} + 2\lambda_{\mathrm{m}}^2 \mathbb{E}\left[\|\theta_t - \theta_t^*\|_2^2\right] + \frac{\lambda_{\mathrm{m}} C_{\mathrm{gap}} m\rho^k}{\lambda_{\min}} + \frac{L_J B^2 \beta_t}{2}. \tag{88}$$

### C.2  Bound on $\|\eta_t - J(\omega_t)\|$ in NAC

In this section, we bound the term $\eta_t - J(\omega_t)$ for the NAC algorithm.

**Lemma 11.** *If we denote*

$$
\begin{aligned}
\widetilde{G}_t^\eta = 2\gamma_t &\left( R_{\max}^2 C_\pi \sum_{j=t-k}^{t-1}\sum_{i=j}^{t-1}\beta_j + R_{\max}^2 m\rho^k + R_{\max}^2 \sum_{j=t-k}^{t-1}\gamma_j + BC_J R_{\max}\sum_{j=t-k}^{t-1}\beta_j \right) \\
&+ 2R_{\max}^2 \gamma_t^2 + C_J^2 B^2 \beta_t^2 + 2BC_J R_{\max}\beta_t \gamma_t + 2R_{\max} L_J B^2 \beta_t^2,
\end{aligned} \tag{89}
$$

*and set $\gamma_t - \gamma_t^2 \ge \beta_t$, then it holds that*

$$\mathbb{E}\left[\|\eta_{t+1} - J(\omega_{t+1})\|_2^2\right] \le \left((1-\gamma_t)^2 + \beta_t\right)\mathbb{E}\left[\|\eta_t - J(\omega_t)\|_2^2\right] + \beta_t B^2 \mathbb{E}\left[\|\nabla J(\omega_t)\|_2^2\right] + \widetilde{G}_t^\eta. \tag{90}$$

*Proof.* Similar to the AC analysis in Appendix B.2, we have that

$$\|\eta_{t+1} - J(\omega_{t+1})\|_2^2 = \|(1-\gamma_t)(\eta_t - J(\omega_t)) + \gamma_t(R_t - J(\omega_t)) + J(\omega_t) - J(\omega_{t+1})\|_2^2$$

$$\leq (1 - \gamma_t)^2 \|\eta_t - J(\omega_t)\|_2^2 + \gamma_t^2 \|R_t - J(\omega_t)\|_2^2 + \|J(\omega_t) - J(\omega_{t+1})\|_2^2$$
$$+ 2\gamma_t \underbrace{\langle R_t - J(\omega_t), J(\omega_t) - J(\omega_{t+1})\rangle}_{\text{I}} + 2\gamma_t (1 - \gamma_t) \underbrace{\langle \eta_t - J(\omega_t), R_t - J(\omega_t)\rangle}_{\text{II}}$$
$$+ 2(1 - \gamma_t) \underbrace{\langle \eta_t - J(\omega_t), J(\omega_t) - J(\omega_{t+1})\rangle}_{\text{III}}. \tag{91}$$

The term $\|J(\omega_t) - J(\omega_{t+1})\|_2$ can be bounded using its Lipschitz smoothness as follows

$$\|J(\omega_t) - J(\omega_{t+1})\|_2 \leq C_J \|\omega_t - \omega_{t+1}\|_2 \leq C_J B \beta_t. \tag{92}$$

Term II in Equation (91) can be bounded as follows

$$|\mathbb{E}\left[\langle \eta_t - J(\omega_t), R_t - J(\omega_t)\rangle\right]|$$
$$\leq |\mathbb{E}\left[\langle \eta_{t-k} - J(\omega_{t-k}), R_t - J(\omega_t)\rangle\right]| + |\mathbb{E}\left[\langle \eta_t - \eta_{t-k} - J(\omega_t) + J(\omega_{t-k}), R_t - J(\omega_t)\rangle\right]|$$
$$\leq |\mathbb{E}\left[\mathbb{E}\left[\langle \eta_{t-k} - J(\omega_{t-k}), R_t - J(\omega_t)\rangle | \mathcal{F}_{t-k}\right]\right]|$$
$$\quad + \mathbb{E}\left[\|\eta_t - \eta_{t-k} - J(\omega_t) + J(\omega_{t-k})\|_2 \|R_t - J(\omega_t)\|_2\right]$$
$$\overset{(a)}{\leq} R_{\max}^2 \mathbb{E}\left[\|\mathbb{P}\left((s_t, a_t)|\mathcal{F}_{t-k}\right), D_t\|_{\mathcal{TV}}\right] + \mathbb{E}\left[\|\eta_t - \eta_{t-k}\|_2 + \|J(\omega_t) - J(\omega_{t-k})\|_2\right] R_{\max}$$
$$\overset{(b)}{\leq} R_{\max}^2 \left(C_\pi \sum_{j=t-k}^{t-1} \mathbb{E}\left[\|\omega_t - \omega_j\|_2\right] + m\rho^k\right)$$
$$\quad + R_{\max} \left(R_{\max} \sum_{j=t-k}^{t-1} \gamma_j + C_J \mathbb{E}\left[\|\omega_t - \omega_{t-k}\|_2\right]\right)$$
$$\leq R_{\max}^2 C_\pi \sum_{j=t-k}^{t-1} \sum_{i=j}^{t-1} \beta_i + R_{\max}^2 m\rho^k + R_{\max}^2 \sum_{j=t-k}^{t-1} \gamma_j + R_{\max} C_J B \sum_{j=t-k}^{t-1} \beta_j, \tag{93}$$

where $(a)$ follows from that $0 \leq \eta_t \leq R_{\max}$ when $\eta_0 = 0$ and $\mathbb{E}_{D_t}\left[R(s, a) - J(\omega_t)\right] = 0$. From $0 \leq J(\omega_t) \leq R$ and $0 \leq R_t \leq R_{\max}$, it holds that $|\langle \eta_{t-k} - J(\omega_{t-k}), R_t - J(\omega_t)\rangle| \leq R_{\max}^2$. $(b)$ follows from Lemma 4.

Term I can be bounded as follows

$$|\mathbb{E}\left[\langle R_t - J(\omega_t), J(\omega_t) - J(\omega_{t+1})\rangle\right]| \leq \mathbb{E}\left[|R_t - J(\omega_t)| |J(\omega_t) - J(\omega_{t+1})|\right]$$
$$\leq R_{\max} C_J \mathbb{E}\left[\|\omega_{t+1} - \omega_t\|_2\right] \leq B C_J R_{\max} \beta_t. \tag{94}$$

We then bound term III as follows

$$|\mathbb{E}\left[\langle \eta_t - J(\omega_t), J(\omega_t) - J(\omega_{t+1})\rangle\right]|$$
$$= \left|\mathbb{E}\left[\left\langle \eta_t - J(\omega_t), -\nabla^\top J(\omega_t)(\omega_{t+1} - \omega_t) + \frac{\nabla^2 J(\hat\omega_t)}{2}(\omega_{t+1} - \omega_t)^2\right\rangle\right]\right|$$
$$\leq \left|\mathbb{E}\left[\langle \eta_t - J(\omega_t), -\nabla^\top J(\omega_t)(\omega_{t+1} - \omega_t)\rangle\right]\right| + \left|\mathbb{E}\left[\left\langle \eta_t - J(\omega_t), \frac{\nabla^2 J(\hat\omega_t)}{2}(\omega_{t+1} - \omega_t)^2\right\rangle\right]\right|$$
$$= \beta_t \left|\mathbb{E}\left[\langle \eta_t - J(\omega_t), -\nabla^\top J(\omega_t)\theta_t\rangle\right]\right| + \beta_t^2 \left|\mathbb{E}\left[\left\langle \eta_t - J(\omega_t), \frac{\nabla^2 J(\hat\omega_t)}{2}\|\theta_t\|_2^2\right\rangle\right]\right|$$
$$\leq \frac{\beta_t}{2}\mathbb{E}\left[\|\eta_t - J(\omega_t)\|_2^2\right] + \frac{B^2\beta_t}{2}\mathbb{E}\left[\|\nabla J(\omega_t)\|_2^2\right] + \frac{R_{\max} L_J B^2}{2}\beta_t^2, \tag{95}$$

where the first equation is from the Mean Value theorem for some $\hat\omega_t = \lambda\omega_t + (1-\lambda)\omega_{t+1}$, $\lambda \in (0, 1)$.

Plug Equation (93), Equation (94) and Equation (95) in Equation (91), and we have that

$$\mathbb{E}\left[\|\eta_{t+1} - J(\omega_{t+1})\|_2^2\right] \leq \left((1 - \gamma_t)^2 + \beta_t\right)\mathbb{E}\left[\|\eta_t - J(\omega_t)\|_2^2\right] + C_\phi^4 B^2 \beta_t \mathbb{E}\left[\|\nabla J(\omega_t)\|_2^2\right]$$

$$+ 2\gamma_t \left( R_{\max}^2 C_\pi \sum_{j=t-k}^{t-1} \sum_{i=j}^{t-1} \beta_j + R_{\max}^2 m\rho^k + R_{\max}^2 \sum_{j=t-k}^{t-1} \gamma_j + R_{\max} C_J B\beta_t \right)$$

$$+ R_{\max}^2 \gamma_t^2 + C_J^2 B^2 \beta_t^2 + 2BC_J R_{\max} \beta_t \gamma_t + 2R_{\max} L_J B^2 \beta_t^2. \tag{96}$$

This completes the proof. $\qquad\square$

### C.3   TRACKING ERROR ANALYSIS OF NAC

In this section, we bound the tracking error $\theta_t - \theta_t^*$ for NAC. Define

$$\widetilde{G}_t^\theta = \left( 8C_{\text{gap}}^2 + \frac{2C_{\text{gap}}^2}{\beta_t} + \beta_t C_\Theta C_{\text{gap}}^2 \right) \left( \frac{m\rho^k}{\lambda_{\min}} \right)^2 + 4(k+1)C_\phi U_\delta \left( 2B\lambda_{\min} + 3C_{\text{gap}} \right) \alpha_t \frac{m\rho^k}{\lambda_{\min}}$$

$$+ 4B^2 U_\delta C_\pi \alpha_t \sum_{j=t-k}^{t-1} \sum_{i=j}^{t-1} \beta_i + 4(k+1)C_\phi U_\delta \left( (k+1)C_\phi U_\delta \alpha_t \sum_{j=t-2k}^{t-1} \alpha_j + C_\Theta B\alpha_t \sum_{j=t-2k}^{t-1} \beta_j \right)$$

$$+ U_\delta^2 (k+1)^2 C_\phi^2 \alpha_t^2 + 8BC_\phi U_\delta \alpha_t \sum_{j=t-k}^{t-1} \left( C_\pi B \sum_{i=j-k}^{j-1} \sum_{\iota=i}^{j-1} \beta_\iota + L_\pi B \sum_{i=j}^{t-1} \beta_i \right)$$

$$+ 2(k+1)C_\phi BU_\delta C_\Theta \alpha_t \beta_t + 2C_\Theta^2 B^2 \beta_t^2, \tag{97}$$

and set

$$\frac{\bar{\lambda}_{\min} \alpha_t}{2} \geq (4C_\Theta + 2)\beta_t. \tag{98}$$

**Lemma 12.** *It holds that*

$$\mathbb{E}\left[ \left\| \theta_{t+1} - \theta_{t+1}^* \right\|_2^2 \right] \leq \left( 1 - \frac{\bar{\lambda}_{\min} \alpha_t}{2} \right) \mathbb{E}\left[ \left\| \theta_t - \theta_t^* \right\|_2^2 \right]$$

$$+ \frac{C_\Theta}{\lambda_{\min}^2} \beta_t \mathbb{E}\left[ \left\| \nabla J(\omega_t) \right\|_2^2 \right] + \frac{(k+1)^2 C_\phi^2}{\bar{\lambda}_{\min}} \alpha_t \mathbb{E}\left[ \left\| J(\omega_t) - \eta_t \right\|_2^2 \right] + \widetilde{G}_t^\theta. \tag{99}$$

*Proof.* From the update rule of Algorithm 1, we have that

$$\begin{aligned}
\left\| \theta_{t+1} - \theta_{t+1}^* \right\|_2^2 &= \left\| \Pi_B \left( \theta_t + \alpha_t \delta_t z_t \right) - \theta_{t+1}^* \right\|_2^2 \\
&\overset{(a)}{\leq} \left\| \theta_t + \alpha_t \delta_t z_t - \theta_{t+1}^* \right\|_2^2 \\
&\leq \left\| \theta_t + \alpha_t \delta_t z_t - \theta_t^* + \theta_t^* - \theta_{t+1}^* \right\|_2^2 \\
&\leq \left\| \theta_t - \theta_t^* \right\|_2^2 + \alpha_t^2 \left\| \delta_t z_t \right\|_2^2 + \left\| \theta_t^* - \theta_{t+1}^* \right\|_2^2 + 2\alpha_t \left\langle \theta_t - \theta_t^*, \delta_t z_t \right\rangle \\
&\quad + 2\alpha_t \left\langle \delta_t z_t, \theta_t^* - \theta_{t+1}^* \right\rangle + 2 \left\langle \theta_t - \theta_t^*, \theta_t^* - \theta_{t+1}^* \right\rangle, \tag{100}
\end{aligned}$$

where $(a)$ follows from the fact $\left\| \Pi_B(x) - y \right\|_2 \leq \left\| x - y \right\|_2$ when $\left\| y \right\|_2 \leq B$ and $\left\| \theta_{t+1}^* \right\|_2 \leq B$.

Taking expectations on both sides, we have that

$$\mathbb{E}\left[ \left\| \theta_{t+1} - \theta_{t+1}^* \right\|_2^2 \right] \leq \mathbb{E}\left[ \left\| \theta_t - \theta_t^* \right\|_2^2 \right] + \alpha_t^2 \mathbb{E}\left[ \left\| \delta_t z_t \right\|_2^2 \right] + \mathbb{E}\left[ \left\| \theta_t^* - \theta_{t+1}^* \right\|_2^2 \right]$$

$$+ 2\alpha_t \underbrace{\mathbb{E}\left[ \left\langle \theta_t - \theta_t^*, \delta_t z_t \right\rangle \right]}_{\text{I}} + 2\alpha_t \underbrace{\mathbb{E}\left[ \left\langle \delta_t z_t, \theta_t^* - \theta_{t+1}^* \right\rangle \right]}_{\text{II}} + 2 \underbrace{\mathbb{E}\left[ \left\langle \theta_t - \theta_t^*, \theta_t^* - \theta_{t+1}^* \right\rangle \right]}_{\text{III}}. \tag{101}$$

For the term $\left\| \theta_t^* - \theta_{t+1}^* \right\|_2$, we have that

$$\left\| \theta_t^* - \theta_{t+1}^* \right\|_2 = \left\| \bar{\theta}_t^* - \bar{\theta}_{t+1}^* + \theta_t^* - \bar{\theta}_t^* - \theta_{t+1}^* + \bar{\theta}_{t+1}^* \right\|_2$$

$$\leq \left\| \bar{\theta}_t^* - \bar{\theta}_{t+1}^* \right\|_2 + \left\| \theta_t^* - \bar{\theta}_t^* \right\|_2 + \left\| \theta_{t+1}^* - \bar{\theta}_{t+1}^* \right\|_2$$

$$\overset{(a)}{\leq} \left\| \bar{\theta}_t^* - \bar{\theta}_{t+1}^* \right\|_2 + \frac{C_{\text{gap}} m \rho^k}{\lambda_{\min}} + \frac{C_{\text{gap}} m \rho^k}{\lambda_{\min}}$$

$$= \left\| \nabla(\hat{\omega}_t) \left( \omega_t - \omega_{t+1} \right) \right\|_2 + \frac{2 C_{\text{gap}} m \rho^k}{\lambda_{\min}}$$

$$\leq \beta_t C_\Theta \left\| \theta_t \right\|_2 + \frac{2 C_{\text{gap}} m \rho^k}{\lambda_{\min}}$$

$$\leq C_\Theta B \beta_t + \frac{2 C_{\text{gap}} m \rho^k}{\lambda_{\min}}, \tag{102}$$

where $(a)$ follows from the Proposition 4. Hence we have that

$$\mathbb{E} \left[ \left\| \theta_t^* - \theta_{t+1}^* \right\|_2^2 \right] \leq 2 C_\Theta^2 B^2 \beta_t^2 + \frac{8 C_{\text{gap}}^2 m^2 \rho^{2k}}{\lambda_{\min}^2}. \tag{103}$$

By Lemma 5, term I in Equation (101) can be bounded as

$$\mathbb{E} \left[ \langle \theta_t - \theta_t^*, \delta_t z_t \rangle \right] \leq - \frac{\bar{\lambda}_{\min}}{2} \mathbb{E} \left[ \left\| \theta_t - \theta_t^* \right\|_2^2 \right] + \frac{(k+1)^2 C_\phi^2}{2 \bar{\lambda}_{\min}} \mathbb{E} \left[ \left\| J(\omega_t) - \eta_t \right\|_2^2 \right] + G_t^\delta. \tag{104}$$

For term II in Equation (101), we have that

$$\begin{aligned}
\mathbb{E} \left[ \langle \delta_t z_t, \theta_t^* - \theta_{t+1}^* \rangle \right] &\leq \mathbb{E} \left[ \left\| \delta_t z_t \right\|_2 \left\| \theta_t^* - \theta_{t+1}^* \right\|_2 \right] \\
&\leq (k+1) C_\phi U_\delta \mathbb{E} \left[ \left\| \bar{\theta}_{t+1}^* - \bar{\theta}_t^* \right\|_2 + \left\| \theta_t^* - \bar{\theta}_t^* \right\|_2 + \left\| \theta_{t+1}^* - \bar{\theta}_{t+1}^* \right\|_2 \right] \\
&\leq (k+1) C_\phi U_\delta C_\Theta \left\| \omega_{t+1} - \omega_t \right\|_2 + \frac{2(k+1) C_\phi U_\delta C_{\text{gap}} m \rho^k}{\lambda_{\min}} \\
&\leq (k+1) C_\phi B U_\delta C_\Theta \beta_t + \frac{2(k+1) C_\phi U_\delta C_{\text{gap}} m \rho^k}{\lambda_{\min}}. \tag{105}
\end{aligned}$$

For term III in Equation (101), we have that

$$\begin{aligned}
& \mathbb{E} \left[ \langle \theta_t - \theta_t^*, \theta_t^* - \theta_{t+1}^* \rangle \right] \\
&= \mathbb{E} \left[ \langle \theta_t - \theta_t^*, \bar{\theta}_{t+1}^* - \bar{\theta}_{t+1}^* \rangle \right] + \mathbb{E} \left[ \langle \theta_t - \theta_t^*, \bar{\theta}_t^* - \theta_t^* \rangle \right] + \mathbb{E} \left[ \langle \theta_t - \theta_t^*, \bar{\theta}_{t+1}^* - \theta_{t+1}^* \rangle \right] \\
&\leq \\
&\leq \mathbb{E} \left[ \left\| \theta_t - \theta_t^* \right\|_2 C_\Theta \left\| \omega_{t+1} - \omega_t \right\|_2 \right] + \beta_t \mathbb{E} \left[ \left\| \theta_t - \theta_t^* \right\|_2^2 \right] + \frac{1}{\beta_t} \left( \frac{C_{\text{gap}} m \rho^k}{\lambda_{\min}} \right)^2 \\
&= \beta_t \mathbb{E} \left[ \left\| \theta_t - \theta_t^* \right\|_2 C_\Theta \left\| \theta_t \right\|_2 \right] + \beta_t \mathbb{E} \left[ \left\| \theta_t - \theta_t^* \right\|_2^2 \right] + \frac{1}{\beta_t} \left( \frac{C_{\text{gap}} m \rho^k}{\lambda_{\min}} \right)^2 \\
&\leq C_\Theta \beta_t \mathbb{E} \left[ \left\| \theta_t - \theta_t^* \right\|_2 \left\| \bar{\theta}_t^* \right\|_2 \right] + \beta_t \mathbb{E} \left[ \left\| \theta_t - \theta_t^* \right\|_2^2 \right] + \frac{1}{\beta_t} \left( \frac{C_{\text{gap}} m \rho^k}{\lambda_{\min}} \right)^2 \\
&\quad + C_\Theta \beta_t \mathbb{E} \left[ \left\| \theta_t - \theta_t^* \right\|_2^2 \right] + C_\Theta \beta_t \mathbb{E} \left[ \left\| \theta_t - \theta_t^* \right\|_2 \left\| \theta_t^* - \bar{\theta}_t^* \right\|_2 \right] \\
&\leq C_\Theta \beta_t \mathbb{E} \left[ \left\| \theta_t - \theta_t^* \right\|_2 \left\| \bar{\theta}_t^* \right\|_2 \right] + \beta_t \mathbb{E} \left[ \left\| \theta_t - \theta_t^* \right\|_2^2 \right] + \frac{1}{\beta_t} \left( \frac{C_{\text{gap}} m \rho^k}{\lambda_{\min}} \right)^2 \\
&\quad + C_\Theta \beta_t \mathbb{E} \left[ \left\| \theta_t - \theta_t^* \right\|_2^2 \right] + C_\Theta \beta_t \mathbb{E} \left[ \left\| \theta_t - \theta_t^* \right\|_2 \left\| \theta_t^* - \bar{\theta}_t^* \right\|_2 \right] \\
&\leq \frac{1}{2} C_\Theta \beta_t \mathbb{E} \left[ \left\| \theta_t - \theta_t^* \right\|_2^2 \right] + \frac{1}{2} C_\Theta \beta_t \mathbb{E} \left[ \left\| \left( \mathbb{E}_{D_t} \left[ \phi_t^\top(s, a) \phi_t(s, a) \right] \right)^{-1} \nabla J(\omega_t) \right\|_2^2 \right] \\
&\quad + \beta_t \mathbb{E} \left[ \left\| \theta_t - \theta_t^* \right\|_2^2 \right] + \frac{1}{\beta_t} \left( \frac{C_{\text{gap}} m \rho^k}{\lambda_{\min}} \right)^2 + C_\Theta \beta_t \mathbb{E} \left[ \left\| \theta_t - \theta_t^* \right\|_2^2 \right]
\end{aligned}$$

$$+ \frac{C_\Theta}{2} \beta_t \mathbb{E}\left[\|\theta_t - \theta_t^*\|_2^2\right] + \frac{C_\Theta}{2} \left(\frac{C_{\text{gap}} m\rho^k}{\lambda_{\min}}\right)^2 \beta_t$$

$$\leq (2C_\Theta + 1)\beta_t \mathbb{E}\left[\|\theta_t - \theta_t^*\|_2^2\right] + \frac{C_\Theta}{2\lambda_{\min}^2}\beta_t \mathbb{E}\left[\|\nabla J(\omega_t)\|_2^2\right] + \left(\frac{\beta_t C_\Theta}{2} + \frac{1}{\beta_t}\right)\left(\frac{C_{\text{gap}} m\rho^k}{\lambda_{\min}}\right)^2.$$
$$(106)$$

Combine the bounds on terms I, II, III in Equation (101) together, we have that

$$\mathbb{E}\left[\|\theta_{t+1} - \theta_{t+1}^*\|_2^2\right]$$

$$\leq \left(1 - \bar{\lambda}_{\min}\alpha_t + (4C_\Theta + 2)\beta_t\right)\mathbb{E}\left[\|\theta_t - \theta_t^*\|_2^2\right] + \frac{C_\Theta}{\lambda_{\min}^2}\beta_t\mathbb{E}\left[\|\nabla J(\omega_t)\|_2^2\right]$$

$$+ \frac{(k+1)^2 C_\phi^2}{\bar{\lambda}_{\min}}\alpha_t \mathbb{E}\left[\|J(\omega_t) - \eta_t\|_2^2\right] + \left(8C_{\text{gap}}^2 + \frac{2C_{\text{gap}}^2}{\beta_t} + \beta_t C_\Theta C_{\text{gap}}^2\right)\left(\frac{m\rho^k}{\lambda_{\min}}\right)^2$$

$$+ 4B^2 U_\delta C_\pi \alpha_t \sum_{j=t-k}^{t-1}\sum_{i=j}^{t-1}\beta_i + 4(k+1)C_\phi U_\delta \left((k+1)C_\phi U_\delta \alpha_t \sum_{j=t-2k}^{t-1}\alpha_j + C_\Theta B\alpha_t \sum_{j=t-2k}^{t-1}\beta_j\right)$$

$$+ U_\delta^2(k+1)^2 C_\phi^2 \alpha_t^2 + 8BC_\phi U_\delta \alpha_t \sum_{j=t-k}^{t-1}\left(C_\pi B \sum_{i=j-k}^{j-1}\sum_{\iota=i}^{j-1}\beta_\iota + L_\pi B \sum_{i=j}^{t-1}\beta_i\right)$$

$$+ 2(k+1)C_\phi BU_\delta C_\Theta \alpha_t \beta_t + 2C_\Theta^2 B^2 \beta_t^2. \tag{107}$$

This hence completes the proof. $\qquad \square$

## C.4 SAMPLE COMPLEXITY OF NAC

**Lemma 13.** *Denote* $\widetilde{T} = \left\lceil \frac{T}{\hat{t}\log T}\right\rceil \hat{t} \geq \frac{T}{\log T}$*, then for any* $t' \leq T - \widetilde{T}$*, it holds that*

$$\min_{t \leq T}\mathbb{E}\left[J(\pi^*) - J(\omega_t)\right]$$

$$\leq \frac{D(\omega_{t'}) - D(\omega_{t'+\widetilde{T}-1})}{\beta\widetilde{T}} + \frac{C_\phi C_{gap} m\rho^k}{\lambda_{\min}} + 4C_\infty\sqrt{\varepsilon_{actor}} + C_\phi\sqrt{2}\left(\frac{e\lambda_m C^M C_\phi \hat{t}}{\widetilde{T}}\right)$$

$$+ C_\phi\sqrt{2}\sqrt{\frac{1}{\widetilde{T}}\sum_{t=t'}^{t'+\widetilde{T}-1}\sum_{j=t-\hat{t}}^{t-1}(1-q)^{t-j}\left(\widetilde{G}_j^\omega + \widetilde{G}_j^\theta + \widetilde{G}_j^\eta\right)} + C_\phi\sqrt{\frac{8B^2 + 2R_{\max}^2}{T}} + \frac{B^2 L_\phi}{2}\beta$$

$$+ C_\phi\sqrt{2\lambda_m C^M \frac{e\hat{t}}{\widetilde{T}}\frac{D(\omega_{t'}) - D(\omega_{t'+\widetilde{T}-1})}{\beta\widetilde{T}}} + C_\phi\sqrt{\frac{2e\lambda_m\hat{t}}{\widetilde{T}^2}\sum_{t=t'}^{t'+\widetilde{T}-1}\widetilde{G}_t^\omega} + \frac{R_{\max}}{T}. \tag{108}$$

*Proof.* Recall that $\pi^* = \arg\max_\pi J(\pi)$. Denote by $A^{\pi_t} = Q^{\pi_t}(s,a) - V^{\pi_t}(s)$ the relative advantage function, and also denote $D(\omega_t) = -\mathbb{E}_{D_{\pi^*}}\left[\log\frac{\pi_{\omega_t}(a|s)}{\pi^*(a|s)}\right]$. We first have that

$$D(\omega_t) - D(\omega_{t+1}) = \mathbb{E}_{D_{\pi^*}}\left[\log\pi_{t+1}(a|s) - \log\pi_t(a|s)\right]$$

$$\geq \mathbb{E}_{D_{\pi^*}}\left[\nabla\log\pi_t(a|s)\right](\omega_{t+1} - \omega_t) - \frac{L_\phi}{2}\|\omega_{t+1} - \omega_t\|_2^2$$

$$= \beta_t\mathbb{E}_{D_{\pi^*}}\left[\phi_t^\top(s,a)\theta_t\right] - \frac{L_\phi\beta_t^2}{2}\|\theta_t\|_2^2$$

$$\geq \beta_t\mathbb{E}_{D_{\pi^*}}\left[A^{\pi_t}(s,a)\right] + \beta_t\mathbb{E}_{D_{\pi^*}}\left[\phi_t^\top(s,a)\theta_t^* - A^{\pi_t}(s,a)\right] + \beta_t\mathbb{E}_{D_{\pi^*}}\left[\phi_t^\top(s,a)(\theta_t - \theta_t^*)\right]$$

$$- \frac{B^2 L_\phi\beta_t^2}{2}$$

$$\overset{(a)}{=} \beta_t \left( J(\pi^*) - J(\pi_t) \right) + \beta_t \underbrace{\mathbb{E}_{D_{\pi^*}} \left[ \phi_t^\top(s,a)\theta_t^* - A^{\pi_t}(s,a) \right]}_{\text{I}} + \beta_t \underbrace{\mathbb{E}_{D_{\pi^*}} \left[ \phi_t^\top(s,a)(\theta_t - \theta_t^*) \right]}_{\text{II}}$$

$$- \frac{B^2 L_\phi \beta_t^2}{2}, \tag{109}$$

where $(a)$ from the fact that $\mathbb{E}_{D_{\pi^*}} \left[ A^{\pi_t}(s,a) \right] = J(\pi^*) - J(\pi_t)$.

To bound Term I, note that

$$\left| \mathbb{E}_{D_{\pi^*}} \left[ \phi_t^\top(s,a)\theta_t^* - A^{\pi_t}(s,a) \right] \right|$$

$$\leq \left| \mathbb{E}_{D_{\pi^*}} \left[ \phi_t^\top(s,a)\bar{\theta}_t^* - A^{\pi_t}(s,a) \right] \right| + \left| \mathbb{E}_{D_{\pi^*}} \left[ \phi_t^\top(s,a)(\bar{\theta}_t^* - \theta_t^*) \right] \right|$$

$$\leq \left\| \frac{D_{\pi^*}}{D_t} \right\|_\infty \left| \mathbb{E}_{D_t} \left[ \phi_t^\top(s,a)\bar{\theta}_t^* - \sum_a \pi_t(a|s)\phi_t^\top(s,a)\bar{\theta}_t^* - A^{\pi_t}(s,a) \right] \right| + C_\phi \left\| \bar{\theta}_t^* - \theta_t^* \right\|_2$$

$$\leq \left\| \frac{D_{\pi^*}}{D_t} \right\|_\infty \left| \mathbb{E}_{D_t} \left[ \phi_t^\top(s,a)\bar{\theta}_t^* - \sum_a \pi_t(a|s)\phi_t^\top(s,a)\bar{\theta}_t^* - Q^{\pi_t}(s,a) + V^{\pi_t}(s) \right] \right| + \frac{C_\phi C_{\text{gap}} m \rho^k}{\lambda_{\min}}$$

$$\leq \left\| \frac{D_{\pi^*}}{D_t} \right\|_\infty \sqrt{\mathbb{E}_{D_t} \left[ \left\| \phi_t^\top(s,a)\bar{\theta}_t^* - Q^{\pi_t}(s,a) \right\|_2^2 \right]} + \frac{C_\phi C_{\text{gap}} m \rho^k}{\lambda_{\min}}$$

$$+ \left\| \frac{D_{\pi^*}}{D_t} \right\|_\infty \sqrt{\mathbb{E}_{D_t} \left[ \left\| \sum_a \pi_t(a|s)\phi_t^\top(s,a)\bar{\theta}_t^* - \sum_a \pi_t(a|s)\phi_t^\top(s,a)Q^{\pi_t}(s,a) \right\|_2^2 \right]}$$

$$\overset{(a)}{\leq} \left\| \frac{D_{\pi^*}}{D_t} \right\|_\infty 2\sqrt{\varepsilon_{\text{actor}}} + \frac{C_\phi C_{\text{gap}} m \rho^k}{\lambda_{\min}}, \tag{110}$$

where $(a)$ follows from Equation (2), the definition of $\bar{\theta}_t^*$, the definition of $\varepsilon_{\text{actor}}$ and the facts that

$$\mathbb{E}_{D_t} \left[ \left\| Q^{\pi_t}(s,a) - \phi_t^\top(s,a)\bar{\theta}_t^* \right\|_2^2 \right] \leq \varepsilon_{\text{actor}}, \tag{111}$$

and

$$\mathbb{E}_{D_t} \left[ \left\| \sum_a \pi_t(a|s)\phi_t^\top(s,a)\bar{\theta}_t^* - \sum_a \pi_t(a|s)\phi_t^\top(s,a)Q^{\pi_t}(s,a) \right\|_2^2 \right]$$

$$\leq \mathbb{E}_{D_t} \left[ \left\| Q^{\pi_t}(s,a) - \phi_t^\top(s,a)\bar{\theta}_t^* \right\|_2^2 \right] \leq \varepsilon_{\text{actor}}. \tag{112}$$

Next, consider term II. We have that

$$\left| \mathbb{E}_{D_{\pi^*}} \left[ \phi_t^\top(s,a)(\theta_t - \theta_t^*) \right] \right| \leq C_\phi \left\| \theta_t - \theta_t^* \right\|_2. \tag{113}$$

Plug the two bounds on terms I and II in Equation (109), and we have that

$$\mathbb{E} \left[ D(\omega_t) \right] - \mathbb{E} \left[ D(\omega_{t+1}) \right] \geq \beta_t \left( J(\pi^*) - \mathbb{E} \left[ J(\omega_t) \right] \right) - \beta_t \left\| \frac{D_{\pi^*}}{D_t} \right\|_\infty 2\sqrt{\varepsilon_{\text{actor}}} - \beta_t \frac{C_\phi C_{\text{gap}} m \rho^k}{\lambda_{\min}}$$

$$- \beta_t C_\phi \mathbb{E} \left[ \left\| \theta_t - \theta_t^* \right\|_2 \right] - \frac{L_\phi}{2} \beta_t^2 B^2, \tag{114}$$

which implies

$$\beta \left( J(\pi^*) - \mathbb{E} \left[ J(\omega_t) \right] \right) \leq \mathbb{E} \left[ D(\omega_t) \right] - \mathbb{E} \left[ D(\omega_{t+1}) \right] + \beta \left\| \frac{D_{\pi^*}}{D_t} \right\|_\infty 2\sqrt{\varepsilon_{\text{actor}}} + \beta \frac{C_\phi C_{\text{gap}} m \rho^k}{\lambda_{\min}}$$

$$+ \beta C_\phi \mathbb{E} \left[ \left\| \theta_t - \theta_t^* \right\|_2 \right] + \frac{B^2 L_\phi}{2} \beta^2. \tag{115}$$

We set $M_{t+1} = \mathbb{E}\left[\left\|\theta_{t+1} - \theta_{t+1}^*\right\|_2^2\right] + \mathbb{E}\left[\left\|\eta_{t+1} - J(\omega_{t+1})\right\|_2^2\right]$, we now aim to bound $M_t$. Combine the bounds we obtained in Equation (90) and Equation (99), and we have that

$$M_{t+1} \leq \left(1 - \frac{1}{2}\bar{\lambda}_{\min}\alpha\right)\mathbb{E}\left[\|\theta_t - \theta_t^*\|_2^2\right] + \frac{C_\Theta}{\lambda_{\min}^2}\beta\mathbb{E}\left[\|\nabla J(\omega_t)\|_2^2\right] + \frac{(k+1)^2 C_\phi^2}{\bar{\lambda}_{\min}}\alpha\mathbb{E}\left[\|J(\omega_t) - \eta_t\|_2^2\right]$$

$$+ (1-\gamma)\mathbb{E}\left[\|\eta_t - J(\omega_t)\|_2^2\right] + \frac{B^2}{2}\beta\mathbb{E}\left[\|\nabla J(\omega_t)\|_2^2\right] + \widetilde{G}_t^\theta + \widetilde{G}_t^\eta$$

$$= \left(1 - \frac{1}{2}\bar{\lambda}_{\min}\alpha\right)\mathbb{E}\left[\|\theta_t - \theta_t^*\|_2^2\right] + \frac{(k+1)^2 C_\phi^2}{\bar{\lambda}_{\min}}\alpha\mathbb{E}\left[\|J(\omega_t) - \eta_t\|_2^2\right] + (1-\gamma)\mathbb{E}\left[\|\eta_t - J(\omega_t)\|_2^2\right]$$

$$+ \left(\frac{C_\Theta}{\lambda_{\min}^2} + \frac{B^2}{2}\right)\beta\mathbb{E}\left[\|\nabla J(\omega_t)\|_2^2\right] + \widetilde{G}_t^\theta + \widetilde{G}_t^\eta$$

$$\overset{(a)}{\leq} \left(1 - \frac{1}{2}\bar{\lambda}_{\min}\alpha\right)\mathbb{E}\left[\|\theta_t - \theta_t^*\|_2^2\right] + \frac{(k+1)^2 C_\phi^2}{\bar{\lambda}_{\min}}\alpha\mathbb{E}\left[\|J(\omega_t) - \eta_t\|_2^2\right] + (1-\gamma)\mathbb{E}\left[\|\eta_t - J(\omega_t)\|_2^2\right]$$

$$+ \left(\frac{C_\Theta}{\lambda_{\min}^2} + \frac{B^2}{2}\right)\beta\left(\frac{\lambda_{\mathrm{m}}}{\beta}\left(\mathbb{E}\left[J(\omega_{t+1})\right] - \mathbb{E}\left[J(\omega_t)\right]\right) + 2\lambda_{\mathrm{m}}^2\mathbb{E}\left[\|\theta_t - \theta_t^*\|_2^2\right]\right)$$

$$+ \left(\frac{C_\Theta}{\lambda_{\min}^2} + \frac{B^2}{2}\right)\beta\left(C_{\mathrm{gap}}\frac{\lambda_{\mathrm{m}} m\rho^k}{\lambda_{\min}} + \frac{L_J B^2 \beta^2}{2}\right) + \widetilde{G}_t^\theta + \widetilde{G}_t^\eta$$

$$\leq \left(1 - \frac{1}{2}\bar{\lambda}_{\min}\alpha + 2\lambda_{\mathrm{m}}^2 C^M \beta\right)\mathbb{E}\left[\|\theta_t - \theta_t^*\|_2^2\right] + \left(\lambda_{\mathrm{m}} C^M\right)\left(\mathbb{E}\left[J(\omega_{t+1})\right] - \mathbb{E}\left[J(\omega_t)\right]\right)$$

$$+ \left(1 - \gamma + \frac{(k+1)^2 C_\phi^2}{\bar{\lambda}_{\min}}\alpha\right)\mathbb{E}\left[\|\eta_t - J(\omega_t)\|_2^2\right] + C^M\left(\frac{C_{\mathrm{gap}}\lambda_{\mathrm{m}} m\rho^k \beta}{\lambda_{\min}} + \frac{L_J B^2 \beta^2}{2}\right)$$

$$+ \widetilde{G}_t^\theta + \widetilde{G}_t^\eta, \tag{116}$$

where $(a)$ is obtained by plugging Equation (88), and $C^M = \frac{C_\Theta}{\lambda_{\min}^2} + \frac{B^2}{2}$. To convenience, we set

$$\widetilde{G}_t^\omega = C^M\left(\frac{C_{\mathrm{gap}}\lambda_{\mathrm{m}} m\rho^k \beta}{\lambda_{\min}} + \frac{L_J B^2 \beta^2}{2}\right). \tag{117}$$

We set $k$ large enough such that $\bar{\lambda}_{\min} \geq \frac{1}{2}\lambda_{\min}$, and further set the step sizes such that $\frac{1}{6}\bar{\lambda}_{\min}\alpha \geq 2\lambda_{\mathrm{m}}^2 C^M \beta$ and $\frac{\gamma}{2} \geq \frac{(k+1)^2 C_\phi^2}{\bar{\lambda}_{\min}}\alpha$. Denote $q = \frac{1}{3}\bar{\lambda}_{\min}\alpha$. Then the inequality above can be written as

$$M_{t+1} \leq (1-q)M_t + \left(\lambda_{\mathrm{m}} C^M\right)\left(\mathbb{E}\left[J(\omega_{t+1})\right] - \mathbb{E}\left[J(\omega_t)\right]\right)$$
$$+ \widetilde{G}_t^\omega + \widetilde{G}_t^\theta + \widetilde{G}_t^\eta. \tag{118}$$

Set $\hat{t} = \left\lceil\frac{1}{q}\log T\right\rceil$. For $t \geq 2k + \hat{t}$, we recursively apply Equation (118) for $\hat{t}$ times, and have that

$$M_t \leq (1-q)^{\hat{t}} M_{t-\hat{t}} + \lambda_{\mathrm{m}} C^M \sum_{j=t-\hat{t}}^{t-1}(1-q)^{t-j}\left(\mathbb{E}\left[J(\omega_{j+1})\right] - \mathbb{E}\left[J(\omega_j)\right]\right)$$

$$+ \sum_{j=t-\hat{t}}^{t-1}(1-q)^{t-j}\left(\widetilde{G}_j^\omega + \widetilde{G}_j^\theta + \widetilde{G}_j^\eta\right)$$

$$\overset{(a)}{\leq} \frac{4B^2 + 4R_{\max}^2}{T} + \lambda_{\mathrm{m}} C^M \sum_{j=t-\hat{t}}^{t-1}(1-q)^{t-j}\left(\mathbb{E}\left[J(\omega_{j+1})\right] - \mathbb{E}\left[J(\omega_j)\right]\right)$$

$$+ \sum_{j=t-\hat{t}}^{t-1}(1-q)^{t-j}\left(\widetilde{G}_j^\omega + \widetilde{G}_j^\theta + \widetilde{G}_j^\eta\right), \tag{119}$$

where $(a)$ follows from $(1-q)^{\hat{t}} \leq e^{-q\hat{t}} \leq e^{\log T} \leq T$.

Denote the time length $\widetilde{T} = \left\lceil \frac{T}{\hat{t}\log T} \right\rceil \hat{t} \geq \frac{T}{\log T}$. For any $t' \leq T - \widetilde{T}$, together with Equation (115) we have that

$$
\frac{1}{\widetilde{T}} \sum_{t=t'}^{t'+\widetilde{T}-1} J(\pi^*) - \mathbb{E}\left[J(\omega_t)\right]
$$

$$
\leq \frac{D(\omega_{t'}) - D(\omega_{t'+\widetilde{T}-1})}{\beta\widetilde{T}} + \frac{C_\phi C_{\text{gap}} m \rho^k}{\lambda_{\min}} + \frac{1}{\widetilde{T}} \sum_{t=t'}^{t'+\widetilde{T}-1} \left\| \frac{D_{\pi^*}}{D_t} \right\|_\infty 2\sqrt{\varepsilon_{\text{actor}}}
$$

$$
+ C_\phi \frac{1}{\widetilde{T}} \sum_{t=t'}^{t'+\widetilde{T}-1} \mathbb{E}\left[\|\theta_t - \theta_t^*\|_2\right] + \frac{B^2 L_\phi}{2}\beta^2
$$

$$
\overset{(a)}{\leq} \frac{D(\omega_{t'}) - D(\omega_{t'+\widetilde{T}-1})}{\beta\widetilde{T}} + \frac{C_\phi C_{\text{gap}} m \rho^k}{\lambda_{\min}} + \frac{1}{\widetilde{T}} \sum_{t=t'}^{t'+\widetilde{T}-1} \left\| \frac{D_{\pi^*}}{D_t} \right\|_\infty 2\sqrt{\varepsilon_{\text{actor}}}
$$

$$
+ C_\phi \sqrt{\frac{1}{\widetilde{T}} \sum_{t=t'}^{t'+\widetilde{T}-1} \mathbb{E}\left[\|\theta_t - \theta_t^*\|_2^2\right]} + \frac{B^2 L_\phi}{2}\beta^2
$$

$$
\overset{(b)}{\leq} \frac{D(\omega_{t'}) - D(\omega_{t'+\widetilde{T}-1})}{\beta\widetilde{T}} + \frac{C_\phi C_{\text{gap}} m \rho^k}{\lambda_{\min}} + 2C_\infty \sqrt{\varepsilon_{\text{actor}}}
$$

$$
+ C_\phi \sqrt{\frac{1}{\widetilde{T}} \sum_{t=t'}^{t'+\widetilde{T}-1} M_t} + \frac{B^2 L_\phi}{2}\beta^2, \tag{120}
$$

where $(a)$ follows from the rearrangement inequality and the fact for any random variable $X$, $\|\mathbb{E}\left[X\right]\|_2^2 \leq \mathbb{E}\left[\|X\|_2^2\right]$. $(b)$ follows from Assumption 3.

Moreover, for $2k + \hat{t} \leq t' \leq T - \widetilde{T}$, summing Equation (119) w.r.t. $t$ from $t'$ to $t' + \tilde{T} - 1$ implies

$$
\frac{1}{\widetilde{T}} \sum_{t=t'}^{t'+\widetilde{T}-1} M_t
$$

$$
\leq \lambda_{\text{m}} C^M \frac{1}{\widetilde{T}} \sum_{t=t'}^{t'+\widetilde{T}-1} \sum_{j=t-\hat{t}}^{t-1} (1-q)^{t-j} \left(\mathbb{E}\left[J(\omega_{j+1})\right] - \mathbb{E}\left[J(\omega_j)\right]\right)
$$

$$
+ \frac{4B^2 + R_{\max}^2}{\widetilde{T}} + \frac{1}{\widetilde{T}} \sum_{t=t'}^{t'+\widetilde{T}-1} \sum_{j=t-\hat{t}}^{t-1} (1-q)^{t-j} \left(\widetilde{G}_j^\omega + \widetilde{G}_j^\theta + \widetilde{G}_j^\eta\right)
$$

$$
\overset{(a)}{=} \lambda_{\text{m}} C^M \frac{1}{\widetilde{T}} \sum_{i=0}^{\hat{t}-1} \sum_{t=\hat{t}-i}^{t'+\widetilde{T}-1} (1-q)^{i+1} \left(\mathbb{E}\left[J(\omega_{t-i+1})\right] - \mathbb{E}\left[J(\omega_{t-i})\right]\right)
$$

$$
+ \frac{4B^2 + R_{\max}^2}{\widetilde{T}} + \frac{1}{\widetilde{T}} \sum_{t=t'}^{t'+\widetilde{T}-1} \sum_{j=t-\hat{t}}^{t-1} (1-q)^{t-j} \left(\widetilde{G}_j^\omega + \widetilde{G}_j^\theta + \widetilde{G}_j^\eta\right)
$$

$$
\overset{(b)}{=} \lambda_{\text{m}} C^M \frac{1}{\widetilde{T}} \sum_{i=0}^{\hat{t}-1} (1-q)^{i+1} \left(\mathbb{E}\left[J(\omega_{t'+\tilde{T}})\right] - \mathbb{E}\left[J(\omega_{t'-i})\right]\right)
$$

$$
+ \frac{4B^2 + R_{\max}^2}{\widetilde{T}} + \frac{1}{\widetilde{T}} \sum_{t=t'}^{t'+\widetilde{T}-1} \sum_{j=t-\hat{t}}^{t-1} (1-q)^{t-j} \left(\widetilde{G}_j^\omega + \widetilde{G}_j^\theta + \widetilde{G}_j^\eta\right)
$$

$$
\leq \lambda_{\text{m}} C^M \frac{1}{\widetilde{T}} \sum_{i=0}^{\hat{t}} (1-q)^{i+1} \left(J(\pi^*) - \mathbb{E}\left[J(\omega_{t'-i})\right]\right)
$$

$$+ \frac{4B^2 + R_{\max}^2}{\widetilde{T}} + \frac{1}{q} \left( C^M \left( C_{\text{gap}} \frac{\lambda_{\text{m}} m \rho^k}{\lambda_{\min}} \beta + L_J B^2 \beta^2 \right) + \widetilde{G}^\theta + \widetilde{G}^\eta \right)$$

$$\leq \lambda_{\text{m}} C^M \frac{1}{\widetilde{T}} \sum_{i=0}^{\hat{t}} \left( J(\pi^*) - \mathbb{E}\left[ J(\omega_{t'-i}) \right] \right)$$

$$+ \frac{4B^2 + R_{\max}^2}{\widetilde{T}} + \frac{1}{\widetilde{T}} \sum_{t=t'}^{t'+\widetilde{T}-1} \sum_{j=t-\hat{t}}^{t-1} (1-q)^{t-j} \left( \widetilde{G}_j^\omega + \widetilde{G}_j^\theta + \widetilde{G}_j^\eta \right), \tag{121}$$

where $(a)$ follows from that we set $i = t - j$ and $(b)$ follows from the fact that $\sum_{t=\hat{t}-i}^{t'+\widetilde{T}-1} \left( \mathbb{E}\left[ J(\omega_{t-i+1}) \right] - \mathbb{E}\left[ J(\omega_{t-i}) \right] \right) = \mathbb{E}\left[ J(\omega_{t'+\widetilde{T}}) \right] - \mathbb{E}\left[ J(\omega_{t'-i}) \right]$.

We denote by $X_{t'} = \frac{1}{\hat{t}} \sum_{t=t'-\hat{t}}^{t'-1} \left( J(\pi^*) - \mathbb{E}\left[ J(\omega_t) \right] \right)$, $Y_{t'} = \frac{1}{\widetilde{T}} \sum_{t=t'}^{t'+\widetilde{T}-1} \left( J(\pi^*) - \mathbb{E}\left[ J(\omega_t) \right] \right)$ and $Z_{t'} = \frac{1}{\widetilde{T}} \sum_{t=t'}^{t'+\widetilde{T}-1} M_t$.

Firstly note that there must exist $t' + \hat{t} \leq t'' \leq t' + \widetilde{T}$ s.t. $X_{t''} \leq Y_{t'}$. We then discuss the following two cases.

[Case 1: For any $2k + \hat{t} \leq t' < T - \widetilde{T}$, it holds that $e Y_{t'} \leq X_{t'}$.]

Then, for $\widetilde{t}_0 = 2k + \hat{t}$, we have $X_{\widetilde{t}_0} > e Y_{\widetilde{t}_0}$.

Thus, there must exist $\widetilde{t}_0 + \hat{t} \leq \widetilde{t}_1 \leq \widetilde{t}_0 + T - \hat{t}$, s.t. $X_{\widetilde{t}_1} \leq Y_{\widetilde{t}_0} < \frac{1}{e} X_{\widetilde{t}_0}$; recursively applying this inequality for $j = 0, 1, ..., \lfloor \log T \rfloor$ implies that

$$X_{\widetilde{t}_0} \overset{(a)}{\geq} e Y_{\widetilde{t}_0} \overset{(b)}{\geq} e X_{\widetilde{t}_1} \geq e^2 Y_{\widetilde{t}_1} \geq ... \geq e^j Y_{\widetilde{t}_j} \geq e^{j+1} Y_{\widetilde{t}_j} \geq ... e^{\lfloor \log T \rfloor} X_{\widetilde{t}_{\lfloor \log T \rfloor}} \geq e^{\lfloor \log T \rfloor + 1} Y_{\widetilde{t}_{\lfloor \log T \rfloor}}, \tag{122}$$

where $(a)$ follows from the condition of case 1 and $(b)$ follows from the fact that there must exists $t'' \leq t' + \widetilde{T}$ i.e. $X_{t''} \leq Y_{t'}$. Then, by Equation (122), we can conclude that

$$Y_{\widetilde{t}_{\lfloor \log T \rfloor}} := \frac{1}{\widetilde{T}} \sum_{t=t'}^{t'+\widetilde{T}-1} \left( J(\pi^*) - \mathbb{E}\left[ J(\omega_t) \right] \right) \leq \frac{1}{e^{\lfloor \log T \rfloor + 1}} X_{\widetilde{t}_0} \leq \frac{1}{T} X_{\widetilde{t}_0}. \tag{123}$$

Note that $X_{\widetilde{t}} \leq J(\pi^*) \leq R_{\max}$, hence we have that

$$Y_{\widetilde{t}_{\lfloor \log T \rfloor}} = \frac{1}{\widetilde{T}} \sum_{t=t'}^{t'+\widetilde{T}-1} \left( J(\pi^*) - \mathbb{E}\left[ J(\omega_t) \right] \right) \leq \frac{R_{\max}}{T}. \tag{124}$$

This further implies that

$$\min_{t<T} \mathbb{E}\left[ J(\pi^*) - J(\omega_t) \right] \leq \frac{R_{\max}}{T}. \tag{125}$$

This hence completes the proof of Theorem 2 under Case 1.

[Case 2 There exists some $2k + \hat{t} \leq \widetilde{t}' \leq T - \widetilde{T}$ s.t. $X_{\widetilde{t}} \leq e Y_{\widetilde{t}}$.]

From Equation (121), we obtain that

$$\frac{1}{\widetilde{T}} \sum_{t=t'}^{t'+\widetilde{T}-1} M_t \leq \lambda_{\text{m}} C^M \frac{1}{\widetilde{T}} \sum_{i=0}^{\hat{t}} \left( J(\pi^*) - \mathbb{E}\left[ J(\omega_{t'-i}) \right] \right)$$

$$+ \frac{4B^2 + R_{\max}^2}{\widetilde{T}} + \frac{1}{\widetilde{T}} \sum_{t=t'}^{t'+\widetilde{T}-1} \sum_{j=t-\hat{t}}^{t-1} (1-q)^{t-j} \left( \widetilde{G}_j^\omega + \widetilde{G}_j^\theta + \widetilde{G}_j^\eta \right)$$

$$\leq \lambda_{\mathrm{m}} C^M \frac{\hat{t}}{\widetilde{T}} X_{t'} + \frac{4B^2 + R_{\max}^2}{\widetilde{T}} + \frac{1}{\widetilde{T}} \sum_{t=t'}^{t'+\widetilde{T}-1} \sum_{j=t-\hat{t}}^{t-1} (1-q)^{t-j} \left( \widetilde{G}_j^\omega + \widetilde{G}_j^\theta + \widetilde{G}_j^\eta \right)$$

$$\leq \lambda_{\mathrm{m}} C^M \frac{e\hat{t}}{\widetilde{T}} Y_{t'} + \frac{4B^2 + R_{\max}^2}{\widetilde{T}} + \frac{1}{\widetilde{T}} \sum_{t=t'}^{t'+\widetilde{T}-1} \sum_{j=t-\hat{t}}^{t-1} (1-q)^{t-j} \left( \widetilde{G}_j^\omega + \widetilde{G}_j^\theta + \widetilde{G}_j^\eta \right). \tag{126}$$

Next, from Equation (120), we have that

$$Y_{t'} = \frac{1}{\widetilde{T}} \sum_{t=t'}^{t'+\widetilde{T}-1} J(\pi^*) - \mathbb{E}\left[ J(\omega_t) \right]$$

$$\leq \frac{D(\omega_{t'}) - D(\omega_{t'+\widetilde{T}-1})}{\beta \widetilde{T}} + \frac{C_\phi C_{\mathrm{gap}} m \rho^k}{\lambda_{\min}} + 2C_\infty \sqrt{\varepsilon_{\mathrm{actor}}}$$

$$+ C_\phi \frac{1}{\widetilde{T}} \sum_{t=t'}^{t'+\widetilde{T}-1} \mathbb{E}\left[ \|\theta_t - \theta_t^*\|_2 \right] + \frac{B^2 L_\phi}{2} \beta^2$$

$$\leq \frac{D(\omega_{t'}) - D(\omega_{t'+\widetilde{T}-1})}{\beta \widetilde{T}} + \frac{C_\phi C_{\mathrm{gap}} m \rho^k}{\lambda_{\min}} + 2C_\infty \sqrt{\varepsilon_{\mathrm{actor}}}$$

$$+ C_\phi \sqrt{\frac{1}{\widetilde{T}} \sum_{t=t'}^{t'+\widetilde{T}-1} \mathbb{E}\left[ \|\theta_t - \theta_t^*\|_2 \right]} + \frac{B^2 L_\phi}{2} \beta^2$$

$$\leq \frac{D(\omega_{t'}) - D(\omega_{t'+\widetilde{T}-1})}{\beta \widetilde{T}} + \frac{C_\phi C_{\mathrm{gap}} m \rho^k}{\lambda_{\min}} + 2C_\infty \sqrt{\varepsilon_{\mathrm{actor}}}$$

$$+ C_\phi \sqrt{\frac{1}{\widetilde{T}} \sum_{t=t'}^{t'+\widetilde{T}-1} M_t} + \frac{B^2 L_\phi}{2} \beta^2. \tag{127}$$

Then, we have that

$$\frac{1}{\widetilde{T}} \sum_{t=t'}^{t'+\widetilde{T}-1} M_t$$

$$\leq \frac{4B^2 + R_{\max}^2}{\widetilde{T}} + \frac{1}{\widetilde{T}} \sum_{t=t'}^{t'+\widetilde{T}-1} \sum_{j=t-\hat{t}}^{t-1} (1-q)^{t-j} \left( \widetilde{G}_j^\omega + \widetilde{G}_j^\theta + \widetilde{G}_j^\eta \right) + \frac{e\lambda_{\mathrm{m}} \hat{t}}{\widetilde{T}^2} \sum_{t=t'}^{t'+\widetilde{T}-1} \widetilde{G}_t^\omega$$

$$+ \lambda_{\mathrm{m}} C^M \frac{e\hat{t}}{\widetilde{T}} \left( \frac{D(\omega_{t'}) - D(\omega_{t'+\widetilde{T}-1})}{\beta \widetilde{T}} + 2C_\infty \sqrt{\varepsilon_{\mathrm{actor}}} + C_\phi \sqrt{\frac{1}{\widetilde{T}} \sum_{t=t'}^{t'+\widetilde{T}-1} M_t} \right)$$

$$\leq \frac{1}{2} \frac{1}{\widetilde{T}} \sum_{t=t'}^{t'+\widetilde{T}-1} M_t + \left( \frac{e\lambda_{\mathrm{m}} C^M C_\phi \hat{t}}{\widetilde{T}} \right)^2 + \frac{1}{\widetilde{T}} \sum_{t=t'}^{t'+\widetilde{T}-1} \sum_{j=t-\hat{t}}^{t-1} (1-q)^{t-j} \left( \widetilde{G}_j^\omega + \widetilde{G}_j^\theta + \widetilde{G}_j^\eta \right)$$

$$+ \frac{4B^2 + R_{\max}^2}{\widetilde{T}} + \lambda_{\mathrm{m}} C^M \frac{e\hat{t}}{\widetilde{T}} \left( \frac{D(\omega_{t'}) - D(\omega_{t'+\widetilde{T}-1})}{\beta \widetilde{T}} \right) + \frac{e\lambda_{\mathrm{m}} \hat{t}}{\widetilde{T}^2} \sum_{t=t'}^{t'+\widetilde{T}-1} \widetilde{G}_t^\omega + 2C_\infty^2 \varepsilon_{\mathrm{actor}}. \tag{128}$$

Thus, it follows that

$$\frac{1}{2} \frac{1}{\widetilde{T}} \sum_{t=t'}^{t'+\widetilde{T}-1} M_t \leq \left( \frac{e\lambda_{\mathrm{m}} C^M C_\phi \hat{t}}{\widetilde{T}} \right)^2 + \frac{1}{\widetilde{T}} \sum_{t=t'}^{t'+\widetilde{T}-1} \sum_{j=t-\hat{t}}^{t-1} (1-q)^{t-j} \left( \widetilde{G}_j^\omega + \widetilde{G}_j^\theta + \widetilde{G}_j^\eta \right)$$

$$+ \frac{4B^2 + R_{\max}^2}{\widetilde{T}} + \lambda_{\mathrm{m}} C^M \frac{e\hat{t}}{\widetilde{T}} \left( \frac{D(\omega_{t'}) - D(\omega_{t'+\widetilde{T}-1})}{\beta\widetilde{T}} \right) + \frac{e\lambda_{\mathrm{m}}\hat{t}}{\widetilde{T}^2} \sum_{t=t'}^{t'+\widetilde{T}-1} \widetilde{G}_t^\omega + 2C_\infty^2 \varepsilon_{\mathrm{actor}}. \tag{129}$$

Then, we get that

$$
\begin{aligned}
Y_{t'} &\leq \frac{D(\omega_{t'}) - D(\omega_{t'+\widetilde{T}-1})}{\beta\widetilde{T}} + \frac{C_\phi C_{\mathrm{gap}} m \rho^k}{\lambda_{\min}} + 2C_\infty \sqrt{\varepsilon_{\mathrm{actor}}} + C_\phi \sqrt{\frac{1}{\widetilde{T}} \sum_{t=t'}^{t'+\widetilde{T}-1} M_t} + \frac{B^2 L_\phi}{2}\beta \\
&\leq \frac{D(\omega_{t'}) - D(\omega_{t'+\widetilde{T}-1})}{\beta\widetilde{T}} + \frac{C_\phi C_{\mathrm{gap}} m \rho^k}{\lambda_{\min}} + 4C_\infty \sqrt{\varepsilon_{\mathrm{actor}}} + C_\phi \sqrt{2} \left( \frac{e\lambda_{\mathrm{m}} C^M C_\phi \hat{t}}{\widetilde{T}} \right) \\
&\quad + C_\phi \sqrt{2} \sqrt{\frac{1}{\widetilde{T}} \sum_{t=t'}^{t'+\widetilde{T}-1} \sum_{j=t-\hat{t}}^{t-1} (1-q)^{t-j} \left( \widetilde{G}_j^\omega + \widetilde{G}_j^\theta + \widetilde{G}_j^\eta \right)} + C_\phi \sqrt{2} \sqrt{\frac{4B^2 + R_{\max}^2}{\widetilde{T}}} \\
&\quad + C_\phi \sqrt{2} \sqrt{\lambda_{\mathrm{m}} C^M \frac{e\hat{t}}{\widetilde{T}} \frac{D(\omega_{t'}) - D(\omega_{t'+\widetilde{T}-1})}{\beta\widetilde{T}}} + C_\phi \sqrt{2} \sqrt{\frac{e\hat{t}\lambda_{\mathrm{m}}}{\widetilde{T}^2} \sum_{t=t'}^{t'+\widetilde{T}-1} \widetilde{G}_t^\omega} + \frac{B^2 L_\phi}{2}\beta, \tag{130}
\end{aligned}
$$

which proves the claim under Case 2.

Thus, combine the Case 1 result Equation (124) and the Case 2 result Equation (130), we have that

$$
\begin{aligned}
\min_{t \leq T} &\mathbb{E}\left[ J(\pi^*) - J(\omega_t) \right] \\
&\leq \frac{D(\omega_{t'}) - D(\omega_{t'+\widetilde{T}-1})}{\beta\widetilde{T}} + \frac{C_\phi C_{\mathrm{gap}} m \rho^k}{\lambda_{\min}} + 4C_\infty \sqrt{\varepsilon_{\mathrm{actor}}} + C_\phi \sqrt{2} \left( \frac{e\lambda_{\mathrm{m}} C^M C_\phi \hat{t}}{\widetilde{T}} \right) \\
&\quad + C_\phi \sqrt{2} \sqrt{\frac{1}{\widetilde{T}} \sum_{t=t'}^{t'+\widetilde{T}-1} \sum_{j=t-\hat{t}}^{t-1} (1-q)^{t-j} \left( \widetilde{G}_j^\omega + \widetilde{G}_j^\theta + \widetilde{G}_j^\eta \right)} + C_\phi \sqrt{8\frac{B^2 + R_{\max}^2}{\widetilde{T}}} + \frac{B^2 L_\phi}{2}\beta \\
&\quad + C_\phi \sqrt{2} \sqrt{\lambda_{\mathrm{m}} C^M \frac{e\hat{t}}{\widetilde{T}} \frac{D(\omega_{t'}) - D(\omega_{t'+\widetilde{T}-1})}{\beta\widetilde{T}}} + C_\phi \sqrt{2} \sqrt{\frac{e\hat{t}\lambda_{\mathrm{m}}}{\widetilde{T}^2} \sum_{t=t'}^{t'+\widetilde{T}-1} \widetilde{G}_t^\omega} + \frac{2R_{\max}}{T}. \tag{131}
\end{aligned}
$$

which completes the proof. $\qquad\square$

Next, we prove Theorem 2.

**Theorem 4.** *(Restatement of Theorem 2) Consider the NAC algorithm in Algorithm 1 with constant step sizes, it holds that*

$$
\begin{aligned}
\min_{t < T} \mathbb{E}\left[ J(\pi^*) - J(\omega_t) \right] \leq &\mathcal{O}\left( \frac{\log^2 T}{T\alpha} \right) + \mathcal{O}\left( \frac{\log T}{T\beta} \right) + \mathcal{O}\left( \frac{\log^2 T}{T\sqrt{\alpha\beta}} \right) + \mathcal{O}\left( \frac{\sqrt{\gamma\beta}\log^2 T}{\sqrt{\alpha}} \right) \\
&+ \mathcal{O}\left( \frac{\gamma \log^2 T}{\sqrt{\alpha}} \right) + \mathcal{O}\left( \frac{\beta \log^2 T}{\sqrt{\alpha}} \right) + \mathcal{O}\left( \sqrt{\alpha} \log^2 T \right) \\
&+ \mathcal{O}\left( \sqrt{\beta} \log^2 T \right) + \mathcal{O}\left( \sqrt{\varepsilon_{actor}} \right).
\end{aligned}
$$

*If we set $\gamma = \mathcal{O}(T^{-\frac{2}{3}}), \alpha = \mathcal{O}(T^{-\frac{2}{3}} \log^{-2} T), \beta = \mathcal{O}(T^{-\frac{2}{3}} \log^{-2} T)$, we have*

$$\min_{t < T} J(\pi^*) - J(\omega_t) \leq \mathcal{O}\left( T^{-\frac{1}{3}} \log^4 T \right) + \mathcal{O}\left( \sqrt{\varepsilon_{actor}} \right). \tag{132}$$

*Proof.* From Lemma 13, we have that

$$\min_{t \leq T} \mathbb{E}\left[ J(\pi^*) - J(\omega_t) \right]$$

$$\leq \frac{D(\omega_{t'}) - D(\omega_{t'+\widetilde{T}-1})}{\beta\widetilde{T}} + \frac{C_\phi C_{\text{gap}} m\rho^k}{\lambda_{\min}} + 4C_\infty \sqrt{\varepsilon_{\text{actor}}} + C_\phi \sqrt{2} \left( \frac{e\lambda_{\text{m}} C^M C_\phi \hat{t}}{\widetilde{T}} \right)$$

$$+ C_\phi \sqrt{2} \sqrt{\frac{1}{\widetilde{T}} \sum_{t=t'}^{t'+\widetilde{T}-1} \sum_{j=t-\hat{t}}^{t-1} (1-q)^{t-j} \left( \widetilde{G}_j^\omega + \widetilde{G}_j^\theta + \widetilde{G}_j^\eta \right)} + C_\phi \sqrt{\frac{8B^2 + 2R_{\max}^2}{\widetilde{T}}} + \frac{B^2 L_\phi}{2}\beta$$

$$+ C_\phi \sqrt{2} \sqrt{\lambda_{\text{m}} C^M \frac{e\hat{t}}{\widetilde{T}} \frac{D(\omega_{t'}) - D(\omega_{t'+\widetilde{T}-1})}{\beta\widetilde{T}}} + C_\phi \sqrt{2} \sqrt{\frac{e\hat{t}\lambda_{\text{m}}}{\widetilde{T}^2} \sum_{t=t'}^{t'+\widetilde{T}-1} \widetilde{G}_t^\omega} + \frac{R_{\max}}{T}. \tag{133}$$

We then set the stepsize as follows

$$\gamma = T^{-\frac{2}{3}};$$

$$\alpha = \left( \frac{k^2 C_\phi^2}{\bar{\lambda}_{\min}} \right)^{-1} \gamma = \frac{\lambda_{\min}}{2k^2 C_\phi^2} T^{-\frac{2}{3}} = \mathcal{O}\left( T^{-\frac{2}{3}} \log^{-2} T \right);$$

$$\beta = \left( \frac{24\lambda_{\text{m}}^2 C^M}{\lambda_{\min}} \right)^{-1} \alpha = \frac{\lambda_{\min}^2}{48k^2 C_\phi^2 \lambda_{\text{m}} C^M} T^{-\frac{2}{3}} = \mathcal{O}\left( T^{-\frac{2}{3}} \log^{-2} T \right). \tag{134}$$

Applying the above stepsizes Equation (134), for $t \leq T$, we can have that

$$\widetilde{G}_t^\eta = \mathcal{O}\left( (m\rho^k)\gamma + \beta^2 + k^2\beta\gamma + k\gamma^2 \right) = \mathcal{O}\left( T^{-\frac{4}{3}} \log T \right);$$

$$\widetilde{G}_t^\theta = \mathcal{O}\left( k(m\rho^k)\alpha + k(m\rho^k)\beta + \frac{(m\rho^k)^2}{\beta} + k^3\alpha^2 + k^3\alpha\beta + k^2\beta^2 \right) = \mathcal{O}\left( T^{-\frac{4}{3}} \log^2 T \right);$$

$$\widetilde{G}_t^\omega = \mathcal{O}\left( (m\rho^k)\beta + \beta^2 \right) \frac{\beta^2 \log^2 T}{\alpha T} + \frac{\beta^2 \log^2 T}{\alpha T} \right) = \mathcal{O}\left( T^{-\frac{4}{3}} \log^{-4} T \right). \tag{135}$$

Thus, it holds that

$$\sqrt{\frac{1}{\widetilde{T}} \sum_{t=t'}^{t'+\widetilde{T}-1} \sum_{j=t-\hat{t}}^{t-1} (1-q)^{t-j} \left( \widetilde{G}_j^\omega + \widetilde{G}_j^\theta + \widetilde{G}_j^\eta \right)} = \mathcal{O}\left( T^{-\frac{2}{3}} \log^2 T \right). \tag{136}$$

We further recall that

$$q = \frac{\bar{\lambda}_{\min}\alpha}{2} = \mathcal{O}\left( \alpha \right);$$

$$\hat{t} = \left\lceil \frac{1}{q} \log T \right\rceil = \mathcal{O}\left( \frac{\log T}{\alpha} \right);$$

$$\widetilde{T} = \left\lceil \frac{T}{\hat{t} \log T} \right\rceil \hat{t} = \mathcal{O}\left( \frac{T}{\log T} \right). \tag{137}$$

Plugging the above equations to Equation (131) , we have that

$$\min_{t \leq T} \mathbb{E}\left[ J(\pi^*) - J(\omega_t) \right] \leq \left( T^{-\frac{1}{3}} \log^4 T \right) + \mathcal{O}\left( \sqrt{\varepsilon_{\text{actor}}} \right). \tag{138}$$

This concludes the proof. □

# D    PROOF OF LEMMAS

*Proof of Lemma 1.* Recall the definition of $\nabla J(\omega)$ in Equation (1):

$$\nabla J(\omega) = \mathbb{E}_{D_{\pi_\omega}} \left[ Q^{\pi_\omega}(s, a)\phi_\omega(s, a) \right], \tag{139}$$

which implies that

$$
\begin{aligned}
\|\nabla J(\omega)\|_2 &= \left\| \mathbb{E}_{D_{\pi_\omega}} \left[ Q^{\pi_\omega}(s,a)\phi_\omega(s,a) \right] \right\|_2 \\
&\overset{(a)}{=} \left\| \mathbb{E}_{D_{\pi_\omega}} \left[ \phi_\omega^\top(s,a)\bar{\theta}_\omega^* \phi_\omega(s,a) \right] \right\|_2 \\
&\leq C_\phi^2 \left( \|\theta_\omega^*\|_2 + \|\bar{\theta}_\omega^* - \theta_\omega^*\|_2 \right) \\
&\overset{(b)}{\leq} C_\phi^2 \left( B + C_{\text{gap}} \frac{m\rho^k}{\lambda_{\min}} \right) = C_J,
\end{aligned}
\tag{140}
$$

where $(a)$ follows from Equation (3) and $(b)$ follows from Proposition 4. It hence proves the first claim. The second claim is proved in (Xu & Gu, 2020; Wu et al., 2020). $\qquad \square$

*Proof of Lemma2.* Recall $F_\omega = \mathbb{E}_{D_{\pi_\omega}} \left[ \phi_\omega(s,a)\phi_\omega^\top(s,a) \right]$. From the definition of $\bar{\theta}_\omega^*$ in Equation (2), it can be verified that

$$
\bar{\theta}_\omega^* = F_\omega^{-1} \nabla J(\omega).
\tag{141}
$$

Hence

$$
\begin{aligned}
\|\bar{\theta}_\omega^* &- \bar{\theta}_{\omega'}^*\|_2 \\
&= \left\| F_\omega^{-1} \nabla J(\omega) - F_{\omega'}^{-1} \nabla J(\omega') \right\|_2 \\
&\leq \left\| F_\omega^{-1} \nabla J(\omega) - F_{\omega'}^{-1} \nabla J(\omega) \right\|_2 + \left\| F_{\omega'}^{-1} \nabla J(\omega) - F_{\omega'}^{-1} \nabla J(\omega') \right\|_2 \\
&\overset{(a)}{\leq} \left\| F_\omega^{-1} \right\|_2 \left\| F_{\omega'}^{-1} \right\|_2 \left\| F_\omega - F_{\omega'} \right\|_2 \left\| \nabla J(\omega) \right\|_2 + \left\| (F_{\omega'})^{-1} \right\|_2 \left\| \nabla J(\omega) - \nabla J(\omega') \right\|_2,
\end{aligned}
\tag{142}
$$

where $(a)$ follows from the facts that for positive definite matrices $A_1$ and $A_2$,

$$
\begin{aligned}
\left\| A_1^{-1} - A_2^{-1} \right\|_2 &\leq \left\| A_1^{-1} (A_1 - A_2) A_2^{-1} \right\|_2 \\
&\leq \left\| A_1^{-1} \right\|_2 \left\| A_1 - A_2 \right\|_2 \left\| A_2^{-1} \right\|_2.
\end{aligned}
\tag{143}
$$

Note that $F_\omega$ can be shown to be Lipschitz as follows

$$
\begin{aligned}
\|F_\omega - F_{\omega'}\|_2 &= \left\| \mathbb{E}_{D_{\pi_\omega}} \left[ \phi_\omega(s,a)\phi_\omega^\top(s,a) \right] - \mathbb{E}_{D_{\pi_{\omega'}}} \left[ \phi_{\omega'}(s,a)\phi_{\omega'}^\top(s,a) \right] \right\|_2 \\
&\leq \left\| \mathbb{E}_{D_{\pi_\omega}} \left[ \phi_\omega(s,a)\phi_\omega^\top(s,a) \right] - \mathbb{E}_{D_{\pi_\omega}} \left[ \phi_{\omega'}(s,a)\phi_{\omega'}^\top(s,a) \right] \right\|_2 \\
&\quad + \left\| \mathbb{E}_{D_{\pi_\omega}} \left[ \phi_{\omega'}(s,a)\phi_{\omega'}^\top(s,a) \right] - \mathbb{E}_{D_{\pi_{\omega'}}} \left[ \phi_{\omega'}(s,a)\phi_{\omega'}^\top(s,a) \right] \right\|_2 \\
&\leq 2C_\phi L_\phi \|\omega' - \omega\|_2 + C_\phi^2 \left\| D_{\pi_\omega} - D_{\pi_{\omega'}} \right\|_{\mathcal{TV}} \\
&\overset{(a)}{\leq} 2C_\phi L_\phi \|\omega' - \omega\|_2 + C_\phi^2 L_\pi \|\omega - \omega'\|_2,
\end{aligned}
\tag{144}
$$

where $(a)$ follows from (Zou et al., 2019) and Theorem 1 in (Li et al., 2021b), recall $L_\pi = \frac{1}{2} C_\pi \left( 1 + \lceil \log m^{-1} \rceil + \frac{1}{1-\rho} \right)$,

$$
\left\| D_{\pi_\omega} - D_{\pi_{\omega'}} \right\|_{\mathcal{TV}} \leq L_\pi \|\omega - \omega'\|_2.
\tag{145}
$$

Hence combining Equation (142), Equation (144) and Lemma 1, we obtain that

$$
\|\bar{\theta}_\omega^* - \bar{\theta}_{\omega'}^*\|_2 \leq \left( \frac{C_J}{\lambda_{\min}^2} \left( 2C_\phi L_\phi + C_\phi^2 L_\pi \right) + \frac{L_J}{\lambda_{\min}} \right) \|\omega - \omega'\|_2.
\tag{146}
$$

This completes the proof.

$\qquad \square$

*Proof of Lemma 3.* From the definition, we first have that

$$A_\omega = \mathbb{E}_{D_{\pi_\omega}} \left[ \mathbb{E} \left[ \phi_\omega(s_0, a_0) \left( \phi_\omega(s_k, a_k) - \phi_\omega(s_0, a_0) \right)^\top | s_0 = s, a_0 = a, \pi_\omega \right] \right]$$

$$\overset{(a)}{=} \mathbb{E}_{D_{\pi_\omega}} \left[ \phi_\omega(s, a) \left( \mathbb{E} \left[ \phi_\omega^\top(s_k, a_k) | s_0 = s, a_0 = a, \pi_\omega \right] - \mathbb{E}_{D_{\pi_\omega}} \left[ \phi_\omega^\top(s, a) \right] \right) \right]$$

$$- \mathbb{E}_{D_{\pi_\omega}} \left[ \phi_\omega(s, a) \phi_\omega^\top(s, a) \right], \tag{147}$$

where $(a)$ follows from $\phi_\omega(s, a) = \nabla \log \pi_\omega(a|s)$ and $\mathbb{E}_{D_{\pi_\omega}} \left[ \phi_\omega^\top(s, a) f(s) \right] = 0$, where $f(s)$ is the function which is not determined by action $a$.

Define $\Delta A_\omega = \mathbb{E}_{D_{\pi_\omega}} \left[ \phi_\omega(s, a) \left( \mathbb{E} \left[ \phi_\omega^\top(s_k, a_k) | s_0 = s, a_0 = a, \pi_\omega \right] - \mathbb{E}_{D_{\pi_\omega}} \left[ \phi_\omega^\top(s, a) \right] \right) \right]$. Thus,

$$\frac{A_\omega + A_\omega^\top}{2} = \frac{\Delta A_\omega + \Delta A_\omega^\top}{2} - \mathbb{E}_{D_{\pi_\omega}} \left[ \phi_\omega(s, a) \phi_\omega^\top(s, a) \right]. \tag{148}$$

For any symmetric matrices $X$ and $Y$, $\lambda_{\max}(X + Y) \leq \lambda_{\max}(X) + \lambda_{\max}(Y)$. Thus, we have that

$$\lambda_{\max} \left( \frac{A_\omega + A_\omega^\top}{2} \right) \leq \lambda_{\max} \left( \frac{\Delta A_\omega + \Delta A_\omega^\top}{2} \right) + \lambda_{\max} \left( -\mathbb{E}_{D_{\pi_\omega}} \left[ \phi_\omega(s, a) \phi_\omega^\top(s, a) \right] \right)$$

$$\leq C_\phi^2 \mathbb{E} \left[ \| P(s_k, a_k | s_0 = s, a_0 = s, \pi_\omega), D_{\pi_\omega} \|_{\mathcal{TV}} \right] - \lambda_{\min}$$

$$\leq d C_\phi^2 m \rho^k - \lambda_{\min} = -\bar{\lambda}_{\min}. \tag{149}$$

$\square$

*Proof of Lemma 4.* Conditioned on $(s_{t-k}, a_{t-k})$, the sample trajectory in Algorithm 1 is generated according to the following Markov chain:

$$(s_{t-k}, a_{t-k}) \xrightarrow{\pi_{t-k} \times P} (s_{t-k+1}, a_{t-k+1}) \xrightarrow{\pi_{t-k+1} \times P} ...(s_t, a_t) \xrightarrow{\pi_t \times P} (s_{t+1}, a_{t+1}). \tag{150}$$

Using the technique in (Zou et al., 2019), we construct an auxiliary Markov chain as follows. Before time $t - k$, the states and actions are generated according to Algorithm 1; and after time $t - k$, all the subsequent state-action pairs, denoted by $(\tilde{s}_l, \tilde{a}_l)$, are generated according to a fixed policy $\pi_t$ and transition kernel $P$:

$$(s_{t-k}, a_{t-k}) \xrightarrow{\pi_t \times P} (\tilde{s}_{t-k+1}, \tilde{a}_{t-k+1}) \xrightarrow{\pi_t \times P} ...(\tilde{s}_t, \tilde{a}_t) \xrightarrow{\pi_t \times P} (\tilde{s}_{t+1}, \tilde{a}_{t+1}). \tag{151}$$

Denote by $\widetilde{\mathcal{F}}_t$ the filtration corresponding to the auxiliary Markov chain designed in Equation (151).

Then follow steps similar to those in (Zou et al., 2019, Appendix B) and (Li et al., 2021b, Lemma 6), it can be shown that

$$\| \mathbb{P}(s_t, a_t | \mathcal{F}_{t-k}) - D_t \|_{\mathcal{TV}} \leq m \rho^k + \sum_{j=t-k}^{t} C_\pi \| \omega_t - \omega_j \|_2. \tag{152}$$

$\square$

*Proof of Lemma 5.* Define the sum of the feature along the trajectory as follows:

$$z_t = \sum_{j=t-k}^{t} \phi_j(s_j, a_j), \quad \hat{z}_t = \sum_{j=t-k}^{t} \phi_t(s_j, a_j) \text{ and } \tilde{z}_t = \sum_{j=t-k}^{t} \phi_t(\tilde{s}_j, \tilde{a}_j). \tag{153}$$

For every policy $\pi_t$, we construct another auxiliary Markov chain, denoted by $\{(\bar{s}_j, \bar{a}_j)\}_{j=0}^{\infty}$, which is under the stationary distribution induced by policy $\pi_t$ and transition kernel $P$, i.e.,

$$(\bar{s}_0, \bar{a}_0) \sim D_t, \tag{154}$$

and all the subsequent actions are generated by $\pi_t$. Define

$$\bar{z}_t = \sum_{j=t-k}^{t} \phi_t(\bar{s}_j, \bar{a}_j). \tag{155}$$

Denote by $\bar{\delta}_t(\bar{s}_t, \bar{a}_t; \theta_t, \omega_t) = R(\bar{s}_t, \bar{a}_t) - J(\omega_t) + \phi_t^\top(\bar{s}_{t+1}, \bar{a}_{t+1})\theta_t - \phi_t^\top(\bar{s}_t, \bar{a}_t)\theta_t$.

**Lemma 14.** *It holds that*

$$\mathbb{E}\left[\bar{z}_t\bar{\delta}_t(\bar{s}_t, \bar{a}_t; \theta_t, \omega_t)|\pi_t\right] = A_{\omega_t}\theta_t. \tag{156}$$

From the definition in Equation (4), $\theta_t^*$ is the fixed point of the $k$-step TD operator $\mathcal{T}_{\pi_t}^{(k)}$. Then, it follows that

$$A_{\omega_t}\theta_t^* = \mathbb{E}_{D_t}\left[\phi_t^\top(s, a)\left(\mathcal{T}_{\pi_t}^{(k)}\phi_t^\top(s, a)\theta_t^* - \phi_t^\top(s, a)\theta_t^*\right)\right] = \mathbf{0}. \tag{157}$$

Together with Lemma 14, we have that

$$\mathbb{E}\left[\bar{z}_t\bar{\delta}_t(\bar{s}_t, \bar{a}_t; \theta_t^*, \omega_t)\right] = 0. \tag{158}$$

Thus we have that

$$\begin{aligned}
\mathbb{E}\left[\langle\theta_t - \theta_t^*, \bar{z}_t\bar{\delta}_t(\bar{s}_t, \bar{a}_t; \theta_t, \omega_t)\rangle\right] &= \mathbb{E}\left[\langle\theta_t - \theta_t^*, \bar{z}_t\bar{\delta}_t(\bar{s}_t, \bar{a}_t; \theta_t, \omega_t) - \bar{z}_t\bar{\delta}_t(\bar{s}_t, \bar{a}_t; \theta_t^*, \omega_t)\rangle\right] \\
&= \mathbb{E}\left[\langle\theta_t - \theta_t^*, A_{\omega_t}(\theta_t - \theta_t^*)\rangle\right] \\
&\leq \lambda_{\max}\left(\frac{A_{\omega_t} + A_{\omega_t}^\top}{2}\right)\mathbb{E}\left[\|\theta_t - \theta_t^*\|_2^2\right] \\
&\stackrel{(a)}{\leq} -\bar{\lambda}_{\min}\mathbb{E}\left[\|\theta_t - \theta_t^*\|_2^2\right],
\end{aligned} \tag{159}$$

where $(a)$ follows from Lemma 3.

Then, recall $\hat{z}_t = \sum_{j=t-k}^{t} \phi_t(s_j, a_j)$. Denote by $\hat{\delta}_t = R(s_t, a_t) - J(\omega_t) + \phi_t^\top(s_{t+1}, a_{t+1})\theta_t - \phi_t^\top(s_t, a_t)\theta_t$, we have that

$$\begin{aligned}
\mathbb{E}\left[\langle\theta_t - \theta_t^*, \delta_t z_t\rangle\right] &= \mathbb{E}\left[\langle\theta_t - \theta_t^*, \bar{z}_t\bar{\delta}_t(\bar{s}_t, \bar{a}_t; \theta_t, \omega_t)\rangle\right] + \mathbb{E}\left[\left\langle\theta_t - \theta_t^*, z_t\hat{\delta}_t - \bar{z}_t\bar{\delta}_t(\bar{s}_t, \bar{a}_t; \theta_t, \omega_t)\right\rangle\right] \\
&\quad + \mathbb{E}\left[\left\langle\theta_t - \theta_t^*, z_t\delta_t - \hat{z}_t\hat{\delta}_t\right\rangle\right] \\
&\stackrel{(a)}{\leq} -\bar{\lambda}_{\min}\mathbb{E}\left[\|\theta_t - \theta_t^*\|_2^2\right] + \mathbb{E}\left[\left\langle\theta_t - \theta_t^*, z_t\hat{\delta}_t - \bar{z}_t\bar{\delta}_t(\bar{s}_t, \bar{a}_t; \theta_t)\right\rangle\right] \\
&\quad + \mathbb{E}\left[\langle\theta_t - \theta_t^*, z_t(J(\omega_t) - \eta_t)\rangle\right] \\
&\leq -\bar{\lambda}_{\min}\mathbb{E}\left[\|\theta_t - \theta_t^*\|_2^2\right] + \mathbb{E}\left[\left\langle\theta_t - \theta_t^*, z_t\hat{\delta}_t - \bar{z}_t\bar{\delta}_t(\bar{s}_t, \bar{a}_t; \theta_t)\right\rangle\right] \\
&\quad + \frac{\bar{\lambda}_{\min}}{2}\mathbb{E}\left[\|\theta_t - \theta_t^*\|_2^2\right] + \frac{(k+1)^2C_\phi^2}{2\bar{\lambda}_{\min}}\mathbb{E}\left[\|J(\omega_t) - \eta_t\|_2^2\right],
\end{aligned} \tag{160}$$

where $(a)$ follows from Equation (159).

Consider the term $\mathbb{E}\left[\left\langle\theta_t - \theta_t^*, z_t\hat{\delta}_t - \bar{z}_t\bar{\delta}_t(\bar{s}_t, \bar{a}_t; \theta_t, \omega_t)\right\rangle\right]$, and we have that

$$\begin{aligned}
&\mathbb{E}\left[\left\langle\theta_t - \theta_t^*, z_t\hat{\delta}_t - \bar{z}_t\bar{\delta}_t(\bar{s}_t, \bar{a}_t; \theta_t, \omega_t)\right\rangle\right] \\
&= \mathbb{E}\left[\left\langle\theta_t - \theta_{t-2k} - \theta_t^* + \theta_{t-2k}^*, z_t\hat{\delta}_t - \bar{z}_t\bar{\delta}_t(\bar{s}_t, \bar{a}_t; \theta_t, \omega_t)\right\rangle\right] \\
&\quad + \mathbb{E}\left[\left\langle\theta_{t-2k} - \theta_{t-2k}^*, z_t\hat{\delta}_t - \bar{z}_t\bar{\delta}_t(\bar{s}_t, \bar{a}_t; \theta_t, \omega_t)\right\rangle\right]
\end{aligned}$$

$$\leq \underbrace{\mathbb{E}\left[\left\|\theta_t - \theta_{t-2k}\right\|_2 + \left\|\theta_t^* - \theta_{t-2k}^*\right\|_2 \left(\left\|z_t\right\|_2 \left\|\hat{\delta}_t\right\|_2 + \left\|\bar{z}_t\right\|_2 \left\|\bar{\delta}_t(\bar{s}_t, \bar{a}_t; \theta_t, \omega_t)\right\|_2\right)\right]}_{(i)}$$

$$+ \underbrace{\mathbb{E}\left[\left\langle\theta_{t-2k} - \theta_{t-2k}^*, z_t\hat{\delta}_t - \hat{z}_t\hat{\delta}_t\right\rangle\right]}_{(ii)}$$

$$+ \underbrace{\mathbb{E}\left[\left\langle\theta_{t-2k} - \theta_{t-2k}^*, \hat{z}_t\hat{\delta}_t - \bar{z}_t\bar{\delta}_t(\bar{s}_t, \bar{a}_t; \theta_t, \omega_t)\right\rangle\right]}_{(iii)}. \tag{161}$$

In AC algorithm, recall $U_\delta = R_{\max} + C_\phi B$. Then, consider $\left\|\theta_t - \theta_{t-2k}\right\|_2$ and $\left\|\theta_t^* - \theta_{t-2k}^*\right\|_2$, we have that

$$\left\|\theta_t - \theta_{t-2k}\right\|_2 \leq \left\|\sum_{j=t-2k}^{t-1} \alpha_j\delta_j z_j\right\|_2 \leq \sum_{j=t-2k}^{t-1} \alpha_j \left|\delta_j\right| \left\|z_j\right\|_2 \overset{(a)}{\leq} (k+1)C_\phi U_\delta \sum_{j=t-2k}^{t-1} \alpha_j, \tag{162}$$

where $(a)$ follows from the fact that $\left\|z_t\right\|_2 \leq (k+1)C_\phi$.

Then, it can be shown that

$$\begin{aligned}
\left\|\theta_t^* - \theta_{t-2k}^*\right\|_2 &= \left\|\bar{\theta}_t^* - \bar{\theta}_{t-2k}^* + \theta_t^* - \bar{\theta}_t^* + \bar{\theta}_{t-2k}^* - \theta_{t-2k}^*\right\|_2 \\
&\leq \left\|\bar{\theta}_t^* - \bar{\theta}_{t-2k}^*\right\|_2 + \left\|\theta_t^* - \bar{\theta}_t^*\right\|_2 + \left\|\bar{\theta}_{t-2k}^* - \theta_{t-2k}^*\right\|_2 \\
&\leq C_\Theta \left\|\omega_t - \omega_{t-2k}\right\|_2 + \frac{C_{\text{gap}}m\rho^k}{\lambda_{\min}} + \frac{C_{\text{gap}}m\rho^k}{\lambda_{\min}} \\
&\leq C_\Theta \left\|\sum_{j=t-2k}^{t-1} \beta_j\phi_j^\top(s_j, a_j)\theta_j\phi_j(s_j, a_j)\right\|_2 + \frac{2C_{\text{gap}}m\rho^k}{\lambda_{\min}} \\
&\leq C_\Theta C_\phi^2 B \sum_{j=t-2k}^{t-1} \beta_j + \frac{2C_{\text{gap}}m\rho^k}{\lambda_{\min}}. \tag{163}
\end{aligned}$$

Thus, from Equation (163), the term $(i)$ in Equation (161) can be bounded as follows

$$(i) \leq 2(k+1)C_\phi U_\delta \left((k+1)C_\phi U_\delta \sum_{j=t-2k}^{t-1} \alpha_j + C_\Theta C_\phi^2 B \sum_{j=t-2k}^{t-1} \beta_j + 2C_{\text{gap}}\frac{m\rho^k}{\lambda_{\min}}\right). \tag{164}$$

Then, for term $(ii)$ in Equation (161), it can be bounded as follows

$$\begin{aligned}
(ii) &\leq \mathbb{E}\left[\left\|\theta_{t-2k} - \theta_{t-2k}^*\right\|_2 \left\|z_t - \hat{z}_t\right\|_2 \left|\delta_t(\theta_t)\right|\right] \leq 2BU_\delta \left\|\sum_{j=t-k}^{t} \phi_j(s_j, a_j) - \phi_t(s_j, a_j)\right\|_2 \\
&\leq 2BU_\delta \sum_{j=t-k}^{t} C_\pi\mathbb{E}\left[\left\|\omega_t - \omega_j\right\|_2\right] \leq 2BU_\delta C_\pi \sum_{j=t-k}^{t}\sum_{i=j}^{t} \mathbb{E}\left[\left\|\beta_i\phi_i^\top(s_i, a_i)\theta_i\phi_i(s_i, a_i)\right\|_2\right] \\
&\leq 2B^2C_\phi^2 U_\delta C_\pi \sum_{j=t-k}^{t}\sum_{i=j}^{t} \beta_i. \tag{165}
\end{aligned}$$

Next, for term $(iii)$ in Equation (161), we can show that

$$\begin{aligned}
(iii) &= \mathbb{E}\left[\left\langle\theta_{t-2k} - \theta_{t-2k}^*, \hat{z}_t\hat{\delta}_t - \bar{z}_t\bar{\delta}_t(\bar{s}_t, \bar{a}_t; \theta_t)\right\rangle\right] \\
&= \mathbb{E}\left[\mathbb{E}\left[\left\langle\theta_{t-2k} - \theta_{t-2k}^*, \hat{z}_t\hat{\delta}_t - \bar{z}_t\bar{\delta}_t(\bar{s}_t, \bar{a}_t; \theta_t)\right\rangle | \mathcal{F}_{t-2k}\right]\right]
\end{aligned}$$

$$\leq 4BC_\phi U_\delta \sum_{j=t-k}^{t} \mathbb{E}\left[\|\mathbb{P}(s_j, a_j | \mathcal{F}_{t-2k}) - D_t\|_{\mathcal{TV}}\right]$$

$$\leq 4BC_\phi U_\delta \sum_{j=t-k}^{t} \mathbb{E}\left[\|\mathbb{P}(s_j, a_j | \mathcal{F}_{t-2k}) - D_j\|_{\mathcal{TV}} + \|D_j - D_t\|_{\mathcal{TV}}\right]$$

$$\leq 4BC_\phi U_\delta \sum_{j=t-k}^{t} \left(C_\pi \sum_{i=j-k}^{j-1} \mathbb{E}\left[\|\omega_i - \omega_j\|_2\right] + m\rho^k + L_\pi \mathbb{E}\left[\|\omega_t - \omega_j\|_2\right]\right)$$

$$\leq 4BC_\phi U_\delta \sum_{j=t-k}^{t} \left(C_\pi \sum_{i=j-k}^{j-1} \sum_{\iota=i}^{j-1} \beta_\iota C_\phi^2 B + m\rho^k + L_\pi \sum_{i=j}^{t-1} \beta_i C_\phi^2 B\right). \tag{166}$$

Thus, combining Equation (164),Equation (165) and Equation (166), we can bound term as follows

$$\mathbb{E}\left[\langle \theta_t - \theta_t^*, \delta_t z_t \rangle\right] \leq -\frac{\bar{\lambda}_{\min}}{2}\mathbb{E}\left[\|\theta_t - \theta_t^*\|_2^2\right] + \frac{(k+1)^2 C_\phi^2}{2\bar{\lambda}_{\min}}\mathbb{E}\left[\|J(\omega_t) - \eta_t\|_2^2\right]$$

$$+ 2B^2 C_\phi^2 U_\delta C_\pi \sum_{j=t-k}^{t} \sum_{i=j}^{t} \beta_i + 4BC_\phi U_\delta \sum_{j=t-k}^{t} \left(BC_\phi^2 C_\pi \sum_{i=j-k}^{j-1} \sum_{\iota=i}^{j-1} \beta_\iota + m\rho^k + BC_\phi^2 L_\pi \sum_{i=j}^{t-1} \beta_i\right)$$

$$+ 2(k+1)C_\phi U_\delta \left((k+1)C_\phi U_\delta \sum_{j=t-2k}^{t-1} \alpha_j + C_\Theta C_\phi^2 B \sum_{j=t-2k}^{t-1} \beta_j + \frac{2C_{\text{gap}} m\rho^k}{\lambda_{\min}}\right). \tag{167}$$

In NAC algorithm, terms $\|\theta_t - \theta_{t-2k}\|_2$ and $\|\theta_t^* - \theta_{t-2k}^*\|_2$ can be bounded as follows

$$\|\theta_t - \theta_{t-2k}\|_2 \leq \left\|\sum_{j=t-2k}^{t-1} \alpha_j \delta_j z_j\right\|_2 \leq (k+1)C_\phi U_\delta \sum_{j=t-2k}^{t-1} \alpha_j, \tag{168}$$

and

$$\|\theta_t^* - \theta_{t-2k}^*\|_2 = \|\bar{\theta}_t^* - \bar{\theta}_{t-2k}^* + \theta_t^* - \bar{\theta}_t^* + \bar{\theta}_{t-2k}^* - \theta_{t-2k}^*\|_2$$

$$\leq \|\bar{\theta}_t^* - \bar{\theta}_{t-2k}^*\|_2 + \|\theta_t^* - \bar{\theta}_t^*\|_2 + \|\bar{\theta}_{t-2k}^* - \theta_{t-2k}^*\|_2$$

$$\leq C_\Theta \|(\omega_t - \omega_{t-2k})\|_2 + \frac{C_{\text{gap}} m\rho^k}{\lambda_{\min}} + \frac{C_{\text{gap}} m\rho^k}{\lambda_{\min}}$$

$$\leq C_\Theta \left\|\sum_{j=t-2k}^{t-1} \beta_j \theta_j\right\|_2 + \frac{2C_{\text{gap}} m\rho^k}{\lambda_{\min}}$$

$$\leq C_\Theta B \sum_{j=t-2k}^{t-1} \beta_j + \frac{2C_{\text{gap}} m\rho^k}{\lambda_{\min}}. \tag{169}$$

Thus, using Equation (169) and Equation (168), term $(i)$ in Equation (161) can be bounded as

$$(i) \leq 2(k+1)C_\phi U_\delta \left((k+1)C_\phi U_\delta \sum_{j=t-2k}^{t-1} \alpha_j + C_\Theta B \sum_{j=t-2k}^{t-1} \beta_j + 2C_{\text{gap}} \frac{m\rho^k}{\lambda_{\min}}\right). \tag{170}$$

Next, we bound the term $(ii)$ in Equation (161) as follows

$$(ii) \leq \mathbb{E}\left[\|\theta_{t-2k} - \theta_{t-2k}^*\|_2 \|z_t - \hat{z}_t\|_2 \left|\hat{\delta}_t\right|\right] \leq 2BU_\delta \mathbb{E}\left[\left\|\sum_{j=t-k}^{t} \phi_j(s_j, a_j) - \phi_t(s_j, a_j)\right\|_2\right]$$

$$\leq 2BU_\delta \sum_{j=t-k}^{t} C_\pi \mathbb{E}\left[\|\omega_t - \omega_j\|_2\right] \leq 2BU_\delta C_\pi \sum_{j=t-k}^{t}\sum_{i=j}^{t-1}\|\beta_i\theta_i\|_2 \leq 2B^2 U_\delta C_\pi \sum_{j=t-k}^{t}\sum_{i=j}^{t-1}\beta_i. \tag{171}$$

Term $(iii)$ in Equation (161) can be bounded as

$$(iii) = \mathbb{E}\left[\langle\theta_{t-2k} - \theta_{t-2k}^*, \hat{z}_t\delta_t(\theta_t) - z_t'\delta_t'(\theta_t)\rangle\right]$$

$$= \mathbb{E}\left[\mathbb{E}\left[\langle\theta_{t-2k} - \theta_{t-2k}^*, \hat{z}_t\delta_t(\theta_t) - z_t'\delta_t'(\theta_t)\rangle \,|\mathcal{F}_{t-2k}\right]\right]$$

$$\leq 2B2C_\phi U_\delta \sum_{j=t-k}^{t} \mathbb{E}\left[\|\mathbb{P}\left(s_j, a_j|\mathcal{F}_{t-2k}\right) - D_t\|_{\mathcal{TV}}\right]$$

$$\leq 4BC_\phi U_\delta \sum_{j=t-k}^{t} \mathbb{E}\left[\|\mathbb{P}\left(s_j, a_j|\mathcal{F}_{t-2k}\right) - D_j\|_{\mathcal{TV}} + \|D_j - D_t\|_{\mathcal{TV}}\right]$$

$$\leq 4BC_\phi U_\delta \sum_{j=t-k}^{t} \left(C_\pi \sum_{i=j-k}^{j-1} \mathbb{E}\left[\|\omega_i - \omega_j\|_2\right] + m\rho^k + L_\pi \mathbb{E}\left[\|\omega_t - \omega_j\|_2\right]\right)$$

$$\leq 4B^2 C_\phi U_\delta \sum_{j=t-k}^{t} \left(C_\pi \sum_{i=j-k}^{j-1}\sum_{\iota=i}^{j-1} \beta_\iota + m\rho^k + L_\pi \sum_{i=j}^{t-1} \beta_i\right). \tag{172}$$

Combining the above bounds on terms $(i), (ii), (iii)$, term can be bounded as

$$\mathbb{E}\left[\langle\theta_t - \theta_t^*, \delta_t z_t\rangle\right]$$

$$\leq -\frac{\bar{\lambda}_{\min}}{2}\mathbb{E}\left[\|\theta_t - \theta_t^*\|_2^2\right] + \frac{(k+1)^2 C_\phi^2}{2\bar{\lambda}_{\min}}\mathbb{E}\left[\|J(\omega_t) - \eta_t\|_2^2\right] + 2B^2 U_\delta C_\pi \sum_{j=t-k}^{t}\sum_{i=j}^{t-1}\beta_i$$

$$+ 2(k+1)C_\phi U_\delta \left((k+1)C_\phi U_\delta \sum_{j=t-2k}^{t-1}\alpha_j + C_\Theta B \sum_{j=t-2k}^{t-1}\beta_j + \frac{2C_{\text{gap}}m\rho^k}{\lambda_{\min}}\right)$$

$$+ 4BC_\phi U_\delta \sum_{j=t-k}^{t}\left(BC_\pi \sum_{i=j-k}^{j-1}\sum_{\iota=i}^{j-1}\beta_\iota + m\rho^k + BL_\pi \sum_{i=j}^{t-1}\beta_i\right). \tag{173}$$

This completes the proof. $\qquad\square$

*Proof of Lemma 14.* Consider the probability $\mathbb{P}\left(\bar{s}_j, \bar{a}_j, \bar{s}_{t-k}, \bar{a}_{t-k}\right)$ and term $\mathbb{E}\left[\bar{z}_t\bar{\delta}_t(\bar{s}_t, \bar{a}_t; \theta_t, \omega_t)|\pi_t\right]$.

$$\mathbb{E}\left[\sum_{j=t-k}^{t}\phi_t^\top(\bar{s}_j, \bar{a}_j)\left(R(\bar{s}_t, \bar{a}_t) - J(\omega_t) + \phi_t^\top(\bar{s}_{t+1}, \bar{a}_{t+1})\theta_t - \phi_t^\top(\bar{s}_t, \bar{a}_t)\theta_t\right)\Big|\pi_t\right]$$

$$= \mathbb{E}\left[\sum_{j=t-k}^{t}\phi_t^\top(\bar{s}_j, \bar{a}_j)\bar{\delta}_t(\bar{s}_t, \bar{a}_t; \theta_t, \omega_t)\Big|\pi_t\right]$$

$$= \sum_{s,a}\left[\sum_{j=t-k}^{t}\mathbb{P}\left(\bar{s}_j, \bar{a}_j, \bar{s}_t, \bar{a}_t\right)\phi_t^\top(\bar{s}_j, \bar{a}_j)\delta_t(\bar{s}_t, \bar{a}_t; \theta_t)\Big|\pi_t\right]$$

$$= \sum_{s,a}\left[\sum_{j=t-k}^{t}\mathbb{P}\left(\bar{s}_t, \bar{a}_t, \bar{s}_{2t-j}, \bar{a}_{2t-j}\right)\phi_t^\top(\bar{s}_t, \bar{a}_t)\delta_t(\bar{s}_{2t-j}, \bar{a}_{2t-j}; \theta_t, \omega_t)\Big|\pi_t\right]$$

$$\overset{(a)}{=} \sum_{s,a}\left[\sum_{i=0}^{k}\mathbb{P}\left(\bar{s}_t, \bar{a}_t, \bar{s}_{t+i}, \bar{a}_{t+i}\right))\phi_t^\top(\bar{s}_t, \bar{a}_t)\delta_t(\bar{s}_{t+i}, \bar{a}_{t+i}; \theta_t, \omega_t)\Big|\pi_t\right]$$

$$
\begin{aligned}
&= \mathbb{E}_{(\bar{s}_t,\bar{a}_t)\sim D_t}\left[\phi_t^\top(\bar{s}_t,\bar{a}_t)\sum_{i=0}^k \delta_t(\bar{s}_{t+i},\bar{a}_{t+i};\theta_t,\omega_t)\Big|\pi_t\right]\\
&= \mathbb{E}_{D_t}\left[\phi_t^\top(s,a)\left(\mathcal{T}^{(k)}\phi_t^\top(s,a)\theta_t - \phi_t^\top(s,a)\theta_t\right)\right]\\
&= A_{\omega_t}\theta_t,
\end{aligned}
\tag{174}
$$

where $(a)$ follows from the fact that $\mathbb{P}\left(\bar{s}_j,\bar{a}_j\right) = \mathbb{P}\left(\bar{s}_t,\bar{a}_t\right) \sim D_t$, thus,

$$
\begin{aligned}
\mathbb{P}\left(\bar{s}_j,\bar{a}_j,\bar{s}_t,\bar{a}_t\right) &= \mathbb{P}\left(\bar{s}_t,\bar{a}_t|\bar{s}_j,\bar{a}_j\right)\mathbb{P}\left(\bar{s}_j,\bar{a}_j\right)\\
&= \mathbb{P}\left(\bar{s}_{2t-j},\bar{a}_{2t-j}|\bar{s}_t,\bar{a}_t\right)\mathbb{P}\left(\bar{s}_t,\bar{a}_t\right) = \mathbb{P}\left(\bar{s}_t,\bar{a}_t,\bar{s}_{2t-j},\bar{a}_{2t-j}\right).
\end{aligned}
\tag{175}
$$

$\square$

*Proof of Lemma 6.* Define $b_\omega = \mathbb{E}\left[\sum_{j=0}^k \phi_\omega^\top(s_0,a_0)(R(s_j,a_j) - J(\omega))|(s_0,a_0)\sim D_{\pi_\omega},\pi_\omega\right]$. Recall the definition of $A_\omega$ in Equation (13). Then, the solution to Equation (4) can be written as

$$
\theta_\omega^* = A_\omega^{-1}b_\omega.
\tag{176}
$$

First, $b_\omega$ can be bounded as follows:

$$
\begin{aligned}
\|b_\omega\|_2 &= \left\|\mathbb{E}\left[\sum_{j=0}^k \phi_\omega^\top(s_0,a_0)(R(s_j,a_j) - J(\omega))|(s_0,a_0)\sim D_{\pi_\omega},\pi_\omega\right]\right\|_2\\
&= \left\|\sum_{j=0}^k \mathbb{E}\left[\phi_\omega^\top(s_0,a_0)(R(s_j,a_j) - J(\omega))|(s_0,a_0)\sim D_{\pi_\omega},\pi_\omega\right]\right\|_2\\
&= \left\|\sum_{j=0}^k \mathbb{E}\left[\phi_\omega^\top(s_0,a_0)\left(R(s_j,a_j) - \mathbb{E}_{D_{\pi_\omega}}\left[R(s,a)\right]\right)|(s_0,a_0)\sim D_{\pi_\omega},\pi_\omega\right]\right\|_2\\
&\overset{(a)}{\le} \sum_{j=0}^k C_\phi R_{\max}\mathbb{E}\left[\|D_{\pi_\omega} - \mathbb{P}\left(s_j,a_j|s_0,a_0,\pi_\omega\right)\|_{\mathcal{TV}}|(s_0,a_0)\sim D_{\pi_\omega}\right]\\
&\le C_\phi R_{\max}\sum_{j=0}^k m\rho^k \le \frac{C_\phi R_{\max}m}{1-\rho}.
\end{aligned}
\tag{177}
$$

From the following equation:

$$
\theta_\omega^{*\top}A_\omega\theta_\omega^* = \theta_\omega^{*\top}b_\omega = \left(\theta_\omega^{*\top}b_\omega\right)^\top = \theta_\omega^{*\top}A_\omega^\top\theta_\omega^*,
\tag{178}
$$

it holds that

$$
\lambda_{\max}\left(\frac{A_\omega + A_\omega^\top}{2}\right)\|\theta_\omega^*\|_2^2 \ge \theta_\omega^{*\top}\frac{A_\omega + A_\omega^\top}{2}\theta_\omega^* = \theta_\omega^{*\top}b_\omega \ge -\|\theta_\omega^*\|_2\|b_\omega\|_2.
\tag{179}
$$

Thus, we can bound $\theta_\omega^*$ as follows:

$$
\begin{aligned}
\|\theta_\omega^*\|_2 &\overset{(a)}{\le} -\frac{1}{\lambda_{\max}\left(\frac{A_\omega + A_\omega^\top}{2}\right)}\|b_\omega\|_2\\
&\overset{(b)}{\le} -\frac{1}{\lambda_{\max}\left(\frac{A_\omega + A_\omega^\top}{2}\right)}\frac{mC_\phi R_{\max}}{1-\rho}\\
&\overset{(c)}{\le} \frac{1}{\lambda_{\min} - dC_\phi^2 m\rho^k}\frac{mC_\phi R_{\max}}{1-\rho} = \frac{mC_\phi R_{\max}}{\bar{\lambda}_{\min}(1-\rho)},
\end{aligned}
\tag{180}
$$

where $(a)$ follows from that $\frac{A_\omega + A_\omega^\top}{2}$ is negative definite, $(b)$ follows from Equation (177) and $(c)$ follows from Lemma 3. $\square$

# E  SYMBOL REFERENCE

## E.1  CONSTANTS

| Constant (Expression) | First Appearance | |
|---|---|---|
| $B = \frac{R_{\max}C_\phi}{(1-\rho)\left(\lambda_{\min}-C_\phi^2 dm\rho^k\right)}$ | P8 | |
| $c_\alpha = \frac{C_\Theta}{\lambda_{\min}}$ | Proposition 5 | expression in Equation (74) |
| $c_\eta = \frac{B^2 k^2 C_\phi^6}{(\lambda_{\min})^2}$ | Proposition 5 | expression in Equation (74) |
| $C_{\text{gap}} = C_\phi^2 B + \frac{C_\phi R_{\max}}{1-\rho}$ | Proposition 4 | |
| $C_\infty$ | Assumption 3 | |
| $C_\pi$ | Assumption 2 | |
| $C_\phi$ | Assumption 2 | |
| $L_\phi$ | Assumption 2 | |
| $L_\delta$ | Assumption 2 | |
| $\lambda_{\min}$ | P7 | |

Table 3: Constants in Main Text

| Constant (Expression) | First Appearance |
|---|---|
| $C_J = C_\phi^2 \left( B + C_{\text{gap}} \frac{m\rho^k}{\lambda_{\min}} \right)$ | Lemma 1 |
| $C^M = \frac{C_\Theta}{\lambda_{\min}^2} + \frac{1}{2}B^2$ | Equation (116) |
| $C_\Theta = \frac{C_J}{\lambda_{\min}^2} \left( 2C_\phi L_\phi + C_\phi^2 L_\pi \right) + \frac{L_J}{\lambda_{\min}}$ | Lemma 2 |
| $L_J = \frac{mR_{\max}}{1-\rho} \left( 4L_\pi C_\phi + L_\phi \right)$ | Lemma 1 |
| $L_\pi = \frac{1}{2}C_\pi \left( 1 + \lceil \log m^{-1} \rceil + \frac{1}{1-\rho} \right)$ | Lemma 1 |
| $L_\Theta$ | Lemma 10 |
| $\lambda_{\text{m}}$ | below Equation (86) |
| $\bar{\lambda}_{\min} = \lambda_{\min} - dC_\phi^2 m\rho^k$ | Lemma 9 |
| $U_\delta = 2R_{\max} + 2C_\phi B$ | Lemma 5 |

Table 4: Constants in Appendix

## E.2  VARIABLES

| Variable | Appearance | Order (set $\alpha_t \equiv \alpha, \beta_t \equiv \beta, \gamma_t \equiv \gamma$) |
|---|---|---|
| $G_t^\delta$ | Lemma 5 | $\mathcal{O}(k^3\alpha + k^3\beta + k(m\rho^k))$ |
| $G_t^\omega$ | Lemma 7 | $\mathcal{O}\left((m\rho^k)\beta + k^2\beta^2\right)$ |
| $G_t^\eta$ | Lemma 8 | $\mathcal{O}\left((m\rho^k)\gamma + \beta^2 + k^2\beta\gamma + k\gamma^2\right)$ |
| $G_t^\theta$ | Lemma 9 | $\mathcal{O}\left(k(m\rho^k)\alpha + k(m\rho^k)\beta + \frac{(m\rho^k)^2}{\beta} + k^3\alpha^2 + k^3\alpha\beta + k^2\beta^2\right)$ |
| $\widetilde{G}_t^\eta$ | Lemma 11 | $\mathcal{O}\left((m\rho^k)\gamma + \beta^2 + k^2\beta\gamma + k\gamma^2\right)$ |
| $\widetilde{G}_t^\theta$ | Lemma 12 | $\mathcal{O}\left(k(m\rho^k)\alpha + k(m\rho^k)\beta + \frac{(m\rho^k)^2}{\beta} + k^3\alpha^2 + k^3\alpha\beta + k^2\beta^2\right)$ |
| $\widetilde{G}_t^\omega$ | Equation (117) | $\mathcal{O}\left((m\rho^k)\beta + \beta^2)\right)$ |
| $q = \frac{\bar{\lambda}_{\min}\alpha}{2}$ | Equation (68) | $\mathcal{O}\left(\alpha\right)$ |
| $\hat{t} = \left\lceil \frac{1}{q} \log T \right\rceil$ | Equation (119) | $\mathcal{O}\left(\frac{\log T}{\alpha}\right)$ |
| $\widetilde{T} = \left\lceil \frac{T}{\hat{t} \log T} \right\rceil \hat{t}$ | Equation (120) | $\mathcal{O}\left(\frac{T}{\log T}\right)$ |
| $M_t$ | Equation (116) | - |

Table 5: Variables in Appendix