# OpenReview forum: "Non-Asymptotic Analysis for Single-Loop (Natural) Actor-Critic with Compatible Function Approximation"
_ICLR.cc/2024/Conference — Submitted to ICLR 2024_

### Official Review · Reviewer_2Umy · 2023-10-26

**Soundness:** 3 good
**Presentation:** 3 good
**Contribution:** 3 good
**Rating:** 8
**Confidence:** 2

**Summary:**

The paper provides an improved analysis of the actor-critic algorithm. It introduces compatible function approximation to compute the value function, which eliminates the critic approximation error, while keeping the state-of-the-art rate in the sample complexity.

**Strengths:**

The paper is easy to follow. The main idea is clearly stated and makes sense. The paper delivers interesting results, which justify the novelty of the work.

**Weaknesses:**

1. Since the compatible function approximation (5) is not a straightforward fact, it's better to restate this conclusion in the form of a proposition. It could also be helpful if a quick proof can be attached in the appendix.

2. A high-level roadmap of the proof is appreciated. It could make it easier to follow the proof as it's already notation heavy.

**Questions:**

I didn't read the proof in detail. But intuitively, in the previous works, the source of $\varepsilon_{\textrm{critic}}$ is the previous proof needs to bound a term like $\Vert\nabla_\omega J(\pi_\omega) - \mathbb{E}[Q_\theta^{\pi_\omega}\nabla_\omega\log\pi_\omega(a\vert s)]\Vert^2$, which is the difference between the true gradient and the estimated gradient using the value function class. However, due to the authors' setting, this term automatically becomes 0. In this case, why can't one reuse the previous proofs, just replacing this term by 0? Does the introduction of compatible function approximation leads to new difficulties that need additional theoretical novelty to overcome them?

---

> ### Author Response · Authors · 2023-11-18
> **Part 1**
>
> **Weaknesses 1: Restate compatible function approximation conclusion in eq. (5) in the form of a proposition and provide the proof.**
>
> Thanks for the suggestion. We have added the proof of eq. (5) in the revision. For convenience, we also include the proposition and the proof here.
>
>
> Proposition 1: With compatible function approximation, the policy gradient $\nabla J(\pi_\omega)$ can be rewritten as:
> $$
>  \nabla J(\pi_\omega) =\mathbb E_{D_{\pi_\omega}}[ \nabla \log \pi_\omega(a|s)Q^{\pi_\omega}(s,a)]=
>  \mathbb E_{D_{\pi_\omega}}[\phi_\omega(s,a)(\phi^\top_\omega(s,a)\bar\theta_\omega^*)].$$
>  The natural policy gradient can be written as:
> $$\widetilde{\nabla} J(\pi_\omega)=\bar \theta^*_\omega.$$
>
> **Proof:**
> By eq. (4), $\bar \theta^*_\omega$ is the minimizer of  eq. (4) and therefore it should satisfy that
> $$
> \mathbb E_{D_{\pi_\omega}}[(Q^{\pi_\omega}(s,a)-\phi^\top_\omega(s,a)\bar \theta_\omega^*)\phi_\omega(s,a) ]=0.
> $$
> This further implies that
> $$\mathbb E_{D_{\pi_\omega}}[Q^{\pi_\omega}(s,a)\phi_\omega(s,a) ]=\mathbb E_{D_{\pi_\omega}}[\phi_\omega(s,a)\phi^\top_\omega(s,a)\bar \theta_\omega^*].
> $$
> Now we consider the policy gradient $\nabla J(\pi_\omega)$, and it  can be shown that
> $$
> \nabla J(\pi_\omega) =\mathbb E_{D_{\pi_\omega}}[ \nabla \log \pi_\omega(a|s)Q^{\pi_\omega}(s,a)]=
>  \mathbb E_{D_{\pi_\omega}}[\phi_\omega(s,a)
>  (\phi^\top_\omega(s,a)\bar\theta_\omega^*)].
>  $$
>
> Furthermore, for the natural policy gradient, we have that $$\widetilde{\nabla} J(\pi_\omega)= F_{\omega}^{-1} \mathbb E_{D_{\pi_\omega}}[\phi_\omega(s,a)\phi_\omega(s,a)^\top{\bar\theta}^*_\omega]=F_\omega^{-1} \mathbb E_{D_{\pi_\omega}}[\phi_\omega(s,a)\phi_\omega(s,a)^\top   ]\bar \theta^*_\omega=\bar \theta^*_\omega.
> $$

---

> ### Author Response · Authors · 2023-11-18
> **Part 2**
>
> **Weaknesses 2: high-level roadmap of the proof.**
> Here we provide a proof sketch for the NAC algorithm to highlight major challenges and our technical novelties. The analysis of NAC contains of most major technical novelty in the AC analysis.
>
>
> **Proof Sketch:**
> For simplicity of presentation, we set $\alpha_t=\alpha,\beta_t=\beta,\gamma_t=\gamma$, $\hat t=\lceil \frac{3\log T}{\bar \lambda_{\min}\alpha}\rceil$ and
> $\widetilde T=\hat t\lceil \frac{T}{\hat t \log T} \rceil$.
>  We denote by $\pi^*$ the global optimal policy and $M_t=\mathbb E[||\theta_{t}-\theta^*_{t}||^2_2]+ \mathbb E[(\eta_t-J(\omega_t))^2]$ the sum of the tracking error and the estimation error of the average reward. Denote by $D(\omega_t)=KL(\pi^*|\pi_t)$ the KL divergence between policy $\pi^*$ and $\pi_t$.
>
>
> **Step 1 (Error decomposition):**
> According to the smoothness property of $D(\omega)$ with respect to $\omega$, we bound the performance gap between the current policy and the optimal policy (optimality gap) as follows:
> \begin{align}
>     \frac{1}{\widetilde T}\sum_{j=t}^{t+\widetilde T-1}\mathbb E[J(\pi^*)-J(\omega_j)]
>      \leq \frac{D(\omega_{t+\widetilde T})-D(\omega_t)}{\widetilde T \beta }+ \mathcal{O}\left(\sqrt{\frac{1}{\widetilde T}\sum_{j=t}^{t+\widetilde T-1} M_j}\right)+\mathcal{O}(C_\infty \sqrt{\varepsilon_{\text{actor}}}+ \beta+m\rho^k).
> \end{align}
>
>
>
>
> **Step 2 (Estimation error in the average reward):** In this step, we analyze estimation error in the average reward:
>  $\eta_t-J(\omega_t)$. We provide a tight characterization of this error:
> \begin{align}
>     \mathbb E[(\eta_{t+1}-J(\omega_{t+1}))^2] \leq (1-\gamma)\mathbb E[(\eta_{t}-J(\omega_{t}))^2] + \mathcal{O}(\beta \mathbb E[||\nabla J(\omega_t)||_2^2])
>     + \mathcal{O}( m\rho^k\gamma +k^2 \gamma^2 +k^2 \beta^2).
> \end{align}
> One of our key novelties lies in that we bound this estimation error using the gradient norm $\mathbb E[||\nabla J(\omega_t)||_2^2]$. The above bound itself is tighter than the existing one in (Wu et al., 2020).
>
>
>
>
> **Step 3 (Tracking error)**:
> In this step, we bound the tracking error in the critic: $||\theta_t-\theta_t^*||_2^2$.
>
> By the TD error step in Algorithm 1, we decompose the term $||\theta_{t+1}-\theta_{t+1}^*||^2_2$ as follows:
>
> $    ||\theta_{t+1}-\theta_{t+1}^*||^2_2 \leq || \theta_t-\theta_{t}^* ||^2_2+ \alpha^2 ||\delta_t z_t||^2_2+|| \theta_{t}^*-\theta_{t+1}^*||_2^2
> $
>
> $+ 2 \alpha\langle\theta_t-\theta_{t}^*, \delta_t z_t \rangle  + 2\alpha \langle \delta_t z_t, \theta_{t}^* - \theta_{t+1}^* \rangle+ 2 \langle\theta_t-\theta_{t}^*, \theta_{t}^*-\theta_{t+1}^* \rangle.$
>
> Another key challenge lies in  how to bound the term
> $\mathbb E[\langle \theta_t-\theta_t^*, \delta_t z_t\rangle ] $.  We develop a novel technique of auxiliary Markov chain to decompose this error into two parts: 1) error due to time-varying feature function and 2) error due to time-varying policy. Specifically, consider the first Markov chain generated from the algorithm:
> $$s_0,a_0 \overset{\pi_0\times P}{\to} s_1,a_1\to...\to s_t,a_t \overset{\pi_t\times P}{\to}s_{t+1},a_{t+1},$$ where at each time $j$, the action is chosen according to $\pi_j$ and the transition kernel is $P$. Here $z_t=\sum_{j=t-k}^t \phi_j(s_j,a_j)$ is the eligibility trace used in the algorithm. It can be seen that in $z_t$, the feature $\phi_j$ changes with $j$, and the distribution of $s_j,a_j$ depends on the time-varying policy $\pi_j$. We then design an auxiliary eligibility trace $\hat z_t=\sum_{j=t-k}^t \phi_t(s_j,a_j)$, where the feature is fixed to be $\phi_t$, and only the the distribution of $s_j,a_j$ depends on the time-varying policy $\pi_j$. To further handle the time-varying distribution of  $s_j,a_j$, we design an auxiliary Markov chain (denoted by $A1$) as follows:
> $$A1: ( s_0,\widetilde  a_0 )\sim \pi_t\overset{\pi_t\times P}{\to} \widetilde s_1,\widetilde a_1\overset{\pi_t\times P}{\to}...\overset{\pi_t\times P}{\to}\widetilde  s_t,\widetilde a_t \overset{\pi_t\times P}{\to}\widetilde s_{t+1},\widetilde a_{t+1},$$ where the action at each time $j$ is always chosen according to a fixed policy $\pi_t$. Based on this auxiliary Markov chain, we introduce another auxiliary eligibility trace $\widetilde  z_t=\sum_{j=t-k}^t \phi_t(\widetilde s_j,\widetilde a_j)$, where it uses a fixed feature $\phi_t$, and samples from this auxiliary Markov chain. Lastly, we design a second auxiliary Markov chain (denoted by $A2$):
> $$A2: (\bar s_0,\bar  a_0)\sim D_t \overset{\pi_t\times P}{\to} \bar s_1,\bar a_1\overset{\pi_t\times P}{\to}...\overset{\pi_t\times P}{\to}\bar  s_t,\bar a_t \overset{\pi_t\times P}{\to}\bar s_{t+1},\bar a_{t+1}$$
> where the only difference between A2 and A1 lies in the initial state distribution. Then we define the last auxiliary eligibility trace as $\bar  z_t=\sum_{j=t-k}^t \phi_t(\bar s_j,\bar a_j)$.

---

> > ### Author Response · Authors · 2023-11-18
> > **Part 2 continued**
> >
> > The difference between $z_t $ and $\hat z_t$ measures the error due to the time-varying compatible feature function. We bound this error using the Lipschitz continuity of the feature function.  The difference between $\hat z_t $ and $\widetilde z_t$ measures the error due to the  time-varying sampling policy.   The difference between $\widetilde z_t $ and $\bar z_t$ measures the error due to the difference between the stationary distribution and the actual distribution of the samples, which can be bounded based on Assumption 1. By such a error decomposition, we can show that
> > \begin{align}
> >     \mathbb E[||\theta_{t+1}-\theta^*_{t+1}||^2_2] \leq (1-\bar \lambda_{\min}\alpha /2)\mathbb E[||\theta_{t}-\theta^*_{t}||^2_2]
> >     +\mathcal{O}(k^2 \alpha \mathbb E[(\eta_t-J(\omega_t))^2] )+\mathcal{O}(\beta\mathbb E [||\nabla J(\omega_t)||_2^2])+ \mathcal{O}(k^2 \alpha^2+k^3\beta^2+m\rho^k\alpha ) .
> > \end{align}
> >
> > **Step 4 (Bound on gradient):** As we can see from Steps 2 and 3, we bound the estimation error of the average reward and the tracking error using the gradient norm $\|\nabla J(\omega_t)\|_2^2$. Therefore,  in order to derive the tightest bound, we further develop a novel bound on the gradient norm $\|\nabla J(\omega_t)\|_2^2$. Note that the idea is novel as it serves as a pivotal link connecting the analysis of the tracking error/estimation error in the average reward and the optimality gap. Specifically, we bound the gradient norm using
> > the estimation error in the average reward and tracking error. By the smoothness of $J(\omega)$, we have that
> >
> > $$
> >     \frac{1}{\widetilde T}\sum_{j=t}^{t+\widetilde T-1}\mathbb E [ ||\nabla J(\omega_j) ||_2^2]
> > $$
> >
> > $$ \leq 2C_\phi \frac{J(\omega^*)- E [J(\omega_{t})]}{\beta \widetilde T}+ \mathcal O\left(\frac{1}{\widetilde T}\sum_{j=t}^{t+\widetilde T-1}\mathbb E[||\theta_j-\theta_j^*||_2^2]\right) +\mathcal{O}(m\rho^k+\beta).
> > $$
> >
> > We also note that we bound the gradient norm using the optimality gap, and this is of great importance to establish the tight bound in this paper. In previous works, this  term $\mathbb E [J(\omega_{t+\widetilde T})]- E [J(\omega_{t})] $ is bounded by a constant, and thus the overall complexity is not as tight.
> >
> > **Step 5:** Combining steps 1-4, we conclude the proof.

---

> ### Author Response · Authors · 2023-11-19
> **Part 3**
>
> **Q1:  Why can't one reuse the previous proofs, just replacing the policy gradient bias term by 0?**
>
> In previous AC works, e.g., (Wu et al.,2020) (Xu et al., 2020), the error in the critic is bounded by $$\varepsilon_{\text{approx.}}=\sqrt{\mathbb E[(Q_{\pi_\omega}(s,a)-\phi^\top(s,a) \theta^*_\omega)^2]},$$ where the $\theta_\omega^*$ is the limiting point of the critic under the policy $\pi_\omega$. Their analysis bound the bias in the gradient using the gap between the true $Q$ function and the critic output, $\varepsilon_{\text{approx.}}$, and this gap is typically non-zero. Their analysis are based on the implicit assumption that $\varepsilon_{\text{approx.}}$ will be small, and thus the critic returns an accurate estimation of the true $Q$ function.
>
> We would like to highlight that using compatible function approximation, we do not need $\varepsilon_{\text{approx.}}$ to be small. This term $\varepsilon_{\text{approx.}}$ could be non-zero, however, by eq. (3), the policy gradient will still be unbiased.
> In other words, we do not need to get the exact $Q$ from the critic, as long as the critic solves eq. (2), the gradient will be unbiased using compatible function approximation.  Therefore, the analysis is different, and cannot be obtained by simply letting $\varepsilon_{\text{approx.}}=0$.
>
> Yue Frank Wu, Weitong Zhang, Pan Xu, and Quanquan Gu. "A finite-time analysis of two time-scale actor-critic methods." Advances in Neural Information Processing Systems, 33:17617–17628, 2020
>
> Xu, Tengyu, Zhe Wang, and Yingbin Liang. "Non-asymptotic convergence analysis of two time-scale (natural) actor-critic algorithms." arXiv preprint arXiv:2005.03557 (2020).
>
> **Q2:  What are the new difficulties due to compatible function approximation?**
>
> As we discussed in the proof sketch (provided above), the key challenge lies in the time-varying feature function due to the changing policy. Our key novelty lies in the design of auxiliary Markov chains and auxiliary eligibility trace as in Step 3 in the proof sketch.

---

### Official Review · Reviewer_XG7r · 2023-10-31

**Soundness:** 3 good
**Presentation:** 3 good
**Contribution:** 3 good
**Rating:** 8
**Confidence:** 4

**Summary:**

The authors develop tightest non-asymptotic convergence bounds for both Actor-critic (AC) and natural AC (NAC) algorithms with compatible function approximation. The major technical novelty lies in analyzing the stochastic bias due to policy-dependent and time-varying compatible function approximation in the critic, and handling the non-ergodicity of the MDP due to the single Markovian sample trajectory. This is a nice theoretical contribution for the area of reinforcement learning.

**Strengths:**

The authors develop tightest non-asymptotic convergence bounds for both Actor-critic (AC) and natural AC (NAC) algorithms with compatible function approximation. The major technical novelty lies in analyzing the stochastic bias due to policy-dependent and time-varying compatible function approximation in the critic, and handling the non-ergodicity of the MDP due to the single Markovian sample trajectory. This is a nice theoretical contribution for the area of reinforcement learning.

**Weaknesses:**

There is no discussion on the weakness or limitation of theoretical results, even in the conclusion part.

**Questions:**

What are the limitations of the analysis presented in this paper? How to verify the assumptions made in practice?

---

> ### Author Response · Authors · 2023-11-17
>
> **Weaknesses and Questions:  limitations of the analysis presented in this paper and how to verify the assumptions made in practice**
>
>
> I. As a theoretical analysis work, possible limitation of our work lies in whether the assumptions are reasonable and can be verified.
>
> Assumption 1 of uniform ergodicity  is a widely used assumption in RL analysis. A Markov chain is uniformly ergodic if it is irreducible (i.e., possibly get to any state from any state) and aperiodic (Levin \& Peres 2017). Now consider an environment for which there exists a policy that can map any state to any  state with nonzero probability (i.e., irreducibility holds) and gets back to the same state aperiodically. Such environments are commonly encountered in practice. Then for any $\omega$, as long as the policy  explores (i.e., with non-zero probability to take any action at any state), the induced Markov chain remains to be irreducible and aperiodic, and is hence ergodic. Therefore, such an assumption is realistic, and is guaranteed for aforementioned environments. We also agree that if Assumption 1 does not hold, new techniques will be needed to understand the non-asymptotic error bounds of many RL algorithms.
>
> Assumption 2 can be satisfied if we use some popular policy parameterization, such as softmax,  neural network with analytic activation functions e.g., logistic, tanh, and softplus.  These results can be found in (Du et al., 2019; Miyato et al. 2018; Neyshabur, 2017).
>
> Assumption 3 can be satisfied if we carefully design the policy parameterization so that it is sufficiently explorative, e.g., softmax and $\epsilon$-greedy policy.
>
> II. Another possible limitation in the algorithm and analysis is the use of the projection (Line 9 in Algorithm 1). We conjecture that this projection may not be necessarily needed to guarantee convergence and to derive the non-asymptotic bounds (see e.g., Srikant \& Ying 2019). However, this projection helps to simplify the proof, and without which the proof will be rather cumbersome.
>
> David A. Levin and Yuval Peres. Markov chains and mixing times. Vol. 107. American Mathematical Soc., 2017.
>
> Simon Du, Jason Lee, Haochuan Li, Liwei Wang, and Xiyu Zhai. "Gradient descent finds global minima of deep neural networks." International conference on machine learning. PMLR, 2019.
>
> Takeru Miyato, Toshiki Kataoka, Masanori Koyama, and Yuichi Yoshida. "Spectral Normalization for Generative Adversarial Networks." International Conference on Learning Representations. 2018.
>
> Neyshabur, Behnam. "Implicit regularization in deep learning." arXiv preprint arXiv:1709.01953 (2017).
>
> Srikant, Rayadurgam, and Lei Ying. "Finite-time error bounds for linear stochastic approximation andtd learning." Conference on Learning Theory. PMLR, 2019.

---

> > ### Comment · Reviewer_XG7r · 2023-11-22
> >
> > Thanks authors for pointing out the potential limiations and the ways to verify key assumptions. I would like to raise the socre to 8!

---

### Official Review · Reviewer_VajU · 2023-11-01

**Soundness:** 4 excellent
**Presentation:** 1 poor
**Contribution:** 3 good
**Rating:** 6
**Confidence:** 3

**Summary:**

This paper provides tight analyses for non-asymptotic convergence bounds of actor-critic and natural actor-critic algorithms on single-trajectory single-loop online RL problems.

**Strengths:**

I think this work really pushes the RL community research efforts further by answering:

> can we give tight analyses for non-asymptotic convergence bounds of actor-critic and natural actor-critic algorithms on single-trajectory single-loop online RL problems?

The main contribution of improvement of results is worthy for publication.

**Weaknesses:**

Firstly, I only took a closer look at AC sample complexity analyses (Lemma 7, 8, 10, and Proposition 6) and everything seemed alright.

I have only a few weakness for this work as follows:

- The main paper writing needs to be improved. Yes, the soundness of this paper is excellent. But right from Section 1.2, it's just about "getting sota results compared to other works". The main analyses idea in this work compared to others which helps improve the bounds are missing. To be honest, I am only familiar with high level analyses of (Chen et al 2022) and few generic two-timescale algorithms, which shares the AC analyses in this work. But I am totally missing the improvement bounds provided here! Just providing one line (below quote) as the core idea for the whole 50 pages analyses paper, does not provide any useful information.
> In this paper, we design a novel approach to explicitly bound this error. The central idea is to construct an auxiliary eligibility trace with fixed feature to approximate the eligibility trace with time-varying feature (in the critic, we use k-step TD with compatible function approximation).

So please write more discussions and details for the main results and proof ideas. I understand this is presentation is limited by page restriction, but it can be done through a thorough writing process. For example, AC results can be pushed to appendix looking at the generality of NAC, or the other way around.

- This paper considers average MDP formulation. Most of the comparisons in Table 1 and 2 are in the discounted setting. Please mention these important differences and make fair comparisons by arguing the connection between the two settings. I am not sure if this distinction is what is helping to reach current bounds. Thus my point above is super critical for the quality.

I am open to discussions with the authors and reviewers to make sure the work quality matches the score, which I believe so at this point, but a potential reach to 8 definitely exists. All the best for future decisions!

**Questions:**

-na-

---

> ### Author Response · Authors · 2023-11-18
> **Part 1**
>
> **Weaknesses1:  Details of this paper's novelty.**
> Here we provide a proof sketch for the NAC algorithm to highlight major challenges and our technical novelties. The analysis of NAC contains of most major technical novelty in the AC analysis.
>
>
> **Proof Sketch:**
> For simplicity of presentation, we set $\alpha_t=\alpha,\beta_t=\beta,\gamma_t=\gamma$, $\hat t=\lceil \frac{3\log T}{\bar \lambda_{\min}\alpha}\rceil$ and
> $\widetilde T=\hat t\lceil \frac{T}{\hat t \log T} \rceil$.
>  We denote by $\pi^*$ the global optimal policy and $M_t=\mathbb E[||\theta_{t}-\theta^*_{t}||^2_2]+ \mathbb E[(\eta_t-J(\omega_t))^2]$ the sum of the tracking error and the estimation error of the average reward. Denote by $D(\omega_t)=KL(\pi^*|\pi_t)$ the KL divergence between policy $\pi^*$ and $\pi_t$.
>
>
> **Step 1 (Error decomposition):**
> According to the smoothness property of $D(\omega)$ with respect to $\omega$, we bound the performance gap between the current policy and the optimal policy (optimality gap) as follows:
> \begin{align}
>     \frac{1}{\widetilde T}\sum_{j=t}^{t+\widetilde T-1}\mathbb E[J(\pi^*)-J(\omega_j)]
>      \leq \frac{D(\omega_{t+\widetilde T})-D(\omega_t)}{\widetilde T \beta }+ \mathcal{O}\left(\sqrt{\frac{1}{\widetilde T}\sum_{j=t}^{t+\widetilde T-1} M_j}\right)+\mathcal{O}(C_\infty \sqrt{\varepsilon_{\text{actor}}}+ \beta+m\rho^k).
> \end{align}
>
>
>
>
> **Step 2 (Estimation error in the average reward):** In this step, we analyze estimation error in the average reward:
>  $\eta_t-J(\omega_t)$. We provide a tight characterization of this error:
> \begin{align}
>     \mathbb E[(\eta_{t+1}-J(\omega_{t+1}))^2] \leq (1-\gamma)\mathbb E[(\eta_{t}-J(\omega_{t}))^2] + \mathcal{O}(\beta \mathbb E[||\nabla J(\omega_t)||_2^2])
>     + \mathcal{O}( m\rho^k\gamma +k^2 \gamma^2 +k^2 \beta^2).
> \end{align}
> One of our key novelties lies in that we bound this estimation error using the gradient norm $\mathbb E[||\nabla J(\omega_t)||_2^2]$. The above bound itself is tighter than the existing one in (Wu et al., 2020).
>
>
>
>
> **Step 3 (Tracking error)**:
> In this step, we bound the tracking error in the critic: $||\theta_t-\theta_t^*||_2^2$.
>
> By the TD error step in Algorithm 1, we decompose the term $||\theta_{t+1}-\theta_{t+1}^*||^2_2$ as follows:
>
> $    ||\theta_{t+1}-\theta_{t+1}^*||^2_2 \leq || \theta_t-\theta_{t}^* ||^2_2+ \alpha^2 ||\delta_t z_t||^2_2+|| \theta_{t}^*-\theta_{t+1}^*||_2^2
> $
>
> $+ 2 \alpha\langle\theta_t-\theta_{t}^*, \delta_t z_t \rangle  + 2\alpha \langle \delta_t z_t, \theta_{t}^* - \theta_{t+1}^* \rangle+ 2 \langle\theta_t-\theta_{t}^*, \theta_{t}^*-\theta_{t+1}^* \rangle.$
>
> Another key challenge lies in  how to bound the term
> $\mathbb E[\langle \theta_t-\theta_t^*, \delta_t z_t\rangle ] $.  We develop a novel technique of auxiliary Markov chain to decompose this error into two parts: 1) error due to time-varying feature function and 2) error due to time-varying policy. Specifically, consider the first Markov chain generated from the algorithm:
> $$s_0,a_0 \overset{\pi_0\times P}{\to} s_1,a_1\to...\to s_t,a_t \overset{\pi_t\times P}{\to}s_{t+1},a_{t+1},$$ where at each time $j$, the action is chosen according to $\pi_j$ and the transition kernel is $P$. Here $z_t=\sum_{j=t-k}^t \phi_j(s_j,a_j)$ is the eligibility trace used in the algorithm. It can be seen that in $z_t$, the feature $\phi_j$ changes with $j$, and the distribution of $s_j,a_j$ depends on the time-varying policy $\pi_j$. We then design an auxiliary eligibility trace $\hat z_t=\sum_{j=t-k}^t \phi_t(s_j,a_j)$, where the feature is fixed to be $\phi_t$, and only the the distribution of $s_j,a_j$ depends on the time-varying policy $\pi_j$. To further handle the time-varying distribution of  $s_j,a_j$, we design an auxiliary Markov chain (denoted by $A1$) as follows:
> $$A1: ( s_0,\widetilde  a_0 )\sim \pi_t\overset{\pi_t\times P}{\to} \widetilde s_1,\widetilde a_1\overset{\pi_t\times P}{\to}...\overset{\pi_t\times P}{\to}\widetilde  s_t,\widetilde a_t \overset{\pi_t\times P}{\to}\widetilde s_{t+1},\widetilde a_{t+1},$$ where the action at each time $j$ is always chosen according to a fixed policy $\pi_t$. Based on this auxiliary Markov chain, we introduce another auxiliary eligibility trace $\widetilde  z_t=\sum_{j=t-k}^t \phi_t(\widetilde s_j,\widetilde a_j)$, where it uses a fixed feature $\phi_t$, and samples from this auxiliary Markov chain. Lastly, we design a second auxiliary Markov chain (denoted by $A2$):
> $$A2: (\bar s_0,\bar  a_0)\sim D_t \overset{\pi_t\times P}{\to} \bar s_1,\bar a_1\overset{\pi_t\times P}{\to}...\overset{\pi_t\times P}{\to}\bar  s_t,\bar a_t \overset{\pi_t\times P}{\to}\bar s_{t+1},\bar a_{t+1}$$
> where the only difference between A2 and A1 lies in the initial state distribution. Then we define the last auxiliary eligibility trace as $\bar  z_t=\sum_{j=t-k}^t \phi_t(\bar s_j,\bar a_j)$.

---

> > ### Author Response · Authors · 2023-11-18
> > **Part 1 continued**
> >
> > The difference between $z_t $ and $\hat z_t$ measures the error due to the time-varying compatible feature function. We bound this error using the Lipschitz continuity of the feature function.  The difference between $\hat z_t $ and $\widetilde z_t$ measures the error due to the  time-varying sampling policy.   The difference between $\widetilde z_t $ and $\bar z_t$ measures the error due to the difference between the stationary distribution and the actual distribution of the samples, which can be bounded based on Assumption 1. By such a error decomposition, we can show that
> > \begin{align}
> >     \mathbb E[||\theta_{t+1}-\theta^*_{t+1}||^2_2] \leq (1-\bar \lambda_{\min}\alpha /2)\mathbb E[||\theta_{t}-\theta^*_{t}||^2_2]
> >     +\mathcal{O}(k^2 \alpha \mathbb E[(\eta_t-J(\omega_t))^2] )+\mathcal{O}(\beta\mathbb E [||\nabla J(\omega_t)||_2^2])+ \mathcal{O}(k^2 \alpha^2+k^3\beta^2+m\rho^k\alpha ) .
> > \end{align}
> >
> > **Step 4 (Bound on gradient):** As we can see from Steps 2 and 3, we bound the estimation error of the average reward and the tracking error using the gradient norm $\|\nabla J(\omega_t)\|_2^2$. Therefore,  in order to derive the tightest bound, we further develop a novel bound on the gradient norm $\|\nabla J(\omega_t)\|_2^2$. Note that the idea is novel as it serves as a pivotal link connecting the analysis of the tracking error/estimation error in the average reward and the optimality gap. Specifically, we bound the gradient norm using
> > the estimation error in the average reward and tracking error. By the smoothness of $J(\omega)$, we have that
> >
> > $$
> >     \frac{1}{\widetilde T}\sum_{j=t}^{t+\widetilde T-1}\mathbb E [ ||\nabla J(\omega_j) ||_2^2]
> > $$
> >
> > $$ \leq 2C_\phi \frac{J(\omega^*)- E [J(\omega_{t})]}{\beta \widetilde T}+ \mathcal O\left(\frac{1}{\widetilde T}\sum_{j=t}^{t+\widetilde T-1}\mathbb E[||\theta_j-\theta_j^*||_2^2]\right) +\mathcal{O}(m\rho^k+\beta).
> > $$
> >
> > We also note that we bound the gradient norm using the optimality gap, and this is of great importance to establish the tight bound in this paper. In previous works, this  term $\mathbb E [J(\omega_{t+\widetilde T})]- E [J(\omega_{t})] $ is bounded by a constant, and thus the overall complexity is not as tight.
> >
> > **Step 5:** Combining steps 1-4, we conclude the proof.

---

> ### Author Response · Authors · 2023-11-18
> **Part 2**
>
> **Weaknesses 2: Fair comparisons with discounted MDPs, and whether our approach can be generalized to the discounted setting**
>
> We agree with the reviewer that the average reward setting and the discounted setting are different. In general, the  average MDP setting is more challenging to analyze than the discounted MDP setting due to the lack of the discount factor and thus the contraction property. It is possible to extend our methodology to the discounted MDP setting. For example, in the critic, we could use TD($\lambda$) with a large enough $\lambda$ to solve eq.(4). Our approach of designing auxiliary Markov chain and auxiliary eligibility trace could also be generalized to the discounted setting. It is also of interest to investigate in details the non-asymptotic error bound for the discounted MDP setting.

---

> > ### Comment · Reviewer_VajU · 2023-11-21
> > **Ack**
> >
> > Thank you for the detailed response. I do think the authors have improved the manuscript compared to the pre-rebuttal stage and also seem to have addressed the major concerns of all reviewers. I will update my score to 8 after authors-reviewers discussions since I only verified the newly posed auxiliary MC arguments briefly.

---

### Official Review · Reviewer_CZ8K · 2023-11-01

**Soundness:** 3 good
**Presentation:** 3 good
**Contribution:** 3 good
**Rating:** 6
**Confidence:** 3

**Summary:**

This paper presents a non-asymptotic analysis for single-loop Actor-Critic (AC) and Natural Actor-Critic (NAC) algorithms with compatible function approximation. The authors provide tight convergence bounds for both AC and NAC algorithms, eliminating the non-diminishing constant term $\varepsilon_{critic}$ from the error bounds while maintaining the best known sample complexities. The paper focuses on the challenging single-loop setting with a single Markovian sample trajectory and analyzes the stochastic bias due to policy-dependent and time-varying compatible function approximation in the critic.

**Strengths:**

The paper is well-structured and presents a clear analysis of the convergence properties of AC and NAC algorithms with compatible function approximation. The authors develop the tightest non-asymptotic convergence bounds for both AC and NAC algorithms with compatible function approximation. For the AC algorithm, they achieve the best sample complexity of $O(\varepsilon^{-2})$ with a reduced error from $\varepsilon + \varepsilon_{critic}$ to $\varepsilon$. For the NAC algorithm, they achieve the best known sample complexity of $O(\varepsilon^{-3})$ with a reduced error of $\varepsilon+\sqrt{\varepsilon_{actor}}$.

**Weaknesses:**

1. It is unclear how the result in the paper compared with other non-asymptotic analyses with compatible function approximation. Also, it would be nice to demonstrate the technical challenges in combining (natural) actor-critic with compatible function approximation in a more detailed way.

2. $\varepsilon_{critic}$ can be small when using a neural function approximator, so it is unclear how important it is to eliminate such an error in real-world scenarios. While this is a theoretical work, it would be nice to include some experiments demonstrating the importance of eliminating $\varepsilon_{critic}$.

**Questions:**

See the weakness section above.

---

> ### Author Response · Authors · 2023-11-18
> **part 1**
>
> **Q 1.a: how the result in the paper compared with other non-asymptotic analyses with compatible function approximation.**
>
> The asymptotic convergence of actor-critic using compatible function approximation was derived in (Konda \& Tsitsiklis 2003), and later
> (Peters \& Schaal 2008) combined the idea of compatible function approximation with NAC, but did not provide any non-asymptotic convergence proof. Recently, non-asymptotic analysis of compatible function approximation for NAC was developed in (Agarwal et al. 2021) (Cayci et al. 2022) and (Wang et al. 2019). However, in these recent studies, they only employ the advantage that using compatible function approximation eliminates the need to estimate the Fisher information matrix and its inverse (as shown in eq. (8)). However, their critic design fails to obtain a solution to eq. (4), and therefore, their non-asymptotic error bounds still include the critic approximation error $\varepsilon_{\text{critic}}$. Our critic  employs the $k$-step TD algorithm so that the critic converges to a solution to eq. (4). In this way, we could achieve a zero critic approximation error. Moreover, these studies are not for the single-loop setting, whereas in our paper, we focus on the single loop setting, which is more challenging to analyze.
>
>
> Konda, Vijay R., and John N. Tsitsiklis. "Onactor-critic algorithms." SIAM journal on Control and Optimization 42.4 (2003): 1143-1166.
>
> Peters, Jan, and Stefan Schaal. "Natural actor-critic." Neurocomputing 71.7-9 (2008): 1180-1190.
>
> Alekh Agarwal,Sham M Kakade, Jason D Lee, and Gaurav Mahajan. "On the theory of policy gradient methods: Optimality, approximation, and distribution shift." The Journal of Machine Learning Research 22.1 (2021): 4431-4506.
>
> Cayci, Semih, Niao He, and R. Srikant. "Finite-time analysis of entropy-regularized neural natural actor-critic algorithm." arXiv preprint arXiv:2206.00833 (2022).
>
> Lingxiao Wang, Qi Cai, Zhuoran Yang, and Zhaoran Wang. "Neural Policy Gradient Methods: Global Optimality and Rates of Convergence." International Conference on Learning Representations. 2019.
>
> Qi Cai, Zhuoran Yang, Jason D Lee, and Zhaoran Wang. "Neural Temporal Difference and Q Learning Provably Converge to Global Optima." Mathematics of Operations Research (2023).

---

> ### Author Response · Authors · 2023-11-18
> **part 2**
>
> **Q 1.b Technical challenges.**
> Here we provide a proof sketch for the NAC algorithm to highlight major challenges and our technical novelties. The analysis of NAC contains of most major technical novelty in the AC analysis.
>
>
> **Proof Sketch:**
> For simplicity of presentation, we set $\alpha_t=\alpha,\beta_t=\beta,\gamma_t=\gamma$, $\hat t=\lceil \frac{3\log T}{\bar \lambda_{\min}\alpha}\rceil$ and
> $\widetilde T=\hat t\lceil \frac{T}{\hat t \log T} \rceil$.
>  We denote by $\pi^*$ the global optimal policy and $M_t=\mathbb E[||\theta_{t}-\theta^*_{t}||^2_2]+ \mathbb E[(\eta_t-J(\omega_t))^2]$ the sum of the tracking error and the estimation error of the average reward. Denote by $D(\omega_t)=KL(\pi^*|\pi_t)$ the KL divergence between policy $\pi^*$ and $\pi_t$.
>
>
> **Step 1 (Error decomposition):**
> According to the smoothness property of $D(\omega)$ with respect to $\omega$, we bound the performance gap between the current policy and the optimal policy (optimality gap) as follows:
> \begin{align}
>     \frac{1}{\widetilde T}\sum_{j=t}^{t+\widetilde T-1}\mathbb E[J(\pi^*)-J(\omega_j)]
>      \leq \frac{D(\omega_{t+\widetilde T})-D(\omega_t)}{\widetilde T \beta }+ \mathcal{O}\left(\sqrt{\frac{1}{\widetilde T}\sum_{j=t}^{t+\widetilde T-1} M_j}\right)+\mathcal{O}(C_\infty \sqrt{\varepsilon_{\text{actor}}}+ \beta+m\rho^k).
> \end{align}
>
>
>
>
> **Step 2 (Estimation error in the average reward):** In this step, we analyze estimation error in the average reward:
>  $\eta_t-J(\omega_t)$. We provide a tight characterization of this error:
> \begin{align}
>     \mathbb E[(\eta_{t+1}-J(\omega_{t+1}))^2] \leq (1-\gamma)\mathbb E[(\eta_{t}-J(\omega_{t}))^2] + \mathcal{O}(\beta \mathbb E[||\nabla J(\omega_t)||_2^2])
>     + \mathcal{O}( m\rho^k\gamma +k^2 \gamma^2 +k^2 \beta^2).
> \end{align}
> One of our key novelties lies in that we bound this estimation error using the gradient norm $\mathbb E[||\nabla J(\omega_t)||_2^2]$. The above bound itself is tighter than the existing one in (Wu et al., 2020).
>
>
>
>
> **Step 3 (Tracking error)**:
> In this step, we bound the tracking error in the critic: $||\theta_t-\theta_t^*||_2^2$.
>
> By the TD error step in Algorithm 1, we decompose the term $||\theta_{t+1}-\theta_{t+1}^*||^2_2$ as follows:
>
> $    ||\theta_{t+1}-\theta_{t+1}^*||^2_2 \leq || \theta_t-\theta_{t}^* ||^2_2+ \alpha^2 ||\delta_t z_t||^2_2+|| \theta_{t}^*-\theta_{t+1}^*||_2^2
> $
>
> $+ 2 \alpha\langle\theta_t-\theta_{t}^*, \delta_t z_t \rangle  + 2\alpha \langle \delta_t z_t, \theta_{t}^* - \theta_{t+1}^* \rangle+ 2 \langle\theta_t-\theta_{t}^*, \theta_{t}^*-\theta_{t+1}^* \rangle.$
>
> Another key challenge lies in  how to bound the term
> $\mathbb E[\langle \theta_t-\theta_t^*, \delta_t z_t\rangle ] $.  We develop a novel technique of auxiliary Markov chain to decompose this error into two parts: 1) error due to time-varying feature function and 2) error due to time-varying policy. Specifically, consider the first Markov chain generated from the algorithm:
> $$s_0,a_0 \overset{\pi_0\times P}{\to} s_1,a_1\to...\to s_t,a_t \overset{\pi_t\times P}{\to}s_{t+1},a_{t+1},$$ where at each time $j$, the action is chosen according to $\pi_j$ and the transition kernel is $P$. Here $z_t=\sum_{j=t-k}^t \phi_j(s_j,a_j)$ is the eligibility trace used in the algorithm. It can be seen that in $z_t$, the feature $\phi_j$ changes with $j$, and the distribution of $s_j,a_j$ depends on the time-varying policy $\pi_j$. We then design an auxiliary eligibility trace $\hat z_t=\sum_{j=t-k}^t \phi_t(s_j,a_j)$, where the feature is fixed to be $\phi_t$, and only the the distribution of $s_j,a_j$ depends on the time-varying policy $\pi_j$. To further handle the time-varying distribution of  $s_j,a_j$, we design an auxiliary Markov chain (denoted by $A1$) as follows:
> $$A1: ( s_0,\widetilde  a_0 )\sim \pi_t\overset{\pi_t\times P}{\to} \widetilde s_1,\widetilde a_1\overset{\pi_t\times P}{\to}...\overset{\pi_t\times P}{\to}\widetilde  s_t,\widetilde a_t \overset{\pi_t\times P}{\to}\widetilde s_{t+1},\widetilde a_{t+1},$$ where the action at each time $j$ is always chosen according to a fixed policy $\pi_t$. Based on this auxiliary Markov chain, we introduce another auxiliary eligibility trace $\widetilde  z_t=\sum_{j=t-k}^t \phi_t(\widetilde s_j,\widetilde a_j)$, where it uses a fixed feature $\phi_t$, and samples from this auxiliary Markov chain. Lastly, we design a second auxiliary Markov chain (denoted by $A2$):
> $$A2: (\bar s_0,\bar  a_0)\sim D_t \overset{\pi_t\times P}{\to} \bar s_1,\bar a_1\overset{\pi_t\times P}{\to}...\overset{\pi_t\times P}{\to}\bar  s_t,\bar a_t \overset{\pi_t\times P}{\to}\bar s_{t+1},\bar a_{t+1}$$
> where the only difference between A2 and A1 lies in the initial state distribution. Then we define the last auxiliary eligibility trace as $\bar  z_t=\sum_{j=t-k}^t \phi_t(\bar s_j,\bar a_j)$.

---

> > ### Author Response · Authors · 2023-11-18
> > **part 2 continued**
> >
> > The difference between $z_t $ and $\hat z_t$ measures the error due to the time-varying compatible feature function. We bound this error using the Lipschitz continuity of the feature function.  The difference between $\hat z_t $ and $\widetilde z_t$ measures the error due to the  time-varying sampling policy.   The difference between $\widetilde z_t $ and $\bar z_t$ measures the error due to the difference between the stationary distribution and the actual distribution of the samples, which can be bounded based on Assumption 1. By such a error decomposition, we can show that
> > \begin{align}
> >     \mathbb E[||\theta_{t+1}-\theta^*_{t+1}||^2_2] \leq (1-\bar \lambda_{\min}\alpha /2)\mathbb E[||\theta_{t}-\theta^*_{t}||^2_2]
> >     +\mathcal{O}(k^2 \alpha \mathbb E[(\eta_t-J(\omega_t))^2] )+\mathcal{O}(\beta\mathbb E [||\nabla J(\omega_t)||_2^2])+ \mathcal{O}(k^2 \alpha^2+k^3\beta^2+m\rho^k\alpha ) .
> > \end{align}
> >
> > **Step 4 (Bound on gradient):** As we can see from Steps 2 and 3, we bound the estimation error of the average reward and the tracking error using the gradient norm $\|\nabla J(\omega_t)\|_2^2$. Therefore,  in order to derive the tightest bound, we further develop a novel bound on the gradient norm $\|\nabla J(\omega_t)\|_2^2$. Note that the idea is novel as it serves as a pivotal link connecting the analysis of the tracking error/estimation error in the average reward and the optimality gap. Specifically, we bound the gradient norm using
> > the estimation error in the average reward and tracking error. By the smoothness of $J(\omega)$, we have that
> >
> > $$
> >     \frac{1}{\widetilde T}\sum_{j=t}^{t+\widetilde T-1}\mathbb E [ ||\nabla J(\omega_j) ||_2^2]
> > $$
> >
> > $$ \leq 2C_\phi \frac{J(\omega^*)- E [J(\omega_{t})]}{\beta \widetilde T}+ \mathcal O\left(\frac{1}{\widetilde T}\sum_{j=t}^{t+\widetilde T-1}\mathbb E[||\theta_j-\theta_j^*||_2^2]\right) +\mathcal{O}(m\rho^k+\beta).
> > $$
> >
> > We also note that we bound the gradient norm using the optimality gap, and this is of great importance to establish the tight bound in this paper. In previous works, this  term $\mathbb E [J(\omega_{t+\widetilde T})]- E [J(\omega_{t})] $ is bounded by a constant, and thus the overall complexity is not as tight.
> >
> > **Step 5:** Combining steps 1-4, we conclude the proof.

---

> ### Author Response · Authors · 2023-11-18
> **part 3**
>
> **Q 2: Importance of eliminating $\varepsilon_{\text{critic}}$ when it can be small in neural function approximation.**
>
> One may argue that using a rich (non-linear) function approximation in the critic, e.g., neural network, may also have a small function approximation error in the critic. However, as shown in the literature, e.g., (Cai et.al.,2023), TD with non-linear function approximation may require a strong condition to guarantee convergence. In contrast, using compatible function approximation in the critic has a guaranteed convergence. Moreover, as discussed in eq. (8), there is no need to estimate the inverse Fisher information matrix if we use compatible function approximation in NAC. This could significantly reduce the complexity of implementing the algorithm.
>
> We are currently working on the suggested simulations, and will add them to the final version.
>
> Cai, Q., Yang, Z., Lee, J. D.,  Wang, Z. (2023). Neural Temporal Difference and Q Learning Provably Converge to Global Optima. Mathematics of Operations Research.

---

> > ### Comment · Reviewer_CZ8K · 2023-11-23
> >
> > Thanks for the detailed response. I have increased my confidence to 3.

---

### Official Review · Reviewer_mPQr · 2023-11-01

**Soundness:** 3 good
**Presentation:** 3 good
**Contribution:** 3 good
**Rating:** 6
**Confidence:** 4

**Summary:**

This paper provides the tightest non-asymptotic convergence bounds for both the AC and natural AC (NAC) algorithms. Specifically, existing studies show that AC converges to an $\epsilon+\epsilon_{critic}$ neighborhood of stationary points with the best known sample complexity of $O(\epsilon^{-2})$, and NAC converges to an $\epsilon+\epsilon_{critic}+\sqrt{\epsilon_{actor}}}$ neighborhood of the global optimum with the best known sample complexity of $O(\epsilon^{-3})$, where $\epsilon$-critic is the approximation error of the critic and εactor is the approximation error induced by the insufficient expressive power of the parameterized policy class. This paper analyzes the convergence of both AC and NAC algorithms with compatible function approximation. The major technical novelty lies in analyzing the stochastic bias due to policy-dependent and time-varying compatible function approximation in the critic, and handling the non-ergodicity of the MDP due to the single Markovian sample trajectory.

**Strengths:**

This paper focuses on the challenging single-loop setting with a single Markovian sample trajectory. To develop the tightest bound, this paper develops a novel approach that bounds the tracking error as a function of the policy gradient norm (for AC) and the optimality gap (for NAC). Their analysis for NAC does not need the smoothness assumption on the parameterized policy, which is typically required in existing NAC and AC analyses.

**Weaknesses:**

1. I don't understand the second equal sign of equation (5), why it holds true? Do you need to assume $Q^{\pi_w}=\phi^\top_w\bar{\theta}^*_w$? Also, given equation (5), it is unclear why "This implies that as long as we can solve the finite dimensional problem Equation (4), linear function approximation with the compatible feature and parameter does not induce any function approximation error." could you elaborate on that?

2. why $\phi^\top\phi$ is a matrix in eqn(8)? in this case $\phi^\top \theta$ is also a matrix instead of a scalar?

3. For NAC analysis, you need an additional Assumption 3. Could you explain why this is needed for NAC but not for AC? Also, how is this related to the single concentrability coefficient defined for offline RL (def1 of [1], assumption3 in [2])?

[1] Bridging offline reinforcement learning and imitation learning: A tale of pessimism, NeurIPS21,
[2] Towards Instance-Optimal Offline Reinforcement Learning with Pessimism, NeurIPS21.

4. You mentioned "Our major technical novelty lies in analyzing the stochastic bias due to policy-dependent and time-varying compatible function approximation in the critic, and handling the non-ergodicity of the MDP due to the single Markovian sample trajectory." Where can I find the detail of this novelty in the paper?

**Questions:**

Please answer the questions above.

---

> ### Author Response · Authors · 2023-11-18
> **Part 1**
>
> **Q1: Clarification on eq. (5) and the statement of no critic function approximation error.**
>
> Thanks for the suggestion. We have added the proof of eq. (5) in the revision. For convenience, we also include the proposition and the proof here.
>
>
> **Proposition 1**: With compatible function approximation, the policy gradient $\nabla J(\pi_\omega)$ can be rewritten as:
> $$
>  \nabla J(\pi_\omega) =\mathbb E_{D_{\pi_\omega}}[ \nabla \log \pi_\omega(a|s)Q^{\pi_\omega}(s,a)]=
>  \mathbb E_{D_{\pi_\omega}}[\phi_\omega(s,a)(\phi^\top_\omega(s,a)\bar\theta_\omega^*)].$$
>  The natural policy gradient can be written as:
> $$\widetilde{\nabla} J(\pi_\omega)=\bar \theta^*_\omega.$$
>
> **Proof:**
> By eq. (4), $\bar \theta^*_\omega$ is the minimizer of  eq. (4) and therefore it should satisfy that
> $$
> \mathbb E_{D_{\pi_\omega}}[(Q^{\pi_\omega}(s,a)-\phi^\top_\omega(s,a)\bar \theta_\omega^*)\phi_\omega(s,a) ]=0.
> $$
> This further implies that
> $$\mathbb E_{D_{\pi_\omega}}[Q^{\pi_\omega}(s,a)\phi_\omega(s,a) ]=\mathbb E_{D_{\pi_\omega}}[\phi_\omega(s,a)\phi^\top_\omega(s,a)\bar \theta_\omega^*].
> $$
> Now we consider the policy gradient $\nabla J(\pi_\omega)$, and it  can be shown that
> $$
> \nabla J(\pi_\omega) =\mathbb E_{D_{\pi_\omega}}[ \nabla \log \pi_\omega(a|s)Q^{\pi_\omega}(s,a)]=
>  \mathbb E_{D_{\pi_\omega}}[\phi_\omega(s,a)
>  (\phi^\top_\omega(s,a)\bar\theta_\omega^*)].
>  $$
>
> Furthermore, for the natural policy gradient, we have that $$\widetilde{\nabla} J(\pi_\omega)= F_{\omega}^{-1} \mathbb E_{D_{\pi_\omega}}[\phi_\omega(s,a)\phi_\omega(s,a)^\top{\bar\theta}^*_\omega]=F_\omega^{-1} \mathbb E_{D_{\pi_\omega}}[\phi_\omega(s,a)\phi_\omega(s,a)^\top   ]\bar \theta^*_\omega=\bar \theta^*_\omega.
> $$
>
>
> **Q2: Why is $\phi^\top\phi$ a matrix in eq. (8)? In this case $\phi^\top\theta$  is also a matrix instead of a scalar?**
>
> We would like to thank the reviewer for this comment. This is actually a typo. In eq. (8), it should be $\phi\phi^\top$. We have fixed this error.
>
>
> **Q3.1: Why is Assumption 3 only needed for NAC analysis?**
>
> This is because for the AC algorithm, we establish the convergence to stationary point, i.e., the gradient of $J(\omega_t)$ converges to $0$, and for the NAC algorithm, we establish the convergence to the optimal policy, i.e., $J(\omega_t)$ converges to $J(\pi^*)$. For AC, as the critic performs on-policy learning,  it is able to learn the value function of the current policy $\pi_{\omega_t}$, and thus the gradient  $\nabla J(\omega_t)$. However, for NAC, in order to measure the difference between the current policy and the optimal policy, we need the current policy to be sufficiently explorative so that samples from the current policy $\pi_{\omega_t}$ can be used to infer the optimal policy. This is guaranteed by Assumption 3.
>
> **Q3.2: How is Assumption 3 related to the single concentrability coefficient in offline RL?**
>
> Definition 1 in [1] and Assumption 2.3 in [2] define the range of policies that a given offline dataset could learn since offline reinforcement learning does not allow exploration. Assumption 3  guarantees that the policy is explorative so that the optimal policy could be learned. They are quite related. But here we could design the way that the policy is parameterized to guarantee Assumption 3, whereas in offline RL, the behavior policy cannot be changed.
>
> [1] Rashidinejad P, Zhu B, Ma C, et al. Bridging offline reinforcement learning and imitation learning: A tale of pessimism[J]. Advances in Neural Information Processing Systems, 2021, 34: 11702-11716.
>
>  [2] Yin M, Wang Y X. Towards instance-optimal offline reinforcement learning with pessimism[J]. Advances in neural information processing systems, 2021, 34: 4065-4078.

---

> ### Author Response · Authors · 2023-11-18
> **part 2**
>
> **Q4:  Details of this paper's novelty.**
> Here we provide a proof sketch for the NAC algorithm to highlight major challenges and our technical novelties. The analysis of NAC contains of most major technical novelty in the AC analysis.
>
>
> **Proof Sketch:**
> For simplicity of presentation, we set $\alpha_t=\alpha,\beta_t=\beta,\gamma_t=\gamma$, $\hat t=\lceil \frac{3\log T}{\bar \lambda_{\min}\alpha}\rceil$ and
> $\widetilde T=\hat t\lceil \frac{T}{\hat t \log T} \rceil$.
>  We denote by $\pi^*$ the global optimal policy and $M_t=\mathbb E[||\theta_{t}-\theta^*_{t}||^2_2]+ \mathbb E[(\eta_t-J(\omega_t))^2]$ the sum of the tracking error and the estimation error of the average reward. Denote by $D(\omega_t)=KL(\pi^*|\pi_t)$ the KL divergence between policy $\pi^*$ and $\pi_t$.
>
>
> **Step 1 (Error decomposition):**
> According to the smoothness property of $D(\omega)$ with respect to $\omega$, we bound the performance gap between the current policy and the optimal policy (optimality gap) as follows:
> \begin{align}
>     \frac{1}{\widetilde T}\sum_{j=t}^{t+\widetilde T-1}\mathbb E[J(\pi^*)-J(\omega_j)]
>      \leq \frac{D(\omega_{t+\widetilde T})-D(\omega_t)}{\widetilde T \beta }+ \mathcal{O}\left(\sqrt{\frac{1}{\widetilde T}\sum_{j=t}^{t+\widetilde T-1} M_j}\right)+\mathcal{O}(C_\infty \sqrt{\varepsilon_{\text{actor}}}+ \beta+m\rho^k).
> \end{align}
>
>
>
>
> **Step 2 (Estimation error in the average reward):** In this step, we analyze estimation error in the average reward:
>  $\eta_t-J(\omega_t)$. We provide a tight characterization of this error:
> \begin{align}
>     \mathbb E[(\eta_{t+1}-J(\omega_{t+1}))^2] \leq (1-\gamma)\mathbb E[(\eta_{t}-J(\omega_{t}))^2] + \mathcal{O}(\beta \mathbb E[||\nabla J(\omega_t)||_2^2])
>     + \mathcal{O}( m\rho^k\gamma +k^2 \gamma^2 +k^2 \beta^2).
> \end{align}
> One of our key novelties lies in that we bound this estimation error using the gradient norm $\mathbb E[||\nabla J(\omega_t)||_2^2]$. The above bound itself is tighter than the existing one in (Wu et al., 2020).
>
>
>
>
> **Step 3 (Tracking error)**:
> In this step, we bound the tracking error in the critic: $||\theta_t-\theta_t^*||_2^2$.
>
> By the TD error step in Algorithm 1, we decompose the term $||\theta_{t+1}-\theta_{t+1}^*||^2_2$ as follows:
>
> $    ||\theta_{t+1}-\theta_{t+1}^*||^2_2 \leq || \theta_t-\theta_{t}^* ||^2_2+ \alpha^2 ||\delta_t z_t||^2_2+|| \theta_{t}^*-\theta_{t+1}^*||_2^2
> $
>
> $+ 2 \alpha\langle\theta_t-\theta_{t}^*, \delta_t z_t \rangle  + 2\alpha \langle \delta_t z_t, \theta_{t}^* - \theta_{t+1}^* \rangle+ 2 \langle\theta_t-\theta_{t}^*, \theta_{t}^*-\theta_{t+1}^* \rangle.$
>
> Another key challenge lies in  how to bound the term
> $\mathbb E[\langle \theta_t-\theta_t^*, \delta_t z_t\rangle ] $.  We develop a novel technique of auxiliary Markov chain to decompose this error into two parts: 1) error due to time-varying feature function and 2) error due to time-varying policy. Specifically, consider the first Markov chain generated from the algorithm:
> $$s_0,a_0 \overset{\pi_0\times P}{\to} s_1,a_1\to...\to s_t,a_t \overset{\pi_t\times P}{\to}s_{t+1},a_{t+1},$$ where at each time $j$, the action is chosen according to $\pi_j$ and the transition kernel is $P$. Here $z_t=\sum_{j=t-k}^t \phi_j(s_j,a_j)$ is the eligibility trace used in the algorithm. It can be seen that in $z_t$, the feature $\phi_j$ changes with $j$, and the distribution of $s_j,a_j$ depends on the time-varying policy $\pi_j$. We then design an auxiliary eligibility trace $\hat z_t=\sum_{j=t-k}^t \phi_t(s_j,a_j)$, where the feature is fixed to be $\phi_t$, and only the the distribution of $s_j,a_j$ depends on the time-varying policy $\pi_j$. To further handle the time-varying distribution of  $s_j,a_j$, we design an auxiliary Markov chain (denoted by $A1$) as follows:
> $$A1: ( s_0,\widetilde  a_0 )\sim \pi_t\overset{\pi_t\times P}{\to} \widetilde s_1,\widetilde a_1\overset{\pi_t\times P}{\to}...\overset{\pi_t\times P}{\to}\widetilde  s_t,\widetilde a_t \overset{\pi_t\times P}{\to}\widetilde s_{t+1},\widetilde a_{t+1},$$ where the action at each time $j$ is always chosen according to a fixed policy $\pi_t$. Based on this auxiliary Markov chain, we introduce another auxiliary eligibility trace $\widetilde  z_t=\sum_{j=t-k}^t \phi_t(\widetilde s_j,\widetilde a_j)$, where it uses a fixed feature $\phi_t$, and samples from this auxiliary Markov chain. Lastly, we design a second auxiliary Markov chain (denoted by $A2$):
> $$A2: (\bar s_0,\bar  a_0)\sim D_t \overset{\pi_t\times P}{\to} \bar s_1,\bar a_1\overset{\pi_t\times P}{\to}...\overset{\pi_t\times P}{\to}\bar  s_t,\bar a_t \overset{\pi_t\times P}{\to}\bar s_{t+1},\bar a_{t+1}$$
> where the only difference between A2 and A1 lies in the initial state distribution. Then we define the last auxiliary eligibility trace as $\bar  z_t=\sum_{j=t-k}^t \phi_t(\bar s_j,\bar a_j)$.

---

> > ### Author Response · Authors · 2023-11-18
> > **Part 2 continued**
> >
> > The difference between $z_t $ and $\hat z_t$ measures the error due to the time-varying compatible feature function. We bound this error using the Lipschitz continuity of the feature function.  The difference between $\hat z_t $ and $\widetilde z_t$ measures the error due to the  time-varying sampling policy.   The difference between $\widetilde z_t $ and $\bar z_t$ measures the error due to the difference between the stationary distribution and the actual distribution of the samples, which can be bounded based on Assumption 1. By such a error decomposition, we can show that
> > \begin{align}
> >     \mathbb E[||\theta_{t+1}-\theta^*_{t+1}||^2_2] \leq (1-\bar \lambda_{\min}\alpha /2)\mathbb E[||\theta_{t}-\theta^*_{t}||^2_2]
> >     +\mathcal{O}(k^2 \alpha \mathbb E[(\eta_t-J(\omega_t))^2] )+\mathcal{O}(\beta\mathbb E [||\nabla J(\omega_t)||_2^2])+ \mathcal{O}(k^2 \alpha^2+k^3\beta^2+m\rho^k\alpha ) .
> > \end{align}
> >
> > **Step 4 (Bound on gradient):** As we can see from Steps 2 and 3, we bound the estimation error of the average reward and the tracking error using the gradient norm $\|\nabla J(\omega_t)\|_2^2$. Therefore,  in order to derive the tightest bound, we further develop a novel bound on the gradient norm $\|\nabla J(\omega_t)\|_2^2$. Note that the idea is novel as it serves as a pivotal link connecting the analysis of the tracking error/estimation error in the average reward and the optimality gap. Specifically, we bound the gradient norm using
> > the estimation error in the average reward and tracking error. By the smoothness of $J(\omega)$, we have that
> >
> > $$
> >     \frac{1}{\widetilde T}\sum_{j=t}^{t+\widetilde T-1}\mathbb E [ ||\nabla J(\omega_j) ||_2^2]
> > $$
> >
> > $$ \leq 2C_\phi \frac{J(\omega^*)- E [J(\omega_{t})]}{\beta \widetilde T}+ \mathcal O\left(\frac{1}{\widetilde T}\sum_{j=t}^{t+\widetilde T-1}\mathbb E[||\theta_j-\theta_j^*||_2^2]\right) +\mathcal{O}(m\rho^k+\beta).
> > $$
> >
> > We also note that we bound the gradient norm using the optimality gap, and this is of great importance to establish the tight bound in this paper. In previous works, this  term $\mathbb E [J(\omega_{t+\widetilde T})]- E [J(\omega_{t})] $ is bounded by a constant, and thus the overall complexity is not as tight.
> >
> > **Step 5:** Combining steps 1-4, we conclude the proof.

---

### Official Review · Reviewer_k8iS · 2023-11-02

**Soundness:** 4 excellent
**Presentation:** 3 good
**Contribution:** 3 good
**Rating:** 6
**Confidence:** 2

**Summary:**

This paper conducts a convergence analysis of actor-critic and natural actor-critic algorithms with compatible function approximation, under the single-loop setting that uses onse sample trajectory. The paper presents the tightest convergence bound in both cases.

**Strengths:**

- The paper presents the tightest non-asymptotic bound for the covergenace of AC and NAC algorithms, compared to existing work. The results show that compatible function approximation
- To obtain the tightes bound, paper presents novel technical contributions to analyze the single-trajectory setting and avoid decoupling of the actor and critic updates. The setting considered in this paper is closer to practice.

**Weaknesses:**

- The main contributions of the paper are limited to novel techniques in the analysis of well-known algorthms and theoretical evidence on why compatible function approximation might be advantageous. I am uncertain about potential impact of this work either on practice or theory.

**Questions:**

Could you please clarify whether/how the techniques or results presented in this paper could be useful in practice, for instance, through the development of new algorithmic ideas, or in theory, such as analysis techniques being potentially useful to analyze other algorithms?

---

> ### Author Response · Authors · 2023-11-17
>
> **Q1: Practical and theoretical impact: how can the algorithmic idea and theoretical analysis be useful to develop and analyze other algorithms.**
>
> The algorithmic idea of compatible function approximation can be applied to a wide range of AC and NAC type algorithms e.g., in the multi-task, multi-agent and game settings,  to get rid of the function approximation error in the critic. One may argue that using a rich (non-linear) function approximation in the critic, e.g., neural network, may also have a small function approximation error in the critic. However, as shown in the literature, e.g., (Cai et.al.,2023), TD with non-linear function approximation may require a strong condition to guarantee convergence. In contrast, using compatible function approximation in the critic has a guaranteed convergence. Moreover, as discussed in eq. (8), there is no need to estimate the inverse Fisher information matrix if we use compatible function approximation in NAC. This could significantly reduce the complexity in extensions to e.g., the multi-agent and distributed setting, where the estimate of the Fisher information matrix may be challenging.
>
> Our theoretical analysis can also be generalized to understand the convergence and sample complexity of various extensions of AC and NAC type algorithms e.g., in the multi-task, multi-agent, distributed and game settings. Our approach of analyzing the stochastic bias due to policy-dependent and time-varying compatible function approximation in the critic, and handling the non-ergodicity of the MDP due to the single Markovian sample trajectory can be used to obtain a tight bound of those generalizations of AC and NAC algorithms.
>
>
> Cai, Q., Yang, Z., Lee, J. D.,  Wang, Z. (2023). Neural Temporal Difference and Q Learning Provably Converge to Global Optima. Mathematics of Operations Research.

---

### Meta-Review · Area_Chair_Mtqs · 2023-12-05

**Metareview:**

The reviewers are in agreement regarding the technical innovations and strong contributions; however, the lack of experimental corroboration of the key refinements in the theory makes it difficult to assess whether the single-loop algorithm with compatible function approximation actually realize practical gains. It is hard to see what can be gained by the RL community via another analysis-only work on actor-critic method. To be more specific, the reviewers have raised concerns about the additional assumptions imposed by this work to achieve global optimality, and while these concerns are addressed in theory, they imply function complexity measures that are not estimable in practice, and there is no experimental guidance on the usefulness of the proposed NAC scheme and how compatible function approximation really operates. The authors need to do a better job of explaining the technical novelties of the proposed error decomposition as well as role of the modified assumptions, both in theory and practice, that distinguishes the proposed algorithms from prior art.

**Justification For Why Not Higher Score:**

Lack of experiments whatsoever. Insufficient explanation of why the contributed analysis and technical setting are innovative. The authors should expend more effort to delineate the technical novelty of the proposed error decomposition, assumptions, and their role in practice to improve performance in experiments.

**Justification For Why Not Lower Score:**

NA

---

### Decision · Program_Chairs · 2024-01-16

Reject